# A quantitative modelling approach to zebrafish pigment pattern formation

Jennifer P Owen*, Robert N Kelsh†, Christian A Yates†*

Department of Biology and Biochemistry and Department of Mathematical Sciences, University of Bath, Claverton Down, Bath, United Kingdom

**Abstract** Pattern formation is a key aspect of development. Adult zebrafish exhibit a striking striped pattern generated through the self-organisation of three different chromatophores. Numerous investigations have revealed a multitude of individual cell-cell interactions important for this self-organisation, but it has remained unclear whether these known biological rules were sufficient to explain pattern formation. To test this, we present an individual-based mathematical model incorporating all the important cell-types and known interactions. The model qualitatively and quantitatively reproduces wild type and mutant pigment pattern development. We use it to resolve a number of outstanding biological uncertainties, including the roles of domain growth and the initial iridophore stripe, and to generate hypotheses about the functions of *leopard*. We conclude that our rule-set is sufficient to recapitulate wild-type and mutant patterns. Our work now leads the way for further in silico exploration of the developmental and evolutionary implications of this pigment patterning system.

*For correspondence:
jpo22@bath.ac.uk (JPO);
c.yates@bath.ac.uk (CAY)

†These authors contributed equally to this work

**Competing interests:** The authors declare that no competing interests exist.

## Introduction

Pattern formation - the process generating regular features from homogeneity - is a fascinating phenomenon that is as ubiquitous as it is diverse. It is a major aspect of developmental biology, with key exemplars including segmentation within the syncitial blastoderm of fruit flies (*Clark and Peel, 2018*), digit formation in the vertebrate limb (*Tickle, 2006*), and branching patterns in kidney and lung development (*Davies, 2002*).

Another key example, pigment pattern formation, the process generating functional and often beautiful distributions of pigment cells, represents a classic problem in both developmental and mathematical biology. Pigment patterns allow animals to distinguish between individuals within a group and identify those of different species and are an important characteristic for the survival of most animals in wild populations. Pigment patterns are striking. They form rapidly and, in many cases, autonomously, that is, the process relies on self-organisation and not internal body structures. Additionally, they often vary dramatically between even closely related species, therefore recognising similarities and differences in the development of these related species can allow us insight into the evolutionary change. Finally, pigment pattern formation is made experimentally tractable by the self-labelling nature of pigment cells.

The horizontal blue and gold stripes of zebrafish are now one of the best-studied examples of pigment pattern formation, especially at the level of underlying cellular mechanisms (*Singh and Nüsslein-Volhard, 2015*; *Watanabe and Kondo, 2015*; *Patterson and Parichy, 2019*). Zebrafish are amenable to observational studies, since all development takes place outside the mother and the skin is transparent. This, combined with the availability of multiple key mutants (affecting, for example, cell-type differentiation and patterning), and the development of innovative in vivo cell ablation and in vitro cell culture techniques, have provided a unique opportunity to investigate the cellular and molecular basis for pigment pattern formation experimentally (*Eom et al., 2015*; *Budi et al., 2011*; *Ceinos et al., 2015*; *Yamanaka and Kondo, 2014*; *Eom and Parichy, 2017*; *Hamada et al.,*

*2014*; *Fadeev et al., 2015*; *Inoue et al., 2014*; *Irion et al., 2014*; *Watanabe et al., 2006*; *Frohnhöfer et al., 2013*; *Parichy et al., 2009*; *Svetic et al., 2007*; *Mellgren and Johnson, 2006*; *Parichy et al., 2000b*; *Iwashita et al., 2006*; *Hirata et al., 2005*; *Kelsh et al., 1996*; *Lister et al., 1999*; *Parichy et al., 2000a*; *Maderspacher and Nüsslein-Volhard, 2003*; *Walderich et al., 2016*; *Patterson et al., 2014*; *McMenamin et al., 2014*; *Patterson and Parichy, 2013*; *Krauss et al., 2014*; *Parichy and Turner, 2003*; *Eom et al., 2012*; *Mahalwar et al., 2016*; *Mahalwar et al., 2014*; *Asai et al., 1999*; *Takahashi and Kondo, 2008*).

The cellular composition of the stripes and how these become assembled has been well-described. Zebrafish generate, over a period of a few weeks and beginning around 3 weeks of age (*Frohnhöfer et al., 2013*; *Parichy et al., 2009*), a robust adult stripe pattern of alternating dark blue stripes and golden interstripes comprised of three different pigment-producing cell types: melanocytes, containing black melanin; xanthophores containing yellow and orange carotenoids and pteridines; and iridescent iridophores, containing guanine crystals within reflective platelets (*Hirata et al., 2003*; *Figure 1A*).

Of the iridophores, there are two types distinguished by their platelet distribution (*Hirata et al., 2003*); type L-iridophores, and type S-iridophores. Only type S-iridophores play a role in stripe formation (type L-iridophores appear later and are likely involved in pattern maintenance [*Hirata et al., 2003*]), and they appear in two different forms. In the light interstripes, S-iridophores appear in a 'dense' arrangement (dense S-iridophores), forming a continuous sheet, whilst in the dark stripes the cells are in a 'loose' arrangement and appear more widely spaced (loose S-iridophores) (*Hirata et al., 2003*; *Fadeev et al., 2015*). Pigment cells are found in the hypodermis below the dermis, and organised as layers of cells consistently stacked in the same order (*Hirata et al., 2003*). Starting from the deepest layer of the hypodermis just above the muscle and moving to the dermis, adult dark stripes consist of consecutive layers of L-iridophores, melanocytes, loose S-iridophores

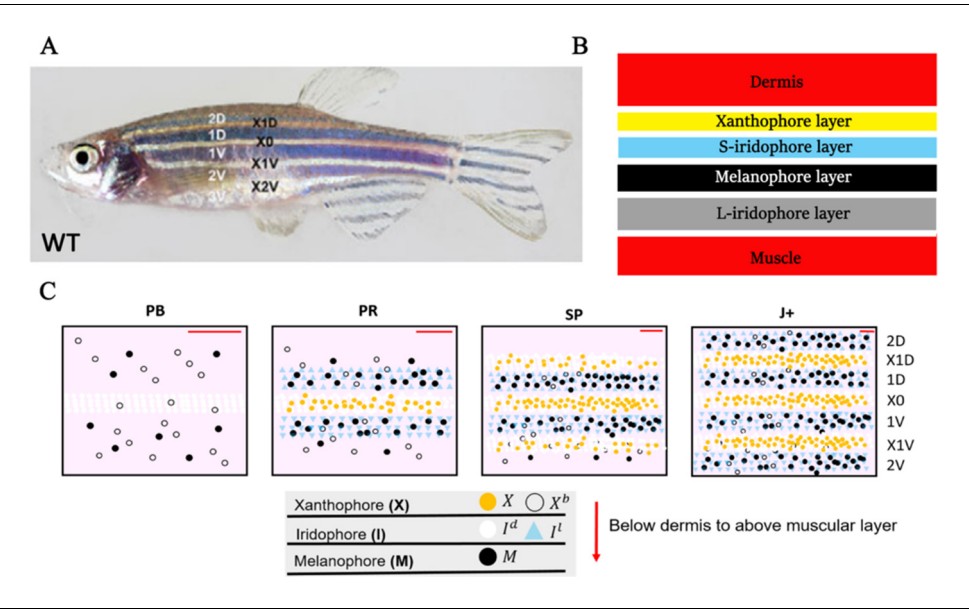

**Figure 1.** WT stripe composition and development. (**A**) An adult wild type (WT) fish. Stripes and interstripes are labelled according to their order of temporal appearance. X0 is the first interstripe to appear. 1D and 1V (D - Dorsal, V- Ventral) are the first two stripes to appear. X1D and X1V are the next two interstripes to appear and so on. Image reproduced from *Frohnhöfer et al., 2013* and licensed under CC-BY 4.0 (https://creativecommons.org/licenses/by/4.0). (**B**) Summary of pigment cell distribution in adult zebrafish. The cells in the xanthophore, S-iridophore, melanophore and L-iridophore layers consist of xanthophores and xanthoblasts, melanophores, S-iridophores and L-iridophores, respectively. Adapted from *Hirata et al., 2003*. (**C**) Schematic of WT patterns on the body of zebrafish. Stages PB, PR, SP, J+ correspond to developmental stages described in 3. Patterns form sequentially outward from the central interstripe, labelled X0, with additional dorsal stripes and interstripes labelled 1D, X1D, 2D, X2D from the centre (horizontal myoseptum) dorsally outward (similarly, ventral stripes and interstripes are labelled 1V, X1V, 2V, etc).

and xanthophores. Similarly, adult light interstripes are made up of layers of dense S-iridophores and xanthophores (*Figure 1B*; *Hirata et al., 2003*). The final striped pattern is generated by the self-organisation of xanthophores, melanocytes, loose and dense S-iridophores into the appropriate positions within the hypodermis.

Prior to the initiation of adult stripe formation zebrafish exhibit a larval pigment pattern, formed in the first 5 days of development. Embryonic pigment cells form a distinctive early larval pattern that is essentially complete by 5 days post-fertilisation (dpf) and remains unchanged until metamorphosis. This pattern consists of melanocytes in four stripes (dorsal to the central nervous system, within the horizontal myoseptum, dorsal to the gut and ventrally under the yolk; S-iridophores are found associated with three of these melanocyte stripes [*Parichy et al., 2009*; *Frohnhöfer et al., 2013*]). Xanthophores lie in a monolayer under the skin, filling the areas between the melanocytes above the CNS and extending ventrally to the level of the gut. Formation of the adult pattern involves replacement of melanocytes and S-iridophores with new cells derived from adult pigment stem cells. Early larval xanthophores dedifferentiate, forming unpigmented xanthoblasts that regain their proliferative ability and proceed to generate the adult xanthophores (*McMenamin et al., 2014*; *Mahalwar et al., 2014*; *Parichy et al., 2000b*); an unknown proportion of the latter may derive from de novo production from adult pigment stem cells (*Kelsh et al., 2017*). Xanthophore de-differentiation is complete by 21dpf, and early metamorphic melanocytes appear in a widely scattered distribution between 14 and 21dpf (*Parichy et al., 2009*), thus forming the initial metamorphic pattern (*Figure 1C*, stage PB). A key event in the initiation of adult pattern metamorphosis is the appearance of newly differentiated dense S-iridophores alongside the horizontal myoseptum. In response to the appearance of these S-iridophores, the first adult xanthophores are generated (*Figure 1C*, stage PR) by differentiation from xanthoblasts in this region, thus initiating the first interstripe, X0. Furthermore, metamorphic melanocytes begin to accumulate either side of this central interstripe, marking the first two stripes denoted 1D and 1V (*Figure 1C*, stage PR). Subsequently, S-iridophores proliferate rapidly and spread bidirectionally; at the edges of the interstripes they switch to a more scattered (less tightly-packed) form as they continue to spread dorsally and ventrally. Spreading loose S-iridophores transition back into dense S-iridophores at the locations of the future interstripes X1V and X1D (*Figure 1C*, stages PR to SP) (*Mahalwar et al., 2014*). Once S-iridophores aggregate at the next interstripe, the process starts again, that is, xanthophores differentiate in response to the dense S-iridophores and melanocytes accumulate either side of the new interstripe generating the subsequent stripe. This process of S-iridophore aggregation predetermining future interstripe locations and subsequent delamination in future stripe regions repeats until S-iridophores cover the domain and all stripes (between 4 and 5) and interstripes are fully formed (*Figure 1C*, stage J+).

In addition to the description of pattern development (*Frohnhöfer et al., 2013*), many studies have identified individual patterning mechanisms that contribute to stripe formation, although it is unclear whether these are sufficient to explain pattern formation. Stripe generation is complex and requires many interactions. During pattern metamorphosis, these interactions may determine cell birth (*Mahalwar et al., 2014*), cell death (*Takahashi and Kondo, 2008*), cell migration (*Yamanaka and Kondo, 2014*; *Takahashi and Kondo, 2008*; *Patterson et al., 2014*), long-distance communication, through stabilisation of elongated cellular projections (*Eom and Parichy, 2017*; *Eom et al., 2015*), as well as the shape transitions of S-iridophores (*Fadeev et al., 2015*). During this period, there is also simultaneous two-dimensional domain growth (*Parichy et al., 2009*). The pattern is formed by cell–cell interactions of all three pigment producing cell types: melanocytes, xanthophores and S-iridophores. Without any one of these cell types, pattern formation is disrupted (*Frohnhöfer et al., 2013*; *Patterson and Parichy, 2013*).

Mathematical modelling has been a complementary tool in assessing possible patterning mechanisms. Until the last few years, these studies have focused on melanocytes and xanthophores, neglecting S-iridophores. The most commonly used mathematical paradigm for stripe formation takes the form of a Turing reaction-diffusion model. In these representations, melanocytes and xanthophores diffuse and interact via a few long- and short-range 'reactions'. This class of model typically rely on a small number of parameters which, upon being altered, can generate a diverse range of patterns. Minimal models such as these have the benefit that they are sometimes analytically tractable, allowing a deep understanding of the model. However, a potential limitation is that parameters do not always have a clear biological interpretation which, can sometimes make it difficult to

link parameters to measurable data. In the context of zebrafish stripe formation, these models have not yet incorporated S-iridophores (*Watanabe and Kondo, 2015*; *Kondo, 2017*; *Painter et al., 2015*; *Bloomfield et al., 2011*; *Binder and Simpson, 2013*; *Volkening and Sandstede, 2015*; *Kondo, 2017*; *Nakamasu et al., 2009*; *Moreira and Deutsch, 2005*; *Bullara and De Decker, 2015*; *Yamaguchi et al., 2007*; *Asai et al., 1999*). They suggest that the role for iridophores is restricted to simply orienting stripes (*Volkening and Sandstede, 2015*; *Nakamasu et al., 2009*; *Binder and Simpson, 2013*). New biological observations demonstrate that S-iridophores play a fundamental role in body stripe formation (*Singh and Nüsslein-Volhard, 2015*; *Frohnhöfer et al., 2013*; *Patterson and Parichy, 2013*). In particular, it has been shown that without S-iridophores, spots of melanocyte aggregates form instead of stripes, which is contrary to what these Turing reaction-diffusion models predict. These findings have paved the way for more detailed modelling, such as that of *Volkening and Sandstede, 2018*, who demonstrated (using an off-lattice individual-based model) the need for understanding S-iridophore behaviour when representing all three cell-types. For these reasons, we consider an inclusive modelling approach, incorporating the crucial cell-type S-iridophores and the full range of interactions depicted above.

Here, in a bottom-up approach, we hypothesise that the current biological understanding is sufficient to explain the major aspects of pigment pattern development and construct a model to test this. In particular, we construct an agent-based model incorporating all three pigment cell-types and their documented cellular interactions. We use observations of a set of three mutants that each lack an individual cell-type, plus the three double mutant combinations lacking pairs of cell-types, to deduce the key rules likely underpinning S-iridophore dynamics. Combining these assumptions with experimentally verified biological mechanisms in the literature, we generate a working model of adult pattern formation. We then run simulations for wild type (WT) and these mutant fish. We show that in each case our model correctly predicts the patterns observed in vivo, and that pattern development displays multiple quantitative matches to that in vivo using a parameter sampling methodology to demonstrate the robustness of these patterns to parameter variation. In an independent test of the model, we simulate mutants with pigment pattern defects caused by changes other than to the presence of pigment cell-types, and show that these too are successfully matched in silico by our model.

Our work demonstrates that current biological understanding, alongside simple assumptions about S-iridophore behaviour, is sufficient to explain adult pigment pattern formation in WT and multiple mutants. Our work reinforces the growing realisation in the field that the previously neglected S-iridophores are crucial for stripe formation, suggests a minimum set of their rules, and reveals unexpected subtleties to the phenotypic impact of the well-studied *leo* mutant.

## Materials and methods

### Modelling overview

We build our model with direct reference to the known biology. We model five cell types as individual agents: melanocytes ($M$), xanthophores ($X$), xanthoblasts ($X^b$), the unpigmented precursor cell to xanthophores) and S-iridophores in either dense or loose form ($I^d$, $I^l$, respectively). These are the cells we deem from the literature to be crucial for successful pattern formation. We do not directly model L-iridophores, since these appear after the adult pattern is formed and are more likely involved in pattern maintenance (*Frohnhöfer et al., 2013*). Unlike previous models of stripe formation (*Nakamasu et al., 2009*; *Bullara and De Decker, 2015*; *Volkening and Sandstede, 2015*; *Painter et al., 2015*; *Bloomfield et al., 2011*; *Volkening and Sandstede, 2018*), we include xanthoblasts as an independent cell-type in our model. This is because the larval xanthoblasts appear principally by dedifferentiation of the embryonic xanthophores, and most metamorphic xanthophores arise from the larval xanthoblasts (*Mahalwar et al., 2014*; *McMenamin et al., 2014*; *Budi et al., 2011*; *Singh et al., 2014*; *Dooley et al., 2013*), whilst xanthoblasts that do not re-differentiate into xanthophores persist in the stripe regions where they play a role in consolidating melanocytes into stripes.

Zebrafish pattern formation generates distinct pigment cell layers in the hypodermis (*Figure 1B*) a melanocyte, xanthophore and S-iridophore layer (*Hirata et al., 2003*; *Hirata et al., 2005*). For

consistency, we model each of the three layers as independent, two-dimensional lattice domains throughout pattern formation (*Figure 2A*).

Agents representing $X$ and $X^b$, $M$, $I^d$ and $I^l$ occupy lattice sites, within xanthophore, melanocyte and S-iridophore domains respectively (*Figure 2A*). To account for the different packing densities of the cell types, lattice sites within the xanthophore and S-iridophore model layers are half the width and length of melanocyte sites size. This packing density does not have an impact on pattern formation, but, is included for biological realism. (For more details, see Appendix 1). Within each layer, volume exclusion properties hold: no two agents can occupy the same site at any one time (i.e. cells do not overlap).

The system is initialised to represent a typical WT fish shortly after the start of adult pigment pattern development ($\approx 25$ dpf). We set the domain height to be 1 mm, since this is the approximate height of the fish at 25 dpf (*Supplementary file 3* for details), we set the domain length to be 2 mm, representing approximately one-third of the full length, from the tip of the snout to the start of the tail, at 25 dpf, and thus equivalent to the trunk (*Parichy et al., 2009*). We populate the domain itself at $t = 0$ as an approximation of the observed larval pattern at 25 dpf (*Frohnhöfer et al., 2013*). At this time, there is a central stripe of dense S-iridophores along the horizontal myoseptum, scattered melanocytes and de-differentiated xanthophores (xanthoblasts) scattered across the domain.

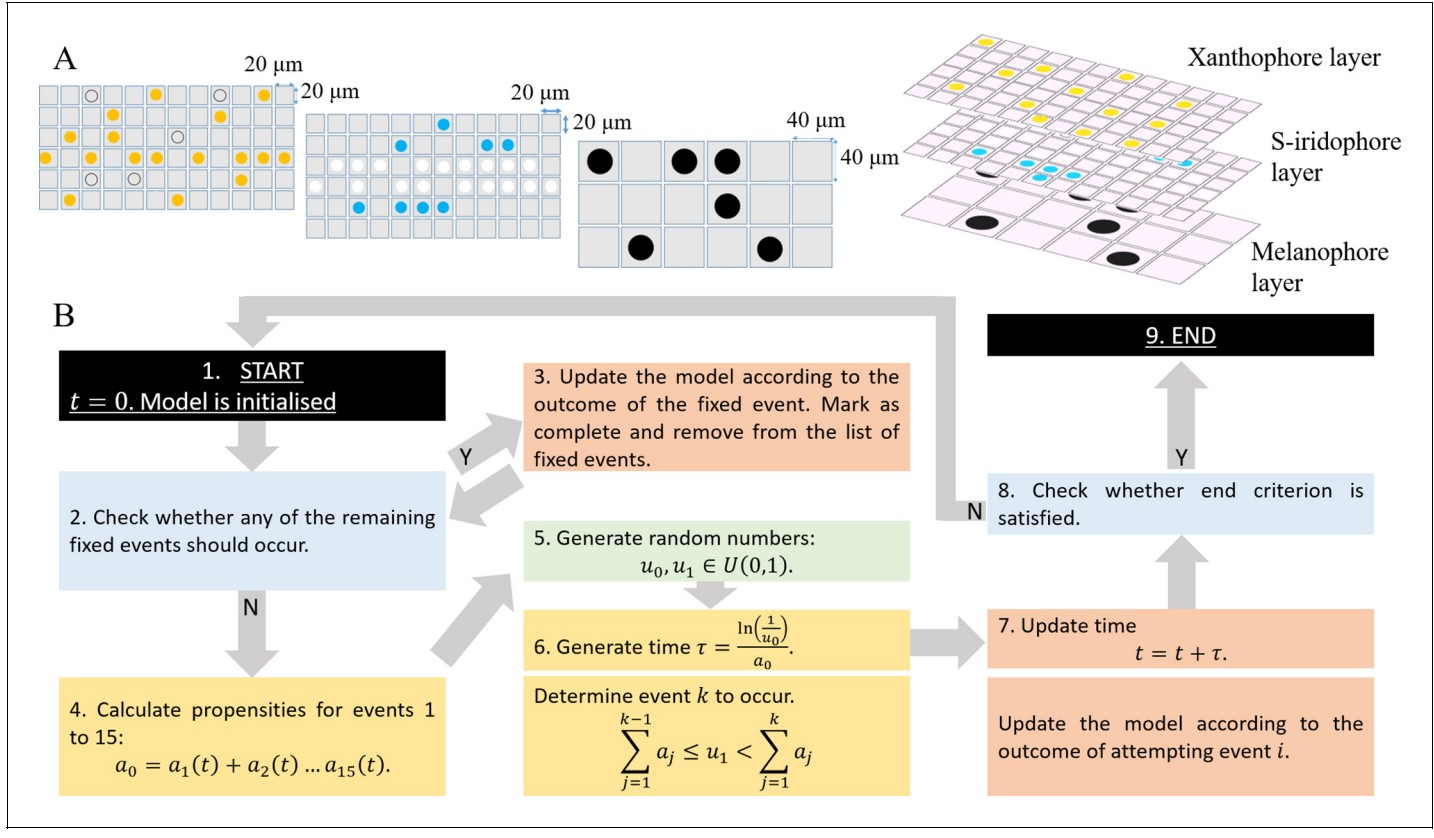

**Figure 2.** Model setup and simulation. (**A**) An example model setup. The domain is made up of three layers. Layer $X$ which contains yellow $X$ (yellow circles) and unpigmented $X^b$ (clear circles with black outline). Layer $I$ which contains silvery $I^d$ (white circles) and blue $I^l$ (blue circles). Layer $M$ which contains black $M$ (black circles) only. Each lattice site on each of the respective layers contain at most one cell at any given time. The layers are stacked on top of each other as seen in real fish. We note that in our simulations the ordering of the layers does not play any role in determining pattern formation. (**B**) Schematics of model implementation. (1) The model is initialised as described in Appendix 1. (2) The model is checked for the requirements for a fixed event to occur (e.g. new cells differentiating at a given time). If there is a fixed event to be implemented, the algorithm moves to stage 3. Otherwise, the algorithm passes to stage 4. (3) The model is updated according to the fixed event and algorithm returns to stage 2. (4) The propensity functions - probabilities for all other events to occur $\alpha_1(t), ...\alpha_{15}(t)$ are calculated. (5) Random numbers $u_0, u_1 \in U(0,1)$ are generated. (6) Numbers $u_0, u_1$ are used to determine the time $\tau$ until the next event and which event will be implemented. (7) The model configuration is updated. (8) The algorithm checks if the end criterion is satisfied, that is in the case of WT, that $\Omega_{SL} = 13.5mm$. If so, the algorithm finishes. Otherwise the algorithm continues, returning to stage 2. (9) The simulation completes.

We model this by populating the central three rows of the S-iridophore layer with dense S-iridophores, and by distributing melanocytes uniformly at random into sites within the melanocyte domain at density 0.04 and xanthoblasts uniformly at random into sites in the xanthophore domain at density 0.4.

The model is then updated according to the Gillespie algorithm (*Gillespie, 1977*). An overview of how the model is updated is given in *Figure 2B* and can be described as follows. At any given time $t$, the model is first assessed for meeting the criteria of a fixed event. Fixed events are all biologically determined events that occur once at a fixed time. For example at the start of pattern formation, the appearance of dense S-iridophores along the horizontal myoseptum is a fixed event. If the model meets the criteria, the fixed event occurs, is subsequently marked as complete and the simulation continues. If no fixed time event is to be implemented then one of 15 possible continuous time events is attempted. To do this, we treat all the potential actions, (for example cell birth or domain growth [as described in Section "Modelling assumptions"]), as individual 'events', each with an exponentially distributed waiting time which corresponds to their rate of occurrence (as specified in the literature *Supplementary file 4*). To update the model at any given time $t = T$, an exponentially distributed waiting time; $\tau$ is generated until the next possible 'event' occurs (based on the rates of all of the possible events). Next, a random number $u_1 \in U(0, 1)$ determines which event occurs based on the relative probability of each event occurring. Once an event is chosen, the domain is updated accordingly: if conditions required for that event to occur are met, the event is implemented, whereas if they are not then there is no change. Time is also now updated to $t = T + \tau$. This process repeats until we reach the end of pattern metamorphosis, defined by the simulated field standard length reaching approximately 13.5 mm (*Supplementary file 3*). The stochastic nature of our algorithm means that in any given simulation, the final pattern and its individual development will be inherently different to any other simulation, just as in real fish. Events incorporated into our model include all processes involved in the self-organisation of pigment cells during pattern metamorphosis as well as uniform domain growth with rate 0.13 mm per day in horizontal axis and 0.033 mm per day in the vertical axis (*Parichy et al., 2009*). These events are described in more detail in Section "Modelling assumptions".

Cells interact in the fish skin at both short (neighbouring cells) and long (up to half a stripe width $\approx 0.25$ mm) range, with interactions thought to use direct contact through cellular extensions (filopodia, dendrites, or longer airenemes). In our model, uniform disks, with radii on the order of the distance between 2 cells ($\approx 0.04$ mm) account for short-range interactions (*Figure 3A–D*), and an annulus with an outer radius of 0.24 mm (12 cells) and inner radius of 0.22 mm, (11 cells) represent long-range dynamics (*Figure 3E–H*). We allow cell interactions across different layers (as in real pattern formation). Cells that are chosen for movement can move into one of eight sites local to them. The probability of movement in one of the eight direction is biased according to how attracted or repelled the focal cell is to its local neighbours (*Figure 3I–J*). For more detail about how short- and long-range interactions are implemented see Appendix 1. See *Supplementary files 4*, *5* and *6* for a detailed justification of the rates, interaction types and parameter values, respectively.

## Modelling assumptions

In this section, we describe our modelling assumptions with regard to cell–cell interactions. These assumptions include all the known interactions between melanocytes, dense S-iridophores, loose S-iridophores, xanthophores and xanthoblasts, as well as some predictions about S-iridophore behaviour which have not been experimentally investigated in the literature. Apart from those involving S-iridophores, all the interactions and wherever possible their quantitative properties (strength, frequency etc) come directly from the literature, and are summarised in *Figure 4G*, 1-14, and described in *Supplementary file 5*. These include interactions influencing the movement, proliferation, differentiation and death of all cell types. These are represented explicitly in the model in as biologically realistic a manner as possible, at their determined rates.

The interactions involving S-iridophores have not been well-characterised experimentally, so we have developed our own predictions based on the literature describing S-iridophore behaviour during pattern metamorphosis. It has been shown using clonal cell analysis that during pattern metamorphosis S-iridophores spread across the skin of the zebrafish bidirectionally by proliferation of existing cells (between once and twice per day) combined with quick migration (*Mahalwar et al., 2014*). We further predict that dense S-iridophores show a directional bias towards xanthophores in

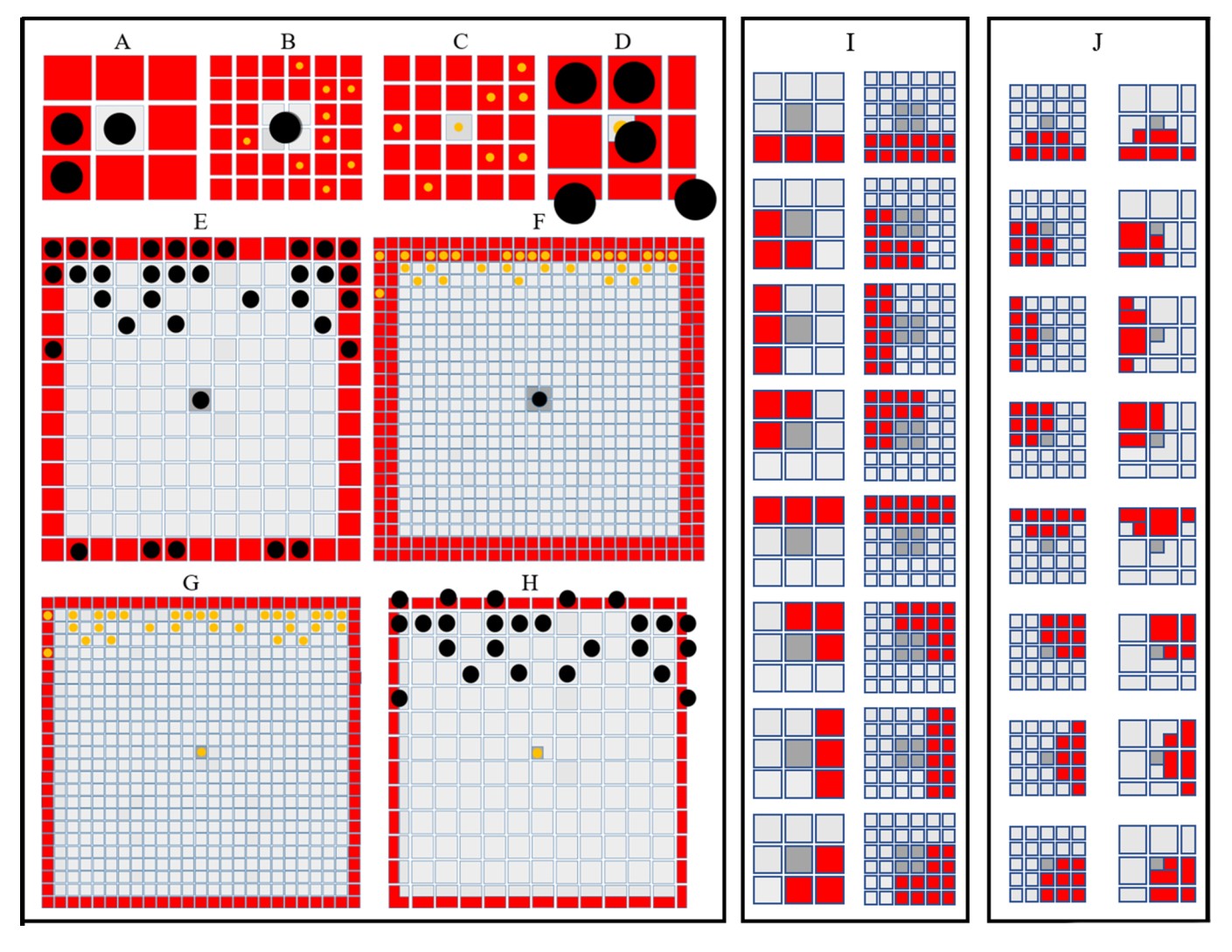

**Figure 3.** Simulating short- and long-range interactions. (A–F) Comparing the number of cells in the short (A–D) and long (E–F) range distance (0.04 mm) from a central site, on different domain types. $M$ are represented as black circles. $X$ are represented as yellow circles. (A–B) A visualisation of sites (marked in red) local to central melanocyte on the (A) melanocyte domain and (B) xanthophore domain.(C–D) A visualisation of sites (marked in red) local to central xanthophore on the (C) xanthophore domain and (D) melanocyte domain. (E–F) A visualisation of sites (marked in red) long-distance to central melanocyte on the (E) melanocyte domain and (F) xanthophore domain.(G–H) A visualisation of sites (marked in red) long-distance to central xanthophore on the (G) xanthophore domain and (H) melanocyte domain. (I) Melanophore movement with respect to local neighbours. If a melanophore located in a central position (marked in dark grey) attempts to move, the melanophore will consider neighbours in all sites marked in red on the melanophore domain (left), xanthophore and S-iridophore domain (right). (J) Movement on xanthophore/iridophore domain with respect to local neighbours. If a xanthophore/iridophore located in a central position (marked in dark grey) attempts to move, the xanthophore/iridophore will consider neighbours in all sites marked in red on the xanthophore and S-iridophore domain (left), melanophore domain (right).

the short range. We propose that dense S-iridophores are attracted in the short range to xanthophores since they are highly associated with each other in each of the mutants and in WT (*Frohnhöfer et al., 2013*), and this mutual attraction may be important for interstripe consolidation. Furthermore, loose S-iridophores show a directional bias away from other loose S-iridophores in the short range. We propose that loose S-iridophores are repelled by other loose S-iridophores as this would facilitate the prompt spreading of loose S-iridophores across the stripe regions.

Interestingly, as the S-iridophores spread they switch between a loose and dense form, predetermining the positioning of stripes and interstripes consecutively. In the loose form S-iridophores are spread and stellate in appearance. In contrast, in the dense form, S-iridophores are compact. The

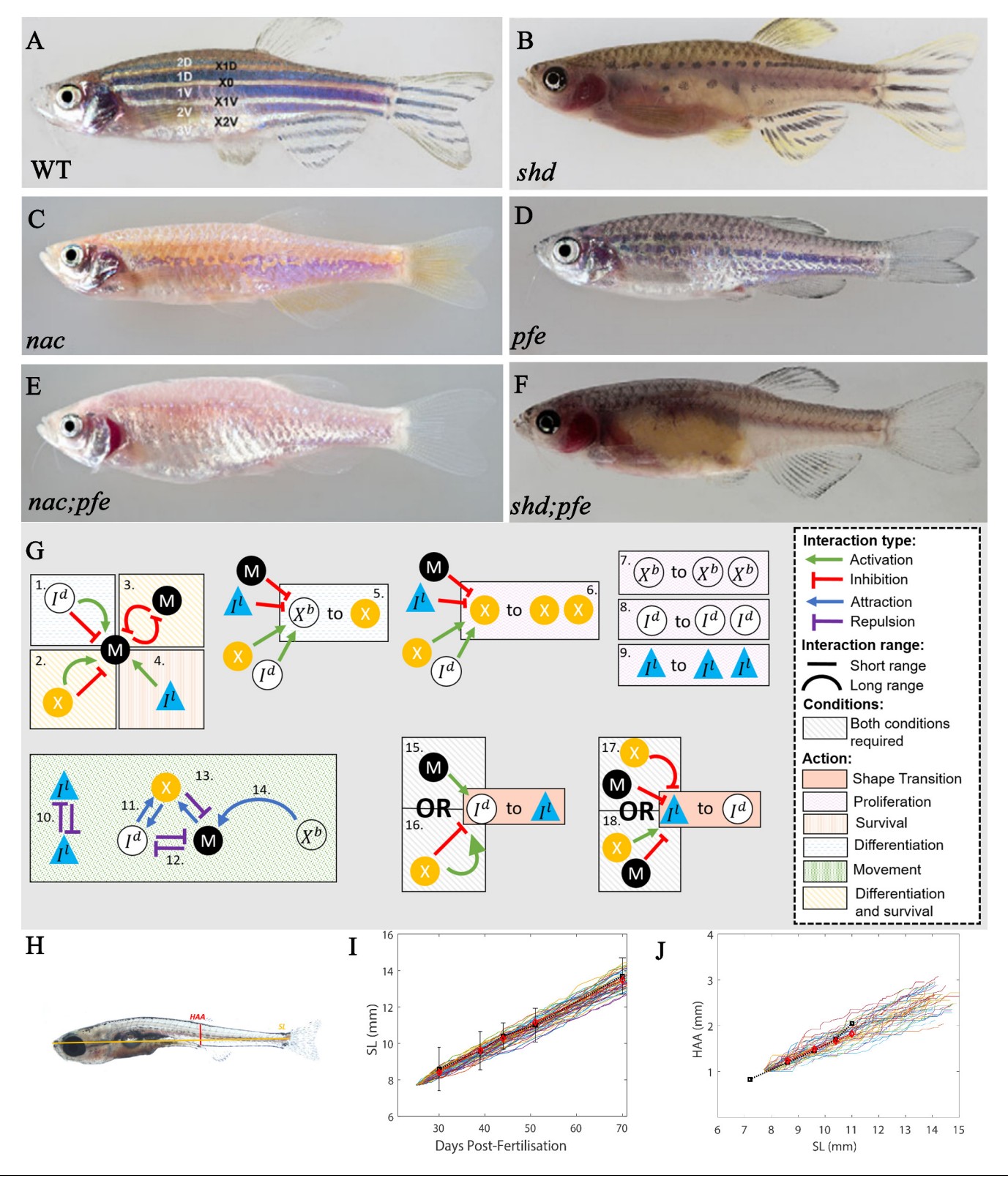

**Figure 4.** 'Missing cell' mutants and cell-cell interactions implemented. (A–F) Adult phenotype of the set of fish used to deduce S-iridophore interactions. (A) WT fish, (B) *shady* (*shd*), (C) *nacre*(*nac*), (D) *pfeffer* (*pfe*), (E) *nac;pfe*, (F) *shd;pfe*. (G) A representation of all the interactions implemented in the model. See *Supplementary files 4* and *5* for a detailed justification of the rates and interaction types, respectively. See *Supplementary file 6* for a detailed justification of all parameters. The direction of the arrow/inhibition sign indicates which cell is acting on which. For example in assumption

*Figure 4 continued on next page*

*Figure 4 continued*

13, the blue arrow from melanophore to xanthophore indicates that melanophores attract (green background) xanthophores. The purple inhibition arrow indicates that the xanthophores repel melanophores. OR statements in assumptions 15–18 are logical OR statements. For example, conditons 15 and 16 indicate that a dense S-iridophore may become loose if there are melanophores in the short range OR, there are no xanthophores in the short range AND many xanthophores in the long range. (H) A visualisation of measurements HAA and SL. (I) Plots of SL versus days post-fertilisation (dpf) for 40 model realisations. Each coloured line is a single realisation. Black squares are SL versus days post-fertilisation extracted from *Parichy et al., 2009* in real fish given in *Supplementary file 3*. Red diamonds are the mean SL versus days post-fertilisation from 40 simulations. (J) Plots of SL (mm) versus HAA (mm) for 40 model realisations. Each coloured line is a single realisation. Black squares are SL versus HAA extracted from *Parichy et al., 2009* given in *Supplementary file 3*. Red diamonds are the mean HAA versus SL from 40 simulations. Error bars are one standard deviation. The model agrees well with the data in both cases. Images (A–F) reproduced from *Frohnhöfer et al., 2013* and licensed under CC-BY 4.0 (https://creativecommons.org/licenses/by/4.0).

transition between the two types appears to be interchangeable. When dense S-iridophores initially spread beyond the boundary of the first X0 interstripe, they can later change to loose type (*Fadeev et al., 2015*). Similarly when loose S-iridophores reach an interstripe region, they can aggregate and form dense S-iridophores. It is not clear exactly what causes these shape transitions physically, and this is not a question we address here. It has, however, been shown that loss of Tjp1a function in *sbr* mutants compromises the transition of S-iridophores from dense to loose state, suggesting that Tjp1a contributes to regulation of the molecular switch that regulates S-iridophore shape changes during their dispersal (*Fadeev et al., 2015*). We envisage iridoblasts as initially differentiating in a dense form along the horizontal myoseptum, proliferate, migrate and spread, later dedifferentiating and then re-differentiating into the opposite form dependent on their location with respective to other cell types (melanocytes and xanthophores).

Here, we hypothesise how the cell types affect S-iridophore type (loose or dense). The cause of these transitions is largely unknown, however, it has been suggested to be dependent on signals from melanocytes and xanthophores transmitted by gap junctions (*Irion et al., 2014*; *Fadeev et al., 2015*). In order to investigate this, we consider a primary set of six mutants known to prevent the formation of one or more individual pigment cell-type. We use these to define the contribution and nature of S-iridophore interactions in pattern formation, by considering the outcomes in fish lacking each of the three cell types. Specifically we consider:

- Mutants lacking S-iridophores: The gene *shady* (*shd*) encodes zebrafish leukocyte tyrosine kinase (Ltk) which plays a role in S-iridophore specification (*Lopes et al., 2008*). As a result, strong *shd* mutants lack all S-iridophore types. The resultant adult pattern consists of a widened X0 region of xanthophores, which are flanked dorsally and ventrally by melanocytes organised as spot-like clusters in a sea of xanthophores, forming broken stripes (*Figure 4B*).
- Mutants lacking melanocytes: The gene *nacre* (*nac*) encodes the transcription factor Mitfa (*Lister et al., 1999*). *nac* mutants lack melanocytes throughout embryonic and larval development (*Lister et al., 1999*). As a result, stripes do not form properly and the adult phenotype consists of a prominent X0 interstripe of dense S-iridophores and xanthophores with irregular borders, accompanied by spots of dense S-iridophores and xanthophores ventrally. The rest of the flank is filled with loose form S-iridophores (*Figure 4C*).
- Mutants lacking the xanthophore lineage: Gene *pfeffer* (*pfe*) (alternatively known as *salz* (*sal*)) encodes colony stimulating factor one receptor (csf1ra) that is expressed and required specifically in xanthophores (*Frohnhöfer et al., 2013*; *Patterson and Parichy, 2013*; *Parichy et al., 2000b*). In the adult fish of strong alleles, xanthophores are almost absent in embryos, and absent in adults. The resultant adult pattern consists of a spotted melanocyte stripe pigmentation of normal alignment which fades out into a 'salt and pepper'-like pattern more posteriorly (i.e., in the tail) (*Figure 4D*). Melanocyte spots are associated with loose S-iridophores. In the regions lacking melanocyte aggregation (the 'salt-and-pepper' region), S-iridophores take a dense form, with melanocytes scattered at very low density, an arrangement never seen in WT patterns.
- Double mutants of the aforementioned mutant types: *nac;pfe*, *nac;shd* and *shd;pfe* (*Figure 4E,F* depict the adult phenotypes of *nac;pfe* and *shd;pfe* respectively, there is no image available for *shd;pfe*). These mutants lack two of the aforementioned cell types. The resultant adult pattern is a uniform distribution of the remaining cell type (*Frohnhöfer et al., 2013*). These mutant phenotypes demonstrate that zebrafish stripe formation is not determined by an underlying pre-pattern, but instead is generated by cell-cell interaction.

Upon evaluating these mutants, we make the following deductions about S-iridophore shape transitions during pattern formation:

- S-iridophores are initially dense and cannot change shape autonomously. This is based on observations of mutants *nac;pfe* which only contain S-iridophores and in which the adult phenotype consists of dense S-iridophores in a coherent sheet across the domain (*Frohnhöfer et al., 2013*). In contrast, *pfe* and *nac* both exhibit loose and dense S-iridophores (*Frohnhöfer et al., 2013*), suggesting that both melanocytes and xanthophores are capable of facilitating S-iridophore shape transitions.
- Melanocytes in the short range promote the transition of dense to loose, conversely, a lack of melanocytes in the short range promotes the transition of S-iridophores from loose to dense. We propose these interactions for the following reasons. Firstly, in *pfe* and WT, dense S-iridophores are associated with lack of melanocytes, for example within the interstripes, whilst loose S-iridophores are associated with melanocytes, for example in the stripe region. Since we predict that melanocytes are required for dense S-iridophores transition the simplest assumption is that melanocytes promotes dense to loose transitions in the short range. Since loose S-iridophores can re-aggregate to dense form in *pfe* we assume that this signal is bidirectional and therefore a lack of melanocytes promotes the loose to dense transition.
- Xanthophores in the long range and lack of xanthophores in the short range promotes the transition from dense to loose; conversely, a lack of xanthophores in the long range as well as many xanthophores in the short range, promotes the transition from loose to dense. We propose these interactions for the following reasons. In *nac* and WT, dense S-iridophores are associated with xanthophores, whilst loose S-iridophores are associated with a lack of xanthophores (*Frohnhöfer et al., 2013*). Since S-iridophores initially appear in dense form and become loose for example in *nac*, when there are xanthophores in a low density local to S-iridophores and high density in the far range, we predict it is this combination that promotes the transition of dense to loose in the long range. Since in *nac*, S-iridophores can transition back from loose to dense when the local xanthophore density is high and far xanthophore density is low, we assume the opposing interaction is also true (*Frohnhöfer et al., 2013*).

These descriptions are summarised in *Figure 4G*, 15-18. We note that in each of these cases variations of these interactions were already hypothesised by *Frohnhöfer et al., 2013*. However, since their predictions did not distinguish loose and dense S-iridophores and did not indicate transition mechanisms, their predictions though similar, are extended here to incorporate these differences. The predictions we describe are the simplest possible for generating the patterns observed in the aforementioned set of fish, upon removing any one of these interactions, the model fails to generate the robust patterns we will describe (Figure 11 and Section "Necessity of S-iridophore assumptions").

## Comparing simulated fish with real data

In order to validate our model, we compare different aspects of our simulation (size, spatial distributions of cells, numbers of melanocytes, stripe and interstripe width) with real fish at different developmental stages. In real fish, developmental stages are categorised according to the standard length (SL) of the fish (*Figure 4H*; *Supplementary file 3*; *Parichy et al., 2009*). For consistency, we calculate the 'stage' of our simulations using the length of our domain and a simple calculation to generate a simulated SL (see Appendix 1). This allows us to make a direct comparison between the range of sizes obtained in model simulations and the natural range in zebrafish HAA and SL. As a test of validity of this measure, *Figure 4I* and *Figure 4J* demonstrate 40 plots of simulated SL versus days post-fertilisation (dpf) and simulated HAA versus SL, respectively, compared with the averaged data (*Parichy et al., 2009*). These figures demonstrate that whilst growth rates are variable within simulations (as seen in real fish), the mean of our simulated rates approximately matches that in real fish.

## Results

### Modelling simulations

Having deduced this minimal rule-set from the literature and our further predictions from the phenotypes of the selected primary set of mutants, indicated in Section "Modelling overview" we use these as the basis for our model. We use our model to generate stochastic simulations of pigment pattern formation corresponding to the the period of adult metamorphic pigment pattern formation, during which the SL extends from 7.6 mm to 13.5 mm. We note that adult pattern metamorphosis and the appearance of metamorphic S-iridophores starts earlier, around 6–7 mm SL (*Parichy et al., 2009*). We initialise later at 7.6 mm as by this time the skin lying over the horizontal myoseptum is populated with an initial stripe of dense S-iridophores. We intialise our model accordingly to match this. Subsequently, we first assess the ability of the model to reproduce natural growth at a quantitative level, and then to generate the WT pigment pattern, both qualitatively and quantitatively. We then go on to simulate conditions corresponding to our primary set of mutants, considering the qualitative fit to the published patterns. To test robustness of the patterns, we provide a rigorous robustness analysis by carrying out one hundred repeats of the WT simulation and 'missing cell' mutants with perturbed parameter values chosen uniformly at random from the range 0.75–1.25 of their described value and show that in each case, the appropriate pattern is preserved. Finally, in a more rigorous test of the predictive power of our model, we explore three further mutant phenotypes that had not been considered in deriving the model's rule-set.

### Simulation of WT pattern

In this section, we compare qualitatively our simulations of WT fish. For WT simulations, the model rules are given in Section "Modelling overview". *Figure 5A–D* depicts WT development, while *Figure 5A'–D'* (*Video 1*) shows a representative simulation using the model described by the rules in Section "Modelling overview". The simulations reproduce qualitatively most aspects of the biological pattern. The model is initialised at stage PB. At stage PR, we begin to see an accumulation of melanocytes either side of the initial interstripe and differentiation of new xanthophores. Furthermore, we see the development of 1D and 1V stripe regions and delamination of S-iridophores from dense to loose form at the edges of interstripe X0. At stage SP, we observe the spreading of loose S-iridophores across the two developing stripe regions. Finally at stage J+, we see three interstripes alternating with five dark stripes. The final pattern matches the stripes seen in the real WT fish and the cellular component of dark stripes ($X$, $I^l$, $M$) and light interstripes ($I^d$, $X$) matches the composition of pigment cells in real fish (*Figure 1C*). We emphasise that the simulations presented here (as well as in future sections) are a representative example of the model output.

### Robustness of the model

Due to the abundance of parameters and cell-cell interactions necessary to capture what is known biologically about zebrafish pigment pattern formation, it is not feasible to perform an exhaustive parameter sweep to demonstrate the robustness of the model. Instead, as a test of robustness, we perform a rigorous robustness analysis by carrying out one hundred repeats of the WT simulation with perturbed parameter values chosen uniformly at random from the range 0.75–1.25 of their described value. The precise value of each parameter is sampled uniformly from this region, independentally for each parameter and each repeat. Twenty of these randomly sampled repeats are given in *Figure 6*. We observe that for all one hundred repeats that small perturbations to the rates still generate consistent striping, demonstrating the robustness of the model.

### Simulation of 'missing' cell mutants

In the next four sections (3.1.2–3.1.2), we compare qualitatively our simulations of mutants lacking one or more cell types. For the case of generating these mutants, we simulate the same WT model except we remove the appropriate cell type from the initial conditions and turn off cell birth of that cell type to match the mutation. For example, in *shd* we remove S-iridophores from the initial conditions and switch off S-iridophore birth. No other changes are made. For more information about mutant implementation see Appendix 2. These mutants often display similar features to WT fish; however, some aspects of the stripe formation are incomplete. In order to describe these

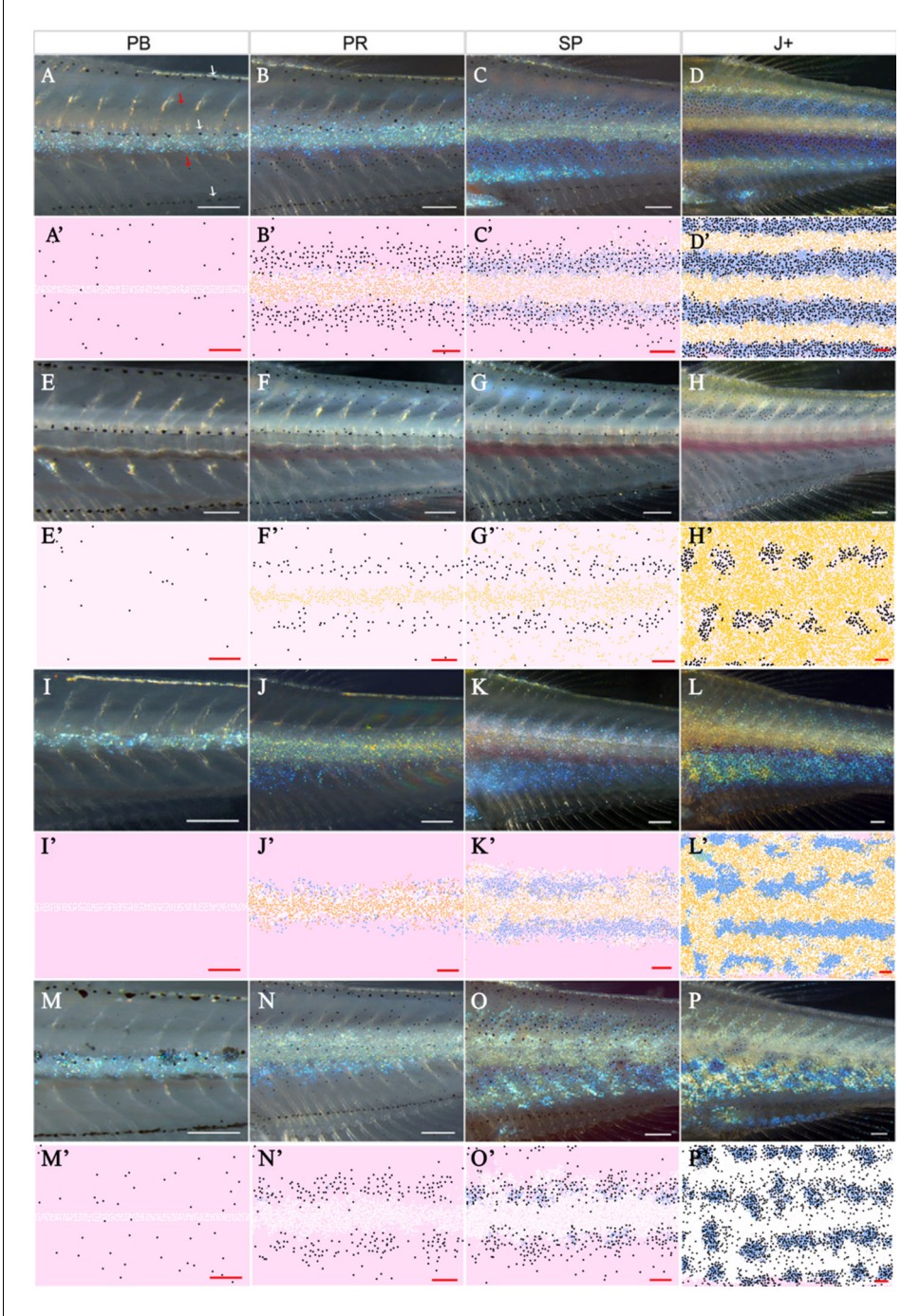

**Figure 5.** Representative simulation of WT, *shd*, *nac* and *pfe*. (A–D), (E–H), (I–L), (M–P) WT, *shd*, *nac*, *pfe* development respectively and (A'–D'), (E'–H'), (I'–L'), (M'–P') corresponding model simulation. Red arrows: melanocytes in the skin layer, modelled in (A). White arrows indicate the embryonic pattern of melanocytes in four stripes, that are deeper than the skin level and are consequently not included in the model. See *Videos 1–4* for representative examples of these simulations in movie format. Scale bar is 0.25 mm for all images. Experimental images (A–D), (E–H), (I–L), (M–P) reproduced from *Frohnhöfer et al., 2013* and licensed under CC-BY 4.0 (https://creativecommons.org/licenses/by/4.0).

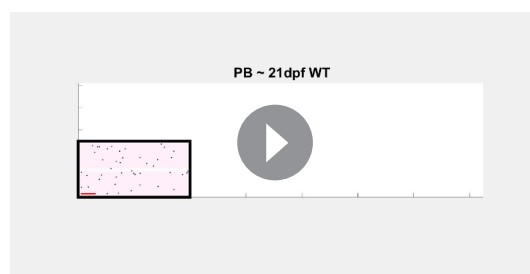

**Video 1.** Simulated development of WT.
https://elifesciences.org/articles/52998#video1

differences, with reference to *pfe*, *shd* and later in *sbr*, we define pseudo-stripes as the spots of melanophore aggregates that appear in a stripe-like orientation reminiscent of that in WT fish. We describe the pseudo-stripes in the order they appear as in WT fish. For example, we define pseudo-1D and pseudo-1V to be the first pseudo-stripes.

We demonstrate here the capability of our model simulations to reproduce qualitatively the pattern development of these mutants. For each mutant, we describe the initialisation of the model domain to match the fish at stage PB as well as all of the similarities observed between our model outputs and real fish at the following three developmental stages, PR, SP and J+. Finally, we describe the variation between many repeats of the model and how this correlates with real same-type siblings.

## The *shady* mutant

At stage PB (*Figure 5E,E'*), we populate the domain with some randomly dispersed melanocytes at a lower density than that in WT (*Frohnhöfer et al., 2013*) and some randomly dispersed xantho-blasts at the same density as WT (*Frohnhöfer et al., 2013*). At stage PR (*Figure 5F,F'*), we observe some melanocytes beginning to differentiate in the usual 1D and 1V stripe regions. At stage SP (*Figure 5G,G'*), we observe the accumulation of melanocytes around the 1D and 1V stripe regions with a central stripe of xanthophores. Finally, at stage J+ (*Figure 5H,H'*), we observe two horizontal pseudo-stripes of melanocyte spots surrounded by xanthophores. We found that in 100 simulations, 100% of *shd* stage J+ mutants observed two pseudo-stripes (1D and 1V) just as in *Figure 5H*. See *Video 2* for a movie of the simulation.

Moreover, pseudo-stripes varied in how stripe-like they were as observed in real fish (*Frohnhöfer et al., 2013*). As a measure of this, we calculated the longest stretch of melanophores in a row without any significant breaks over 100 simulations. This gives an indication of the widest 'spot' or 'pseudo-stripe' of melanophores in a simulation. We found that on the average, the mean of widest spot width over one hundred simulations was 0.18 of the simulated length. The widest spot width in 100 simulations was 0.43 of the simulated length, demonstrating the variance in pseudo-stripe length without break.

## The *nacre* mutant

At stage PB (*Figure 5I,I'*) we populate the domain such that there is an initial stripe of dense S-irido-phores and randomly dispersed xanthoblasts at the same density as in WT. At stage PR (see *Figure 5J,J'*) we see the appearance of newly differentiated xanthophores associated with the dense S-iridophores in the initial X0 interstripe. At stage SP (*Figure 5K,K'*) we observe the switch of dense S-iridophores to loose form and the subsequent spreading of loose S-iridophores. Finally at stage J+ (*Figure 5L,L'*) we observe the jagged edges of

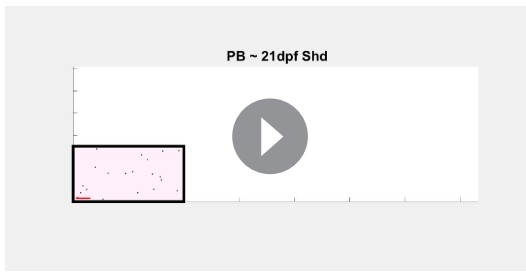

**Video 2.** Simulated development of *shd*.
https://elifesciences.org/articles/52998#video2

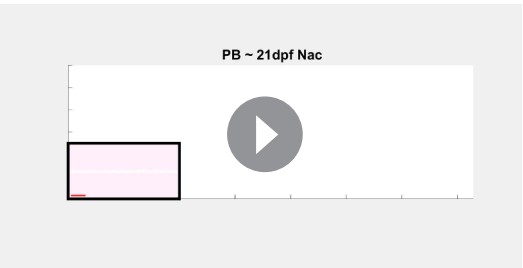

**Video 3.** Simulated development of *nac*.
https://elifesciences.org/articles/52998#video3

the usually straight X0 and the formation of a second pseudo interstripe some distance below X0 just as in real *nac* fish (*Figure 5L*). See *Video 3* for a movie of the simulation.

## The *pfeffer* mutant

At stage PB (*Figure 5M,M′*), we populate the domain with a central stripe of dense S-iridophores and randomly dispersed melanocytes at the same density as observed in WT (*Frohnhöfer et al., 2013*). At stage PR (*Figure 5N,N′*), we observe the arrival of melanocytes into the prospective 1D, 1V stripe regions that is less pronounced compared with

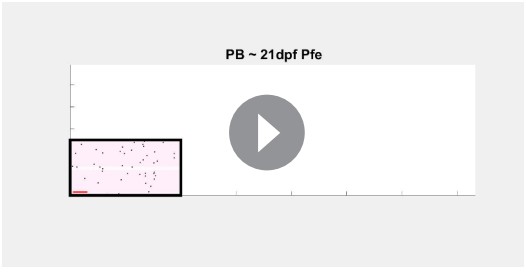

**Video 4.** Simulated development of *pfe*.
https://elifesciences.org/articles/52998#video4

WT simulations. At stage SP (*Figure 5O,O′*), we observe the accumulation of newly differentiated melanocytes into aggregates in prospective stripe regions 1D and 1V. Finally, at stage J+, (*Figure 5P,P′*) we observe the aggregation of melanocytes (associated with loose S-iridophores) into spots, surrounded by a sea of dense S-iridophores peppered with black melanocytes. In one hundred simulations, the median number of pseudo-stripes at stage J+ in these repeats was four, as WT. This is consistent with observations of *pfe* mutants, which typically show the same number of pseudo-stripes and -interstripes as WT fish (*Frohnhöfer et al., 2013*). We observe higher conservation of striping than in simulated *shd* mutants as observed in real fish. For example, in one hundred simulations, the average longest stretch of melanophores in any given simulation was 0.6 of the full length. See *Video 4* for a movie of the simulation.

As a test of robustness, we perform a rigorous robustness analysis by carrying out one hundred repeats of the mutant simulations with perturbed parameter values chosen uniformly at random from the range 0.75–1.25 of their described value as in Section "Simulation of WT pattern". Ten of these randomly sampled repeats are given in *Appendix 4—figure 1*. We observe for all one hundred repeats and in all three mutants, that small perturbations to the rate parameters still generate consistent patterning, demonstrating the robustness of the model.

## Double mutants; *shd;pfe, shd;nac, nac;pfe*

Lastly, we consider the double mutants. *Figure 7A* and *Figure 7B* depict adult patterns in *shd;pfe* and *nac;pfe* respectively. There is no image available for *shd;nac* adult or for the development of the aforementioned mutant phenotypes but it has been described in the literature that in all of the double mutants, the remaining cell type, by adulthood, fills the domain uniformly (*Frohnhöfer et al., 2013*). *Figure 7A′–C′* show a representative simulation for the mutants *shd;pfe, nac;pfe, shd;nac* respectively. In all cases of our model simulations, we observe that by stage J+ the remaining cell type begins to fill the domain. For example, in *nac;pfe* S-iridophores in dense form cover most of the flank by stage J+ (*Figure 7B′*).

## Simulation of other mutants

In Section "Simulation of WT pattern", we demonstrated that our proposed model reproduces the WT, single and double mutant patterns and thus is sufficient to explain pattern formation in the skin. In this section, we perform a more stringent test of the model's completeness, by asking whether it can successfully simulate the outcomes of a set of pigment pattern mutants which were not used to deduce the rules underpinning our model. Since we were particularly interested to test our predictions of the rules relating to S-iridophore interactions, our secondary set comprises mutants with S-iridophore-related phenotypes: *rose* (*rse*) homozygotes, which show a reduction of S-iridophore numbers; *schachbrett* (*sbr*) homozygotes, which show a delay in S-iridophore shape transitions from dense to loose and *choker* (*cho*) homozygotes, in which the absence of the horizontal myoseptum prevents the formation of the initial dense X0 band of dense S-iridophores (*Figure 7D–F*). In the next few sections (3.1.3–3.1.3), we demonstrate the capability of our model to reproduce quantitatively the patterns of these mutants. In each section, we first describe the nature of the mutation and the way in which we adapt our WT model to simulate the mutants. We describe the similarities of our model simulation with real fish at the different developmental stages considered.

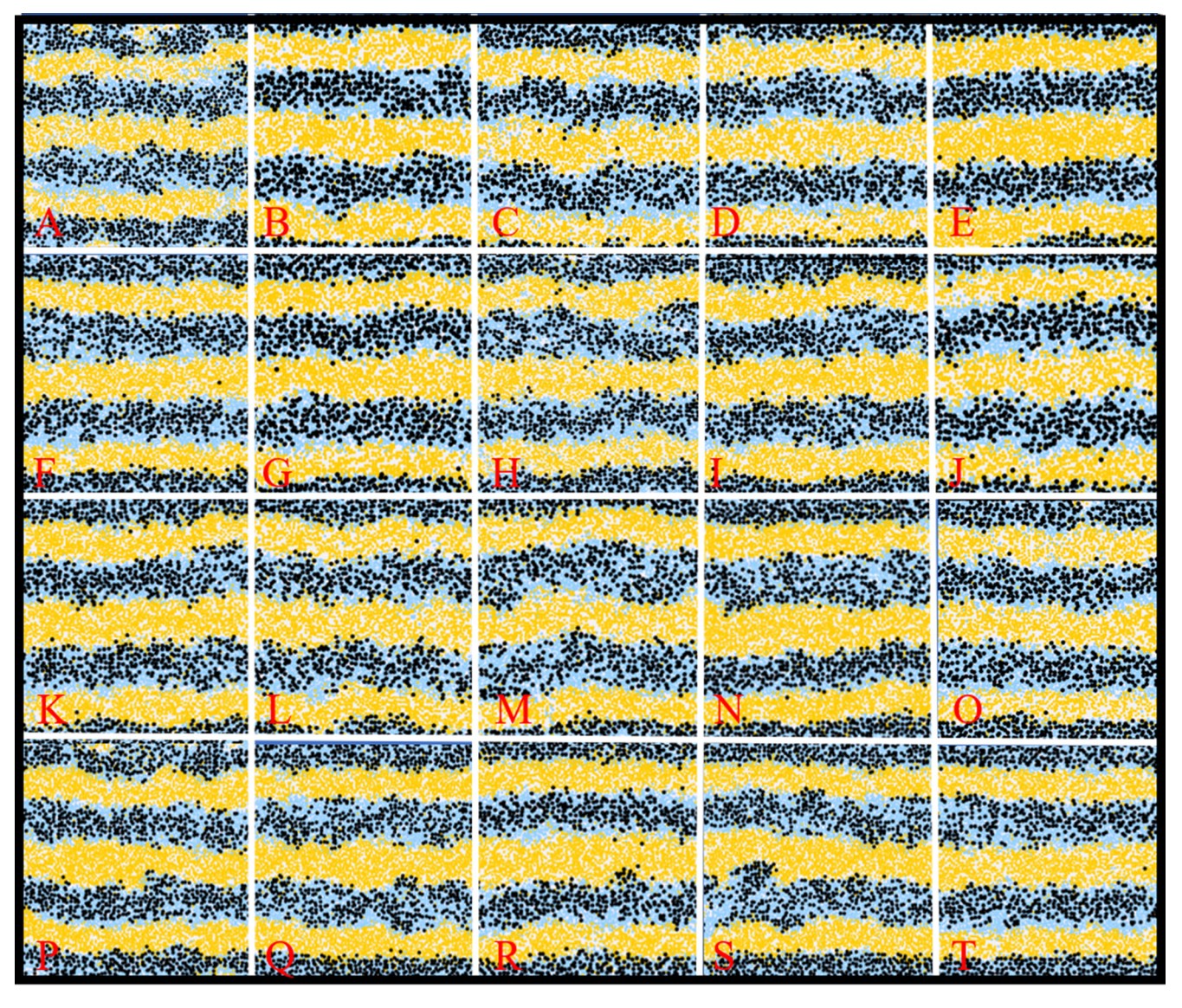

**Figure 6.** Example WT simulations at stage J+ when the parameters governing the rate of proliferation, movement, differentiation and death are perturbed slightly. Each square is an example WT simulation at stage J+ where each rate parameter is perturbed to $1+x$ times its normal value. The value $x$ is chosen uniformly at random from the set $[-0.25, 0.25]$.

## The *rse* mutant

*Rose* (*rse*), encodes the Endothelin receptor B1a (*Krauss et al., 2014*) and has been shown to acts cell-autonomously in S-iridophores; homozygous mutants result in a reduction of S-iridophores to approximately 20% of that seen in WT (observed in stage PB and adult fish [*Frohnhöfer et al., 2013*]). Consequently, adult fish show two broken dark stripes (reduction from four) bordering a widened X0 interstripe region. (*Figure 7D*). To simulate the *rse* mutant, we changed the number of initial dense S-iridophores at stage PB to one fifth of its usual number as observed in real fish at stage PB (*Figure 7G,G'*; *Frohnhöfer et al., 2013*). At stage PR (*Figure 7H,H'*), there is a strong reduction in melanocyte number compared to WT (*Figure 7I,I'*) and we observe that dense S-iridophores spread less far from the horizontal myoseptum. At stage SP, (*Figure 7J,J'*) the stripe boundaries at X0 are poorly defined, and dense S-iridophores are still largely associated with the X0 region, with only a few loose S-iridophores appearing at the dorsal and ventral margins. At stage J+ (*Figure 7K,*

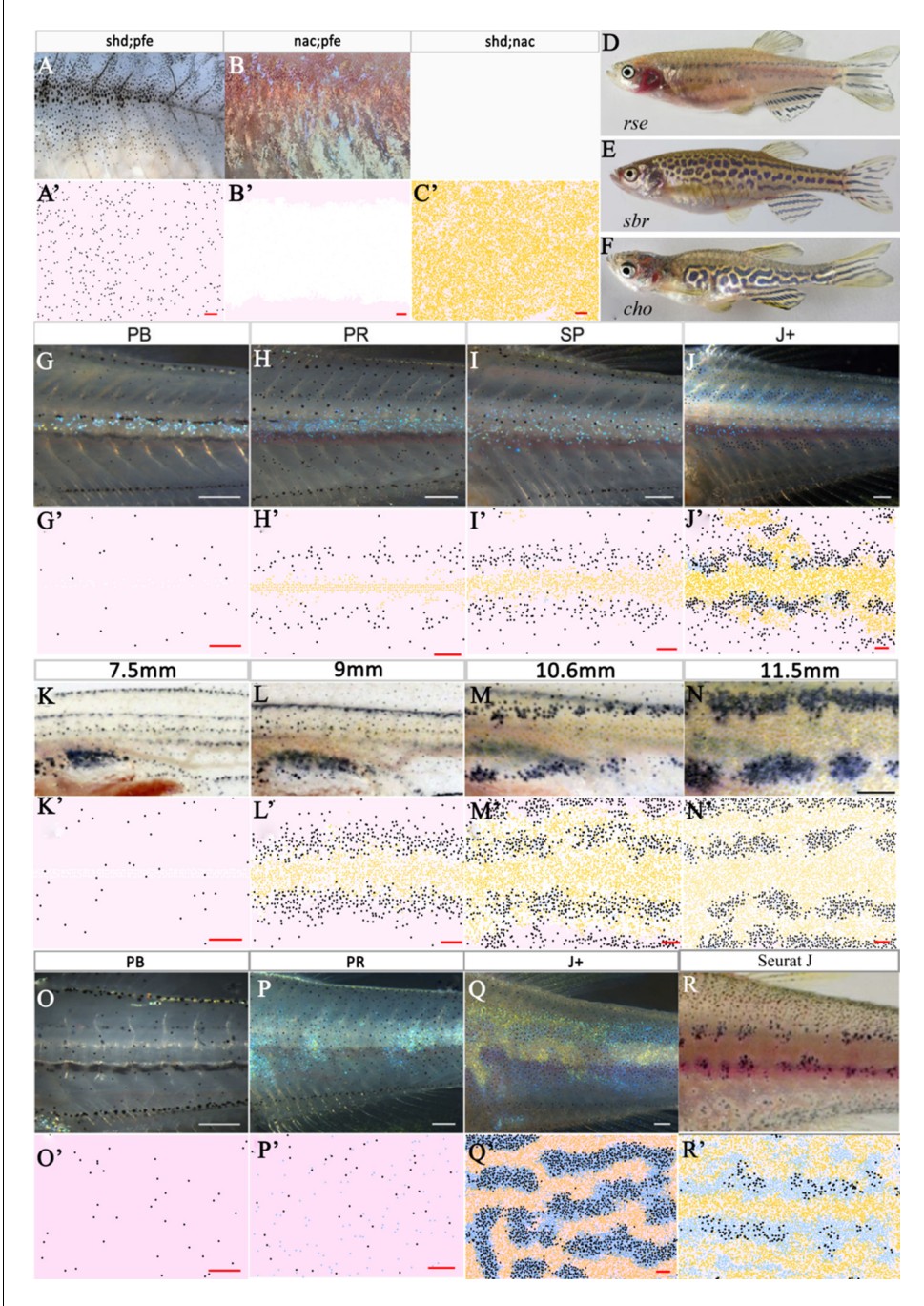

**Figure 7.** Other mutant fish simulations. (A–C) Real stage J+ *shd;pfe*, *nac;pfe*, *shd;nac* mutants, respectively. (A'–C') Model simulations at stage J+ for *shd;pfe*, *nac;pfe*, *shd;nac* mutants, respectively. (D–F) Adult phenotype of selected (D) *rse*, (E) *sbr*, (F) *cho* mutant. (G–J), (K–N), (O–Q) *rse*, *sbr* and *cho* development, respectively. (G'–J'), (K'–N'), (O'–Q') *rse*, *sbr* and *cho* model simulations, respectively. (R) *seurat* at stage J. (R') seurat model simulation at stage J. Scale bar is 0.25 mm for all images. See *Videos 5–7* for representative examples of these simulation in movie format. Experimental images (A–C), (D), (F) (G–J) and (O–Q) are reproduced from *Frohnhöfer et al., 2013*, (E), (K–N) from *Fadeev et al., 2015*, (R) from *Eom et al., 2012* and are all licensed under CC-BY 4.0 (https://creativecommons.org/licenses/by/4.0).

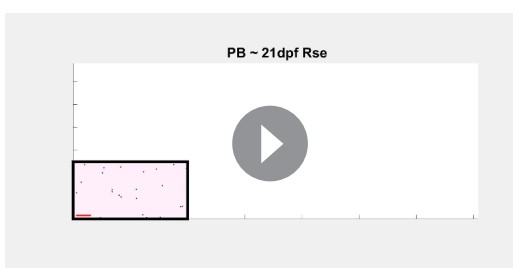

**Video 5.** Simulated development of *rse*.
https://elifesciences.org/articles/52998#video5

*K'*), the stripe boundaries at X0 are more distinct, but the dark stripes are thinner and partly fragmented. See *Video 5* for a movie of the simulation.

## The *sbr* mutant

The *sbr* gene encodes Tight Junction Protein 1a (Tjp-1a), which is expressed cell autonomously in dense S-iridophores (but not in loose S-iridophores) and truncated in *sbr* mutants; in adult *sbr* mutants S-iridophore shape transition from dense to loose is delayed (*Fadeev et al., 2015*). As a result, adult fish exhibit interrupted dark stripes, generating a pattern reminiscent of a checkerboard (*Figure 7E*). *Figure 7K–N* depicts *sbr* early pattern development. During adult pigment pattern formation, differences from normal WT development are not seen until ≈10 mm SL (*Fadeev et al., 2015*), (SP stage) at which point instead of dense S-iridophores transitioning to loose S-iridophores at the edge of the interstripe, in *sbr*, dense S-iridophores remain dense and spread over melanocytes dorsally and ventrally bidirectionally . At later stages, some dense S-iridophores do switch to loose S-iridophores. See *Video 6* for a movie of the simulation.

We interpret the *sbr* mutation as causing a delay in signaling driving the transition of S-iridophores from dense to loose S-iridophore. We model this by reducing the attempted rate of transitioning from dense to loose to a rate 40 × less than the rate of attempting loose to dense S-iridophore transition. Due to available data, we initialise the model for *sbr* at 7.5 mm SL to match that published regarding the real fish (*Figure 7K,K'*). At 9 mm (*Figure 7L,L'*) melanocytes begin to accumulate either side of the widened initial X0 interstripe. At 10.6 mm SL, we observe melanocytes that are associated with dense S-iridophores (white cells) and not just with loose S-iridophores (blue cells) as usually seen in WT at 10.6 mm ≈ stage SA (between stages SP and J+, *Figure 5C–D*). At 11.5 mm (*Figure 7M,M'*), melanocytes are organised into aggregates, approximately one stripe width in size, and only partially connected, thus forming a broken pseudo-stripe pattern.

## The *cho* mutant

Homozygous *cho* mutant larvae lack the horizontal myoseptum (*Svetic et al., 2007*). As a result, dense S-iridophores are prevented from traveling via the horizontal myoseptum to generate the initial stripe of dense S-iridophores seen in WT at stage PB. Instead loose S-iridophores appear only later, at stage PR, uniformly across the domain. *cho* fish then proceed to develop a labyrinthine pigment pattern. Stripes and interstripes of normal width form in a parallel arrangement, but with orientation disrupted, with regions running vertically and horizontally and often strongly curved, sometimes branched and often interrupted (*Figure 7F*).

To model *cho*, we omitted the initial stripe of dense S-iridophores at the PB stage (*Figure 7O,O'*) and instead place 200 loose S-iridophores at random across the S-iridophore domain at stage PR (*Figure 7P,P'*). No other interactions were altered. At stage J+ (*Figure 7Q,Q'*), we see a pattern of normal width stripes and interstripes except with

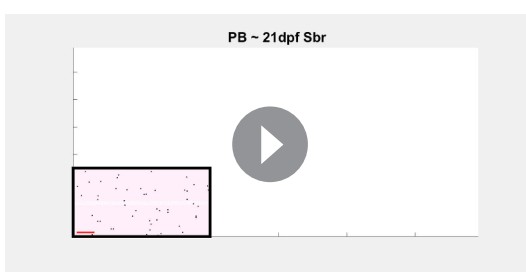

**Video 6.** Simulated development of *sbr*.
https://elifesciences.org/articles/52998#video6

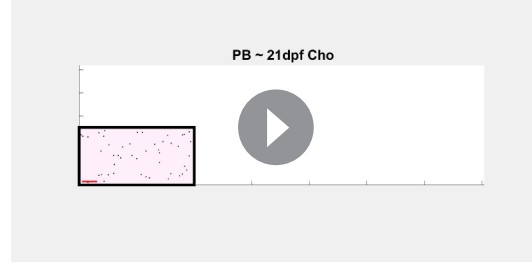

**Video 7.** Simulated development of *cho*.
https://elifesciences.org/articles/52998#video7

varying orientation, as seen in real *cho* fish. See *Video 7* for a movie of the simulation.

## The *seurat* mutant

Homozygous *seurat* mutants develop fewer adult melanophores, thus forming irregular spots rather than stripes. This phenotype arises from lesions in the gene encoding Immunoglobulin superfamily member 11 (Igsf11) (*Eom et al., 2012*) which encodes a cell surface receptor (containing two immunoglobulin-like domains) that is expressed autonomously by the melanophore lineage. Igsfl1 promotes the migration and survival of these cells during adult stripe development as well as mediating adhesive interactions in vitro.

To model *seurat*, we reduced the rate at which melanocytes could differentiate to a twentieth of the usual rate. This was to reflect the inhibition of the migration of melanoblasts (precursors of melanophores) across the domain and increased the rate of attempted melanocyte death to one hundred times per day (usually once per day). No other interactions were altered. At stage J (*Figure 7R,R'*), we see a pattern of normal width stripes broken into spots with a reduced number of melanocytes. Melanocytes are associated with loose S-iridophores and xanthophores with dense S-iridophores, as seen in real *seurat* fish.

By modelling *seurat* and *sbr*, we can also make predictions about the phenotype of a double mutant *seurat;sbr*, shown in *Appendix 5—figure 1*. We predict that by stage J+, this mutant would be covered in dense S-iridophore and associated xanthophores, with a few melanocytes at the very dorsal and ventral region of the fish. We are not aware of a published description of the phenotype of this double mutant, so our prediction remains to be tested.

## Regeneration experiments

In order to further validate our model, we test to see whether we observe similar behaviour when the cells are ablated and the pattern is left to regenerate. In 2007, *Yamaguchi et al., 2007* ablated a rectangular window of pigment cells of adult zebrafish stripes and recorded the regeneration of pigment producing cells (*Figure 8A–C*). They found that after ablation, cells regenerated in a labyrinthine pattern. To model this ablation, we simulated WT development from stage PB to our latest simulation stage, J+ as seen in *Figure 5D'*. At stage J+, we simulate ablation by removing a square region of cells in the centre (horizontally) of three stripes and interstripes. We then observe the pattern regeneration 7, 14 and 21 days later in *Figure 8C–E*. At 14 days, we observe the production of irregular shaped spots of melanophores in the centre of the ablated region as seen in the ablated fish at day 7. At day 21, we observe a regeneration of the pattern where stripes are no longer oriented horizontally. In some regions, spots of melanophores are surrounded by xanthophores.

In 2013, *Patterson and Parichy, 2013* ablated a section of dense S-iridophores along the horizontal myoseptum using a S-iridophore-specific marker *pnp4a*:NTR+Mtz at the beginning of pattern metamorphosis (*Figure 9A*, stage PB). They then observed the subsequent pattern formation (*Figure 9B*). We simulate this by removing a section of dense S-iridophores from the horizontal myoseptum at stage PB (*Figure 9C*) and then simulating as normal. We observe the pattern at 10 days post ablation. Xanthophores are associated with the undamaged portion of the S-iridophore stripe and melanophores surround the damaged interstripe (*Figure 9D*). In both cases, our simulations closely approximate the published observations.

## Quantitative analysis of simulations

In the next few sections (3.2.1)-(3.2.4), we test the consistency of our WT and mutant simulations, averaged over 100 simulations, with real published quantitative measures. We test four criteria using experimental data: (1) the number of melanophores in mutants at different stages with respect to WT at the same stage, (2) the average width of X0 interstripe for WT, *rse*, *pfe*, *shd* and *nac* at stage J+, (3) WT stripe straightness, (4) WT pigment cell density in stripes and interstripes.

## Melanophore density across WT and mutant fish

*Figure 10A* is a comparison between the average number of melanocytes per ventral hemisegment for each mutant with the number of melanocytes at WT at stages PB, PR, SP, J and J+ using the data from *Frohnhöfer et al., 2013*. First, since we do not have simulated data for a whole hemi-segment we normalise our melanocyte numbers against WT numbers at each stage. We do this by, for each respective stage and each respective mutant, multiplying the number of simulated melanocytes

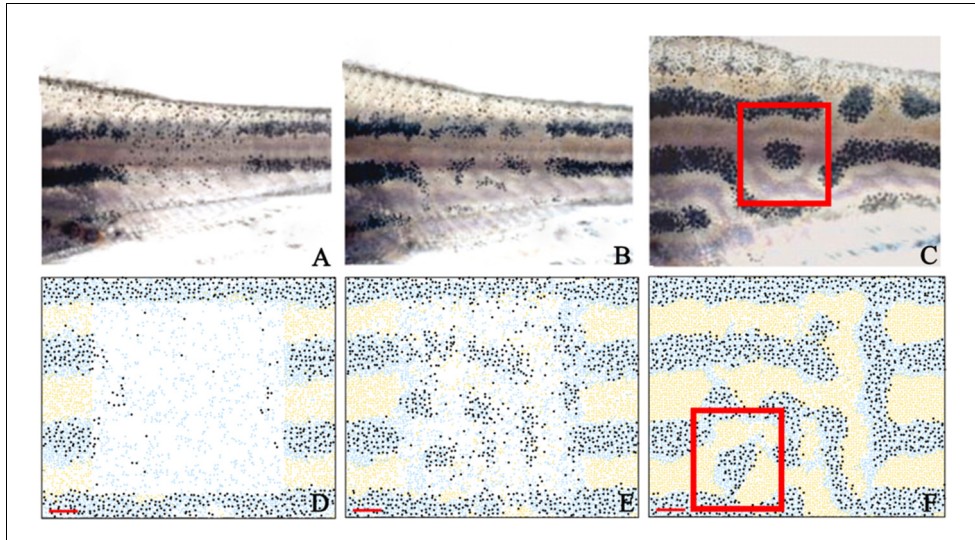

**Figure 8.** Stripe regeneration simulations. (**A–C**) Regeneration of new pigment producing cells 7, 14 and 21 days respectively after a small rectangular window of cells in the adult WT stripes are completely ablated (*Patterson and Parichy, 2013*). (**D–F**) A representative simulation of the regeneration of an adult zebrafish 7, 14 and 21 days after a simulated ablation has occurred. Scale bar is 0.25 mm in all images. Experimental images (**A–C**) are reproduced from *Yamaguchi et al., 2007*. Copyright (2007) National Academy of Sciences U.S.A.; these images not covered by the CC-BY 4.0 licence and further reproduction of this panel would need permission from the copyright holder.

at the given stage for the given mutant by the number of melanocytes at each stage in real WT fish and dividing by the number of melanocytes observed in our WT simulations at this stage. These comparisons are given in *Figure 10A*. We observe the same trends seen in real fish. Moreover, in all stages, except for stage SP, the simulated data falls within the error bars of the measured data in real fish. In particular, we observe that the number of melanocytes remains similar to WT in *pfe* until stage J+, similar to that in WT, whilst the number of melanocytes is significantly lower in *shd* and *rse* in comparison to WT just as in real fish.

## WT stripe straightness

In real life, zebrafish stripes are quite straight, but are not necessarily perfectly so (see X0 in *Figure 1A* for example). To measure stripe straightness, we first generated a line representation ($x$) of the central interstripe (see Appendix 3). From this line $x$, we calculated the stripe straightness SS ($x$), measured by the ratio of the length of our line ($L$) to the straight line distance between the ends of it ($C$), *i.e.*

$$\text{SS}(x) = \frac{C}{L}. \tag{1}$$

The value of SS($x$) lies between 0 and 1, since $C \leq L$. For more information about how $SS(x)$ is generated see Appendix 3. In 10 real WT stage J+ fish, the mean $SS$ value was 0.98. In 100 simulations we observe a high albeit slightly lower mean $SS$ value of 0.92 at stage J+, demonstrating good stripe straightness, that is close to that observed in real fish.

## X0 interstripe width across WT and mutant fish

Finally, we compare the interstripe X0 width in our simulations with real data. We choose this interstripe as it is the only interstripe that the mutants we consider and WT have in common. We

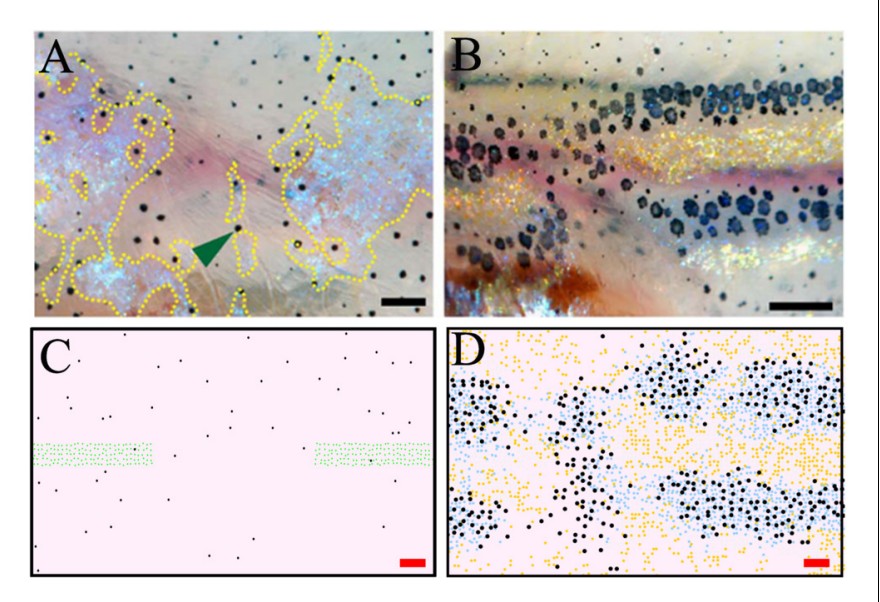

**Figure 9.** Stripe regeneration post S-iridophore ablation. (**A**) S-iridophores are ablated using *pnp4a*:NTR+Mtz at stage PB (**B**) Stripe development in wild-type fish following S-iridophore ablation. Xanthophores are associated with S-iridophore interstripe. Melanophores aggregate around the new interstripe. (**C–D**) A representative simulation of S-iridophore ablation and subsequent development. For clarity, dense S-iridophores are displayed in green in (**C**). Scale bar is 0.25 mm in all images. Experimental images (**A–B**) are reproduced from *Patterson and Parichy, 2013* and licensed under CC-BY 4.0 (http://creativecommons.org/licenses/by/4.0).

compare the width of this interstripe at J+ in our simulations (see Appendix 3 for detailed methods) with the observations of the corresponding J+ mutant in *Frohnhöfer et al., 2013*. We demonstrate in *Figure 5B* that, in all cases, the real J+ mutant X0 interstripe widths fall in the range of ±1 standard deviations of the simulated J+ X0 stripe width averaged over 100 simulations. This demonstrates consistency between our model and real data. Thus, our model demonstrates an excellent ability to quantitatively simulate the patterns of real fish.

## Pigment cell density in WT

There are no published estimates of WT pigment cell density in each of the stripes at the J+ stage when our simulations end. However, our data are comparable to those of adult WT fish measured by *Mahalwar et al., 2016*, who observed that in the stripe regions there were approximately four times more xanthophores in the interstripe region than melanocytes in the stripe region. Furthermore, whilst the light interstripe were completely devoid of melanocytes, there was a low density of xanthophores in the stripe region. In our model simulations, we observe a mean of 4.01 times as many xanthophores in the interstripes than melanocytes in the stripes demonstrating good agreement. We also observe a low density of xanthophores in the stripe regions and negligible numbers of melanocytes in the interstripe regions.

## Simulation reproducibility of pattern formation

To further test the accuracy of our model's outputs, we compare the spatial correlation of different cell types at different distances. We use this measure as an objective test of whether the spatial distributions between cells we observe in our representative simulations, (i.e. the patterns generated), are consistent among different simulated outputs.

To measure spatial correlation, we use a pair correlation function (PCF). A PCF determines whether, given a spatial distribution of agents on a domain, the number of pairs of agents at a certain distance from each other are greater than or fewer than the number expected if the agents were distributed uniformly at random. For example, if the PCF value is unity for a certain distance, this indicates that there is no spatial correlation. If the PCF value is greater or less than unity for a

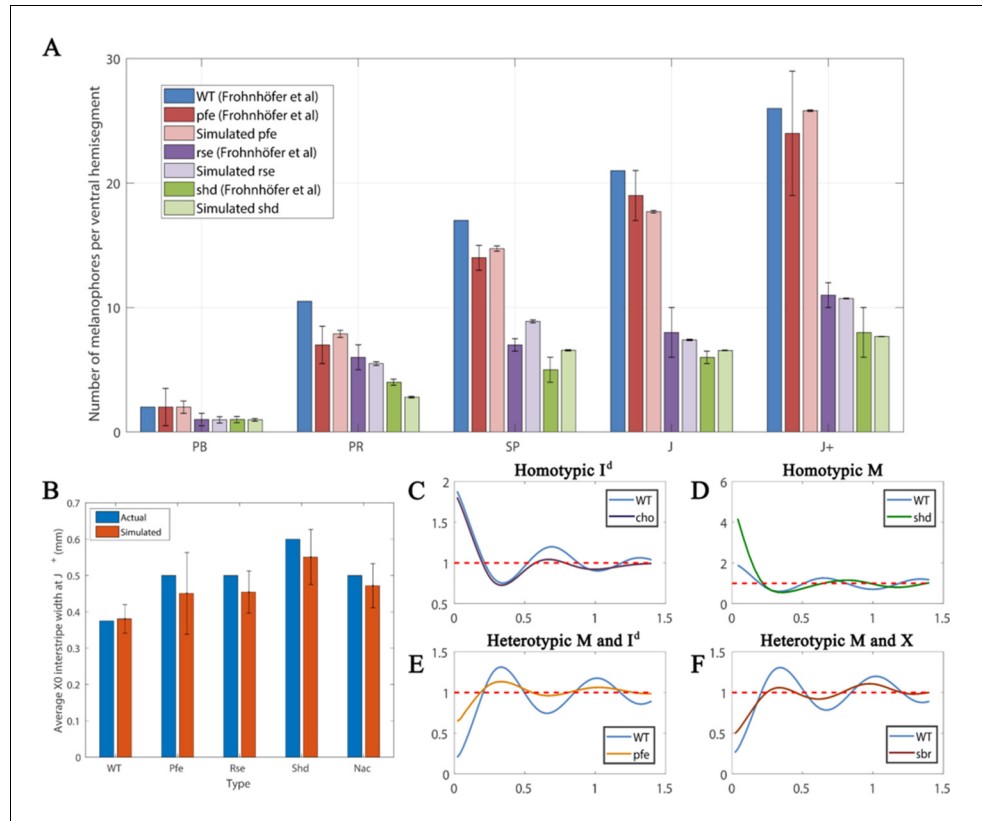

**Figure 10.** Quantitative analyses. (**A**) Number of melanocytes per ventral hemisegment. Quantification of melanocytes. A comparison of melanocyte numbers between mutants and WT seen in our simulations and data recreated from *Frohnhöfer et al., 2013*. For each stage, our simulated number of melanocytes was normalised by the WT number in data by *Frohnhöfer et al., 2013*. (The number of animals used for counting was at least five for each measurement point.) The number of simulations was 100. Error bars indicate the standard deviation. There is no error bar for WT, since WT data was used to normalise the data. (**B**) X0 interstripe width. A comparison of X0 stripe width in 100 simulations with X0 stripe width in real mutants (taken from representative images; *Frohnhöfer et al., 2013*). The darker bars depict experimental data, lighter bars depict our simulated data. Error bars indicate the standard deviation. Since there is only one image for each of the respective mutants, there is no error bar for actual (real experimental data). (**C–F**) PCF for different cell types averaged over 100 simulations using the square uniform PCF (*Gavagnin et al., 2018*). The spatial pair correlation as a function of distance for (**C**) dense S-iridophores only (**D**) melanocytes only, (**E**) melanocytes and dense S-iridophores, (**F**) melanocytes and xanthophores. Experimental data in (**A**) is reproduced from *Frohnhöfer et al., 2013*.

certain distance then this indicates that there is spatial correlation or anticorrelation respectively at that distance. The PCF we employ is specific to on-lattice domains and is called the Square Uniform PCF (*Gavagnin et al., 2018*) adapted for multiple cell-types (see Appendix 3 and *Dini et al., 2018*). We describe the PCF as homotypic when we are measuring the spatial correlation of one cell-type and heterotypic when we consider the spatial correlation between two different cell-types. We choose this PCF as it uses a measure of distance that is complementary to the distance measurement in our simulations.

*Figure 10C–F* shows the square uniform PCF against distance for different mutants and cell types averaged over one hundred simulations. For each plot of a given PCF type (homotypic melanocytes, for example), we repeatedly simulate the relevant mutant to its final stage, compute the PCF of the resultant pattern and then average the PCFs over the number of repeats to give us an averaged PCF.

To interpret the data, consider the representative simulation for a WT fish at stage J+. In this example, we observe stripes (*Figure 5D′*). This is a consistent feature for all of the repeat simulations of WT at stage J+. To quantify average interstripe width (the distance vertically in mm from the top

of any interstripe to the bottom) in our simulations we can consider the averaged homotypic dense S-iridophores PCF for WT in *Figure 10C*. We observe that this shows periodicity (sequential peaks and troughs) at different distances. These are a consequence of the striped pattern at J+ (*Figure 5D'*). Since dense S-iridophores occupy the interstripe regions in WT and not the stripe regions at J+, dense S-iridophores are spatially correlated at short distances, indicated by a positive value of the PCF at short distances. Conversely, they are anti-correlated at distances approximately one half, one and a half or two and a half stripe widths away, as these distances correspond to the relative positions of the dark stripes, which normally lack dense S-iridophores. We see troughs at these distances. The period of the PCF in this case thus quantifies an estimate for average interstripe width.

In the next few paragraphs, we test the reproducibility of different features that are observed in our representative simulations by considering a PCF of appropriate cell types averaged over one hundred repeats.

Real *cho* mutants and WT fish share similar interstripe width (*Frohnhöfer et al., 2013*)(also seen in our model; compare *Figure 7Q'* and *Figure 5D'*). To test reproducibility we consider the homotypic PCF of dense S-iridophores for WT and *cho* in *Figure 10C*. For both *cho* and WT simulations, the averaged homotypic PCF for dense S-iridophores observes periodic behaviour with the same frequency, indicating maintenance of interstripe width between WT and *cho* in our model, consistent with observations in vivo.

In real *shd* at stage J+, there are two pseudo-stripes of melanocytes broken into spots, of a diameter that is approximately equal to the normal stripe width (*Frohnhöfer et al., 2013*) (also seen in our model; compare *Figure 5H'* and *Figure 5D'*). To test whether this is consistent we consider the homotypic PCF of melanocytes for WT and *shd* in *Figure 10D*. The average stripe width of WT and the average aggregate size of *shd* can be approximated from the PCF as the distance related to the first trough, as this is the shortest distance at which melanocytes are most anticorrelated with other melanocytes. For both *shd* mutants and WT simulations, these are both approximately 0.3 mm.

In real *pfe* at stage J+, stripes and interstripes remain aligned and have the same width as in WT, except that stripes are broken into spots and some melanocytes lie ectopically in the usual interstripe region (*Frohnhöfer et al., 2013*) (also seen in our model; compare *Figure 5D'* and *Figure 5P'*). To test reproducibility, we consider the heterotypic PCF of melanocytes and dense S-iridophores for WT and *pfe* in *Figure 10E*. For both the WT and *pfe* simulations, the averaged heterotypic PCF of melanocytes and dense S-iridophores displays periodic behaviour with the same period. However, in *pfe* the peaks and troughs are damped. We interpret this as follows. Firstly, this indicates that, in our model, stripe width is preserved between *pfe* and WT as the period of the PCF is the same. Moreover, as the peaks and troughs are damped in *pfe*, this indicates that, as seen in our representative simulation, some melanocytes tend to remain in the interstripe regions.

In real *sbr* at 11.5 mm SL, stripes are sometimes broken into spots of usual (vertical) width, but the overarching stripe pattern remains (*Fadeev et al., 2015*) (also seen in our model, compare *Figure 7D'* and *Figure 5N'*). To test reproducibility, we consider the heterotypic PCF of melanocytes and xanthophores for WT and *sbr* in *Figure 10F*. The first peak of this PCF corresponds to the shortest distance at which melanocytes and xanthophores are most correlated, which is approximately the stripe width. For both *sbr* mutants and WT simulations these are both approximately 0.3 mm.

In these examples, using appropriate PCFs we have demonstrated the consistency of our simulations in generating patterns that match the qualitative differences we expect when we compare mutant fish with WT. We note that we have only provided the averaged PCF for the scenarios aforementioned for simplicity. For information about how the PCF is calculated, please see Appendix 3.

## Iridophore assumptions and biological redundancy
### Necessity of S-iridophore assumptions

For the less well-studied S-iridophore transitions, we analysed key mutant phenotypes to infer biologically realistic rules for these interactions, aiming to generate assumptions that were the simplest for pattern formation changes seen, but no simpler. These deductions are discussed in Section "Modelling overview". In *Figure 11* B1–J3, we demonstrate the necessity of all of the assumptions for dense-to-loose, loose-to-dense S-iridophore transitions first outlined in *Figure 4G*, 15-18 for stripe formation by showing representative images (the model was run 50 times each) of simulations

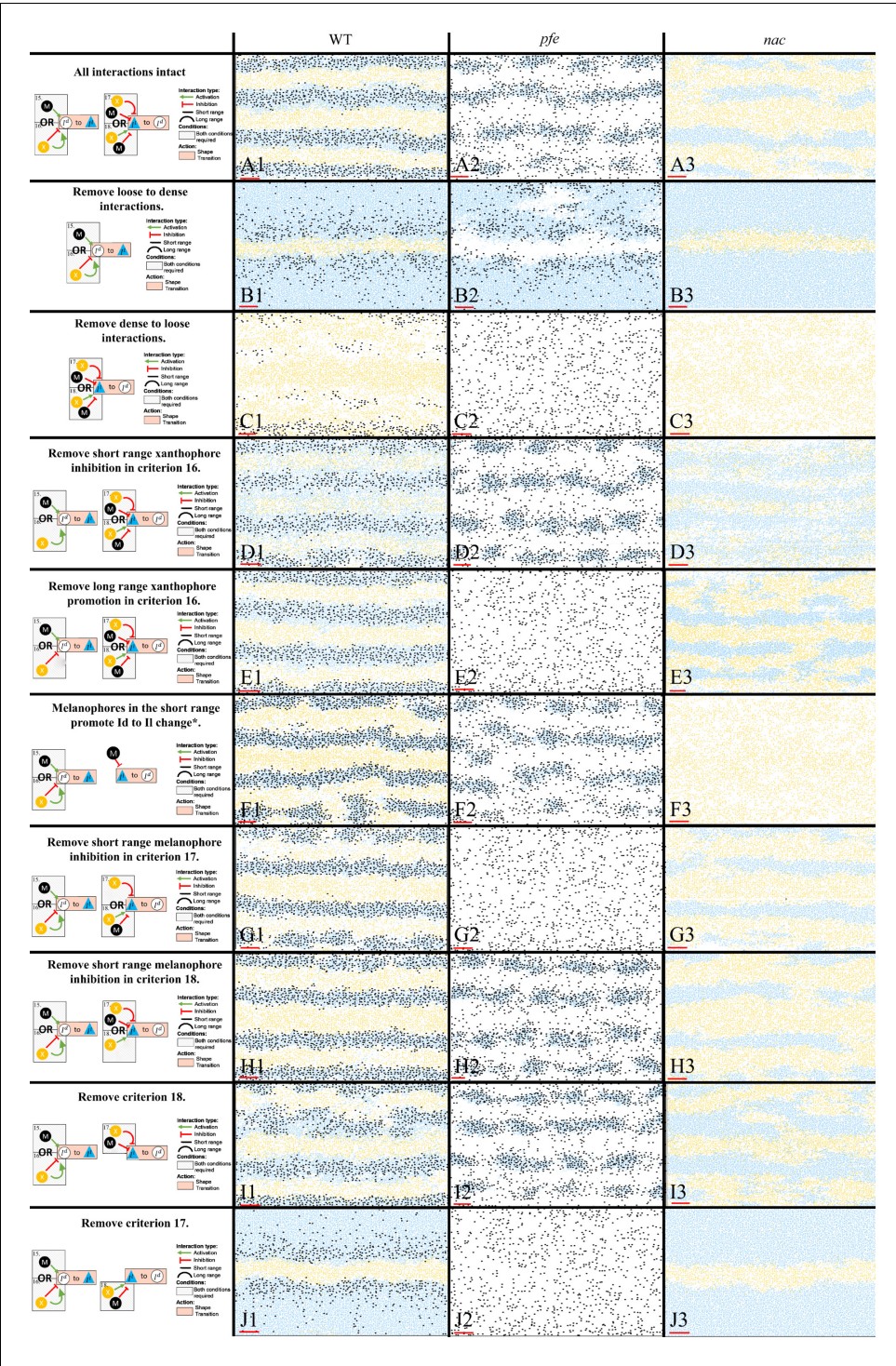

**Figure 11.** Representative simulations of the model with some of the S-iridophore interactions removed. The first column displays a diagram of the S-iridophore interactions which remain (all other cell-cell interactions are unchanged). Columns 2–4 are representative simulations of WT, *pfe* and *nac* under these conditions. *This interaction is equivalent to removal of long range xanthophore inhibition in criterion 17. It is also equivalent to removal of short range xanthophore promotion in criterion 18.

at stage J+ for fish lacking one or more S-iridophore transition mechanisms display patterns which diverge from those seen in real fish.

First, we analyse simulations lacking one of the transition types loose-to-dense or dense-to-loose (*Figure 11* B1–C3). Without a loose-to-dense transition (*Figure 11* B1–B3), note how in all cases (WT, *pfe*, *nac*) only one pseudo-interstripe is preserved: the initial X0 interstripe. The X0 interstripe is surrounded by a sea of loose S-iridophores which have transitioned to loose either by promotion by xanthophores in the long range (*nac Figure 11* B3), promotion by melanophores in the short range (*pfe Figure 11* B2) or a combination of both (WT *Figure 11* B1). This suggests that loose-to-dense interactions are important for generating subsequent interstripes. Interestingly, without loose-to-dense transitions WT fish demonstrate a striped pattern similar to *Danio albolineatus* (*Parichy, 2006*) suggesting a possible route of evolution between these fishes (also noted by *Volkening and Sandstede, 2018*). Without a dense-to-loose transition (*Figure 11* C1–C3), the S-iridophores form a dense sheet over the entire domain with xanthophores and melanophores scattered across the domain. This demonstrates the necessity for S-iridophores to be able to transition between dense and loose form.

Next, we consider removing each of the criteria required for an S-iridophore transition one at a time (*Figure 11* D1–J3). We first note that, in some cases, removal of an interaction in either *nac* or *pfe* results in loss of a transition type. These are not shown in *Figure 11* for simplicity. In other scenarios, however, removal of an interaction leads to the uninhibited possibility of a transition in one of the single cell mutants. For example, consider *Figure 11* E2. Removal of long-range xanthophore promotion in criterion 16, leads to the possibility of a transition from loose to dense provided that there are either melanophores in the short range *or* no xanthophores in the short range. Since in *pfe* there are are always no xanthophores in the short range, or indeed anywhere on the domain, S-iridophores are consistently promoted to dense type, with a non-zero rate, thus *Figure 11* E2 is not distinguishable from *Figure 11* C2 since in effect, in *pfe* the same interactions have been knocked out. We note that this is the case for one of either *nac* or *pfe* in *Figure 11C,E–G,J*.

In *Figure 11* D1–D3, short-range xanthophore inhibition is removed from criteria 16. As this is exclusively a xanthophore-iridophore interaction this only effects WT and *nac*. Without the promotion of xanthophores in the short range, S-iridophores change from dense to loose in the interstripes, making the interstripes appear faded. In *Figure 11* G1–G3, short-range melanophore inhibition is removed from criteria 18. Therefore S-iridophore can change from loose to dense when there are xanthophores in the short range. As a result interstripes become wider as they are unrestricted by local melanophore stripes. Finally, in *Figure 11* I1–I3 criteria 18 is removed, so S-iridophores only change from loose to dense when there are no melanophores in the short range and simultaneously no xanthophores in the long range. In this case stripe integrity is lost in WT.

To summarise, all interactions are necessary for pattern formation in WT and single cell mutants, *nac* and *pfe*.

## Biological redundancy

As part of this in-depth study, we have incorporated all of the interactions that we have identified from the literature. Consequently, there may be some in-built redundancy. However, we keep all interactions for the purposes of biological realism. In this section we explore the idea of biological redundancy by removing some interactions and observing the resultant simulated development.

First we consider movement. In real (and simulated) fish, melanocytes and xanthophores move 0.11 mm per day (*Takahashi and Kondo, 2008*) and 0.033 mm per day, respectively. Furthermore, in the short range their direction of movement is influenced by each other (*Yamanaka and Kondo, 2014*). Xanthophores chase melanocytes, which in turn, are repelled by xanthophores. In *Figure 12C–D*, we turn off the movement of xanthophores and melanocytes in WT and *shd* mutants and simulate to stage J+. We observe that in both cases, the pattern is conserved. This is not surprising since the cells do not move very quickly. However, there is a slight difference in *shd* wherein the interstripe width is slightly smaller than in *shd* without changes (*Figure 12B*). We suggest that this is due to the loss of chase-run dynamics observed between melanocytes and xanthophores. Next we consider the differentiation rate of melanocytes. We model this as being dependent on the number of dense S-iridophores and the number of xanthophores currently on the domain. This is because we assume that dense S-iridophores and xanthophores positively influence the rate of

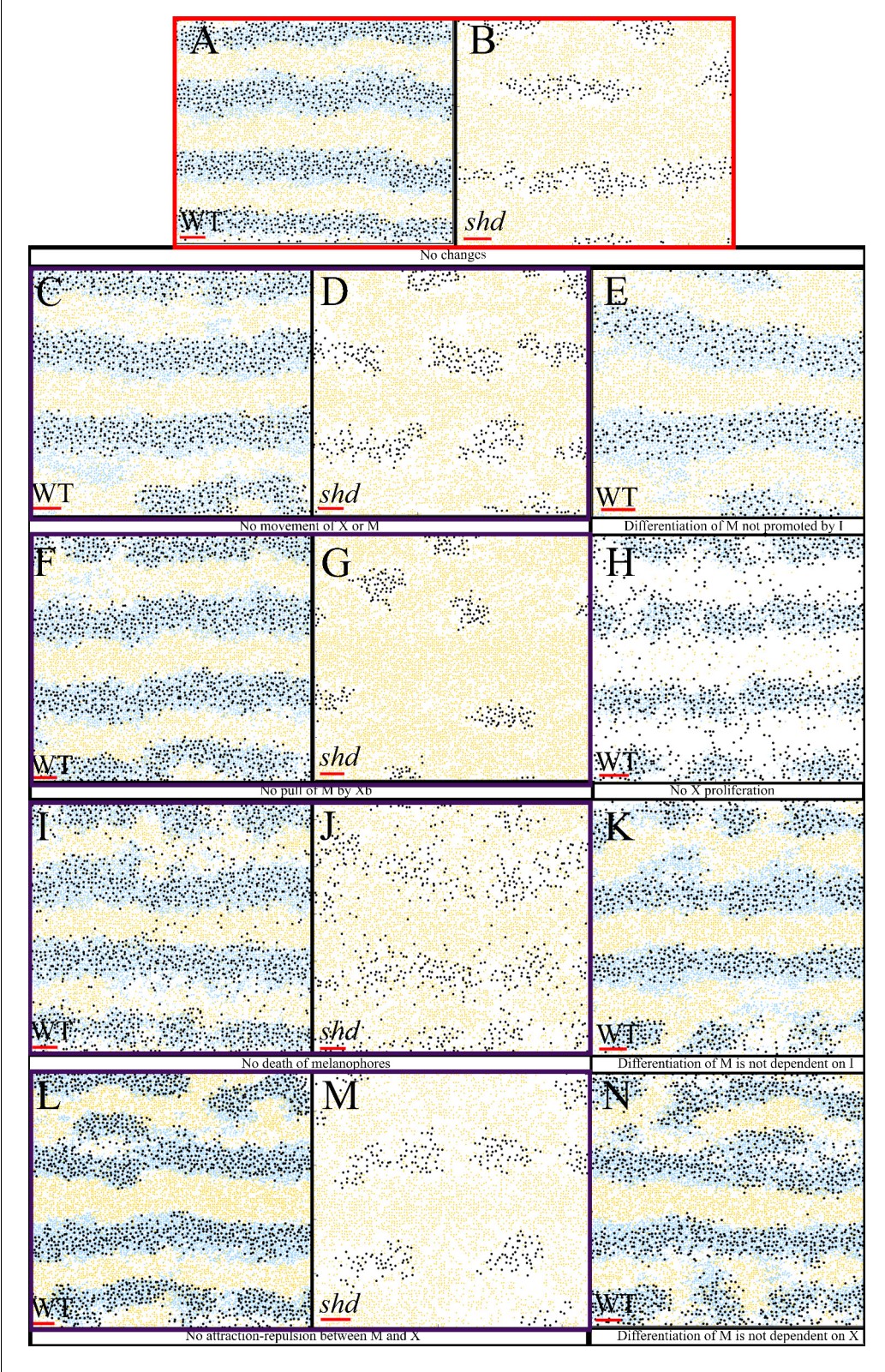

**Figure 12.** Representative simulation of the model with certain rules omitted. (A–B) WT and *shd* respectively with all rules included. (C–D) No movement of melanocytes or xanthophores for WT and *shd*, respectively. (E) Differentiation of melanocytes only promoted by dense S-iridophores for WT fish that is the rate of melanocytes does not depend on the number of S-iridophores. (F–G) No pull of melanocytes by xanthoblasts for WT and *shd* fish, respectively. (H) No proliferation of xanthophores for WT. (I–J) No melanocyte death for WT and *shd* fish. (K) Differentiation of melanocytes is not

*Figure 12 continued on next page*

*Figure 12 continued*

dependent on S-iridophores, (**L–M**) No death of melanocytes for WT and *shd* (**N**) Differentiation of melanocytes is dependent on dense S-iridophores being present in the long distance only.

melanocyte birth in the long range. In *Figure 12E*, we change the differentiation rate so it is only dependent on the number of xanthophores currently on the domain and not the number of S-irido-phores, effectively reducing the rate of differentiation of melanocytes in wild-type fish. The resultant pattern is still striped; however, it is less organised. In *Figure 12F–G*, we remove the mechanism allowing long-range communication between xanthoblasts and melanocytes. In *shd* our model pre-dicts that without the consolidation of spots by xanthoblasts, melanocyte spots become more widely spaced. In *Figure 12H*, we remove xanthophore and xanthoblast proliferation. This limits the num-ber of xanthophores to the number allocated at the start. Remarkably, our model predicts that stripe formation is largely preserved, however, interstripes are fainter due to the lack of xanthophores. Next we consider melanocyte death. In *Figure 12I–J*, we remove melanocyte death from WT and *shd* simulations. Whilst stripe formation is maintained in WT (*Figure 12I*), interstripes are littered with melanocytes. In *shd* (*Figure 12J*), melanocyte spots are more difficult to discern as melanocytes can be observed in the xanthophore regions. We predict that dense S-iridophores and xanthophores in the long-range and/or loose S-iridophores and the lack of xanthophores in the short range pro-mote melanocyte differentiation. In *Figure 12K* we change the criteria for melanocyte differentia-tion, so that it does not depend on S-iridophores (only xanthophores). As a result the stripe pattern looses integrity. A similar phenotypic change happens when we change the criteria for melanocyte differentiation, so that it does not depend on xanthophores (only S-iridophores, *Figure 12N*). In *Figure 12L,M*, we change melanocyte and xanthophore movement so that they are no longer influ-enced by each other (no chase-run dynamics). As a result in WT (*Figure 12L*), stripe integrity is lost. For *shd* (*Figure 12M*) simulations, however, qualitatively there is not much difference (similarly to the case when movement is completely removed (*Figure 12D*) suggesting that movement does not play a significant role in generating spots. Rather, it is the death of melanocytes in xanthophore rich areas and xanthoblast pulling which are important (*Figure 12G,J*).

In summary, whilst removal of these interactions largely do not change the type of pattern gener-ated (e.g. *shd* still generates spots, wild-type fish still generate stripes), and thus could be consid-ered as biologically redundant for pattern generation, they appear to have large impacts on the integrity of the patterns formed. We thus, suggest that the retention of these interactions in vivo act as a buffer to protect the integrity of stripe formation in spite of stochastic variations in stripe patterning.

## Model predictions

A major benefit of developing a fine-grained, cell-level model is the ability to perform in silico experiments that can be directly related to real-life equivalents. This not only allows us to explore parts of the system that may otherwise not be testable experimentally, giving us valuable insight into the biological system. It also gives us the ability to analyse dynamics of different hypothetical mechanisms before devoting expensive resources to experimental tests that could confirm theoreti-cal findings.

In the next few sections, we focus on the ability of the model to make biologically testable predic-tions, demonstrating firstly, in Section "An in silico investigation into important mechanisms for con-trolling pattern formation", how we can use our model to explore important facets of successful pattern formation such as growth, domain size and initial conditions. Then in Section "An in silico investigation into the function of the *leo* gene", we give an example of how we can use our model to generate testable hypotheses about the *leopard* mutant.

# An in silico investigation into important mechanisms for controlling pattern formation

## Initial S-iridophore interstripe orientation alone does not determine the orientation of stripes and interstripes

Previously it has been hypothesised that the horizontal orientation of the initial S-iridophore interstripe (emerging from the horizontal myoseptum) that drives the organisation of subsequent stripes and interstripes horizontally. One way to test this hypothesis is to initialise the interstripe so it is oriented vertically instead of horizontally. If the initial S-iridophore interstripe does orient stripes and interstripes then we would expect to see the same pattern development we observe in WT fish, but rotated 90 degrees. That is, we would expect to see vertical bars across the domain at the time corresponding to stage J+. The position of the dense S-iridophores (a horizontal interstripe along the horizontal myoseptum in WT fish) at the start of pattern metamorphosis, is dictated by the fish's anatomy and cannot be altered experimentally. However, we can simulate an altered iridophore initial distribution in silico by initialising the initial interstripe as a band of width three along the vertical axis instead of as a band of dense iridophores vertically (dorso-ventrally) down the centre of the domain of width three instead of as a band of width three, dense S-iridophores along the horizontal axis. We observe the subsequent pattern development at stages PR, SP and J+ in *Figure 13A*. Interestingly, instead of observing vertical bars, at stage J+ we observe a labyrinthine pattern. This demonstrates that, whilst the initial S-iridophore interstripe plays a role in orientating the pattern, it is not the only part of the initial condition that is important. Further observation of the model output reveals that, for a while, the pattern is oriented in a vertical pattern similar to the initial iridophore interstripe, but as growth continues, this pattern becomes reoriented into a horizontal form. This clearly reveals the impact of growth on pattern formation.

## The position of the initial S-iridophore interstripe is important for successful pattern formation

Another way to better understand the role of the initial S-iridophore interstripe is to alter its position in the dorso-ventral axis so that it appears more ventrally on the initial domain. We might naively predict, given that it has been hypothesised that dense S-iridophores only orientate the stripes and interstripes, that in this case the final pattern would be the same as in WT fish. We simulate this in silico by initialising the initial interstripe to be one quarter of the way up the domain instead of half way. A typical pattern evolution for this initial condition is displayed in *Figure 13B*. We observe that subsequent stripes and interstripes still appear sequentially, either side of the initial interstripe, suggesting that the S-iridophores do play a role in the positioning of new stripes and interstripes. However, we do not observe usual WT patterning. In particular, stripes and interstripes exhibit more breaks compared to WT simulations. Moreover, developing stripes and interstripes become sequentially thinner as a result of the impact of domain growth. Once again, growth is the key factor: growth is centred at the middle of the domain and so when the initial stripe is not similarly centred, growth disrupts pattern formation in our model.

## Initial domain size contributes to the number of stripes and interstripes

In order to test the role of domain size in pattern development, we initialise the domain so that it is three times as tall as in WT simulations. That is, we initialise the domain to be 2 mm × 3 mm instead of 2 mm × 1 mm. All other parameters remain unchanged, including the rate of growth in the horizontal and vertical axis. We present a typical example of subsequent pattern development in *Figure 13C*. At stage PR, pattern development is similar to the pattern seen in WT fish at the same stage. However, at stage SP, we observe more stripes than are observed even by stage J+ in WT fish. By stage J+, instead of three interstripes and four stripes as seen in WT, we observe six stripes and seven interstripes. This suggests that the initial domain height influences the number of stripes and interstripes that develop, provided that growth is uniform and centred.

## Stripe insertion can occur on an initially striped domain

Kondo and Asai observed that, as the size of the marine angelfish *Pomocanthus* doubled, new stripes along the skin would develop between the old ones (*Kondo and Asal, 1995*). This phenomenon has not been observed in zebrafish, where new stripes and interstripes appear consecutively at the dorsal and ventral periphery. We hypothesise that this is likely related to either pattern

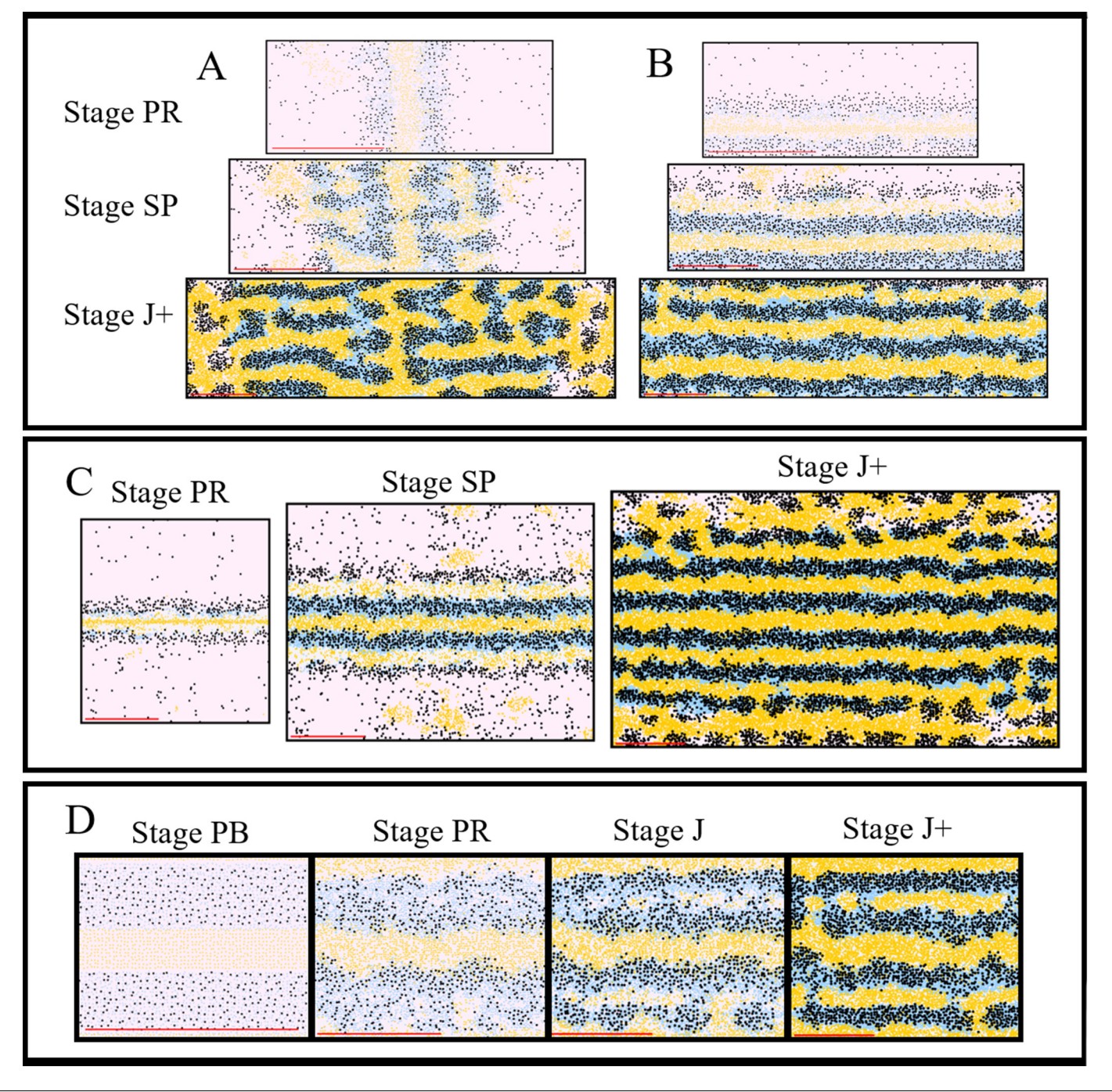

**Figure 13.** In silico investigation into important mechanisms for controlling pattern formation. The simulated domains at stages PR, SP and J+ wherein the following are changed (A) the orientation of the initial S-iridophore interstripe, (B) the initial position of the S-iridophore interstripe, (C) the initial domain size (D) the initial domain so that it is populated with adult-width stripes and interstripes.

maintenance mechanisms or the spatial localisation of growth. Here, we experiment with the model to see whether stripe insertion can occur when the domain is populated at stage PB with adult-width stripes and interstripes. The results of an example realisation with these initial conditions are given in *Figure 13(D)*. We observe that, in this case, new interstripes do appear between pre-established stripes. This is because growth (which is centred in the middle of the domain) creates space within the middle of the already developed stripes and interstripes.

## S-iridophores are more important to the generation of melanocytes than xanthophores

We also used the model to make some more subtle predictions. For example, in the case of melanocyte differentiation, which we model as being promoted in the long range by both xanthophores (from observation of ablation experiments; *Kondo and Asal, 1995*) and S-iridophores (from observations of *pfe*), there were no known parameter values for their relative strengths. We found using our model that by making the strength of S-iridophore promotion of melanocyte differentiation to be much greater than that of xanthophores, qualitatively and quantitatively the model simulated for WT, *pfe* and *shd* was greatly improved (see *Figure 14A–B*).

## Horizontal growth bias during development generates more tortuous stripes in WT fish

Interestingly, we also observed in our simulations that increased height-to-length ratio is correlated with stripes becoming more tortuous (R = −0.617, p<0.01, *Figure 14C*). This phenomenon is not

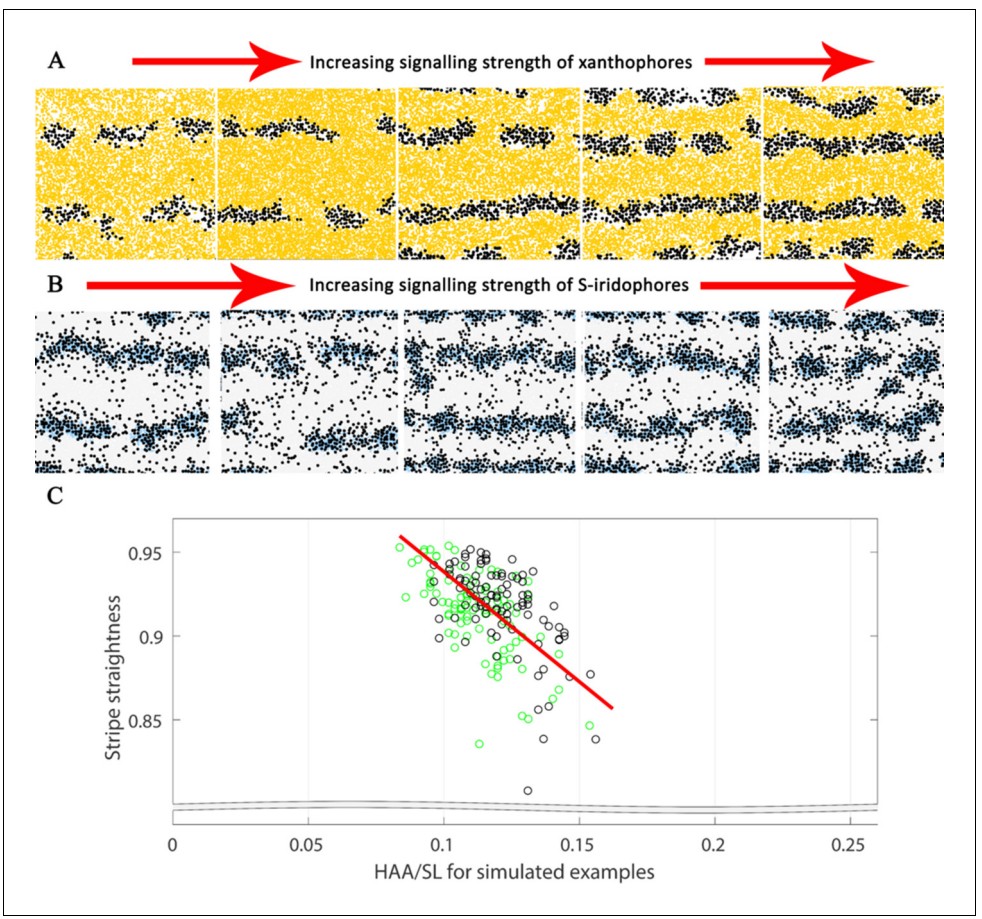

**Figure 14.** Substantial alterations in starting conditions generate major alterations in pattern formation. The effects of different signalling strengths in *pfe* and *shd*. Representative images of J+ simulations of (**A**) *shd* and (**B**) *pfe* where the signalling strength of melanocyte differentiation in the long range is increased (from left to right). Increasing the signalling strength of the respective cell types decreases the width of the interstripe X0 and increases the number of pseudo-stripes from 2 to 4. For our simulations, we choose the weakest shown signalling strength for xanthophores, corresponding to the furthest left of (**A**) and the strongest shown signalling strength for S-iridophores, corresponding to the furthest right of (**B**). (**C**) Stripe straightness of our simulations correlates with the simulated HAA/SL ratio. Each circle is one simulation, green circles indicate simulations up to stage J, black circles indicate simulations of up to stage J+. There are one hundred simulations in total for each stage. The red line is the line of best fit.

something we can see as being consistent with real fish and thus suggests that some interactions may be missing regarding the maintenance of stripe and interstripe formation.

## An in silico investigation into the function of the *leo* gene

The model provides testable hypotheses for cryptic functions of the *leo* gene

The gene *leo* encodes Connexin39.4 (Cx39.4) (*Watanabe et al., 2006*; *Maderspacher and Nüsslein-Volhard, 2003*; *Irion et al., 2014*). As a result, *leo* mutants display a leopard-like spotted pattern across the flank of the fish (*Figure 15A–A'*), instead of the usual striped pattern (*Figure 1A*). In this section, we aim to hypothesise key aspects of the *leo* mutations using our model alongside observations of relevant mutants.

Pattern formation is also altered in the double mutants *leo;shd*, *leo;nac* and *leo;pfe* when compared with *shd*, *nac* and *pfe* (*Irion et al., 2014*). For example, the flank of double mutant *leo;nac* is covered by xanthophores and dense S-iridophores (*Figure 15B''*). This is in contrast to *nac* which also contains large patches of loose S-iridophores (*Figure 15B–B'*). Adult *leo;pfe* fish exhibit randomly distributed melanocytes instead of spots (*Figure 15C–C''*). Finally, adult *leo;shd* exhibit an absence of melanocytes on the flank of the fish, instead, the flank of the fish is entirely covered with xanthophores. This is contrast to the melanocyte spots normally observed on *shd* (*Figure 15D–D''*).

Connexins are involved in cell–cell communication and signalling. Since, Cx39.4 is required for normal function in melanocytes and xanthophores but not in S-iridophores (*Watanabe et al., 2006*; *Maderspacher and Nüsslein-Volhard, 2003*; *Irion et al., 2014*), this suggests that in *leo*, cell–cell communication between melanocyte and xanthophores may be disrupted. Moreover, from observation of the double mutants, it seems that *leo* presumably generate heteromeric gap junctions among and between melanophores and xanthophores, controlling S-iridophore shape transitions (*Irion et al., 2014*).

In order to investigate the influence of the *leo* gene, we first consider the individual cell–cell interactions that can be deduced from the literature and ask if these cell–cell interactions are sufficient for generating the pattern. So far, there has been one experimental study observing the individual behaviour of *leo* cells. *Kondo and Watanabe, 2015* studied the movement in-vitro of *leo* melanocyte and xanthophore cells. They demonstrated that the *leo* melanocyte repulsive response to xanthophores was hardly observed in comparison to the marked repulsion in WT fish. This suggests that melanocyte repulsion from xanthophores is inhibited in *leo* (*Kondo and Watanabe, 2015*). We will refer to this as hypothesis one for the effects of the mutant *leo*.

- Hypothesis 1:Melanocytes are not repelled by xanthophores.

We can simulate pattern development in this case by turning off melanocyte repulsion by xanthophores in our model. This is shown in *Figure 15E* in the column numbered 1. We observe that at J+ the pattern consists of thicker interstripes than in WT fish, but not spots. Simulating the *shd* phenotype (lack of S-iridophores) with hypothesis 1, also does not generate the pattern expected in *leo;shd*. Hypothesis one is also insufficient to explain the phenotype of *leo;nac* or *leo;pfe*. This is because there are either no xanthophores or no melanocytes respectively in these mutants and therefore no melanocyte-xanthophore interactions. From these observations, we conclude that hypothesis one alone is insufficient for *leo* pattern formation.

To deduce the other cell-cell interactions that a mutation in *leo* could affect, we look at the patterns of adult *leo;nac* (*Figure 15B''*) and *leo;shd* (*Figure 15D''*). Of the three double mutants, these are the easiest from which to deduce single cell-cell interaction changes that could generate the double mutant patterns.

The *leo;nac* mutant displays an absence of loose S-iridophores compared to *nac*, suggesting that signalling of xanthophores which promote S-iridophore transition from dense to loose is inhibited in *leo*. The *leo;shd* mutant displays an absence of melanocytes in the adult pattern compared to *shd*, suggesting that long-range survival signal sent by xanthophores to melanocytes is inhibited in *leo*. We propose two further potential hypotheses for the effects of the mutant *leo*.

- Hypothesis 2: Xanthophores do not promote the survival of melanocytes in the long range.
- Hypothesis 3: Xanthophores do not promote the change of S-iridophores from dense to loose in the long range.

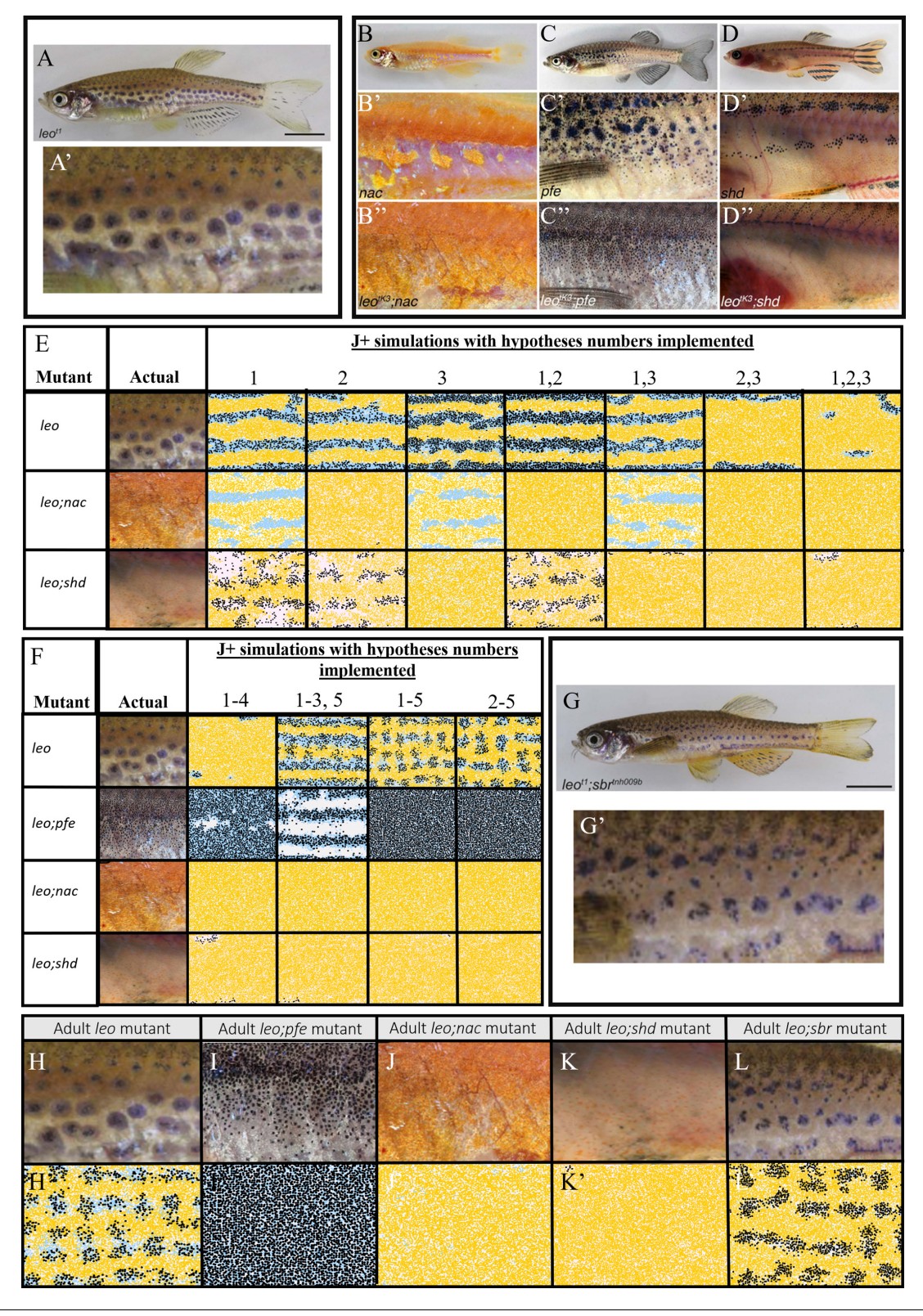

**Figure 15.** Using the model to generate predictions regarding the underlying effects of the *leo* mutant. (A–D) Adult *leo*, *nac*, *pfe* and *shd* respectively. (A'–D') The flanks of *leo*, *nac*, *pfe* and *shd* respectively. (B''–D'') The flanks of *leo;nac*, *leo;pfe* and *leo;shd* respectively. (E) Example simulations where combinations of hypotheses 1–3 in the text are implemented. (F) Example simulations where combinations of hypotheses 1–5 in the text are implemented. Note that simulations of 1–5 are similar to simulations for 2–5, indicating that hypotheses one is not necessary to predict the *leo*

*Figure 15 continued on next page*

*Figure 15 continued*

phenotype. (G) Adult *leo;sbr* (G') Flank of *leo;sbr*. (H–L) Adult mutants contrasting with (H'–L') simulations using hypotheses 1–5 from the main text. (H–L) Flank of *leo, leo;nac, leo;pfe, leo;shd* and *leo;sbr* respectively. (H'–L') Simulated images of *leo, leo;pfe, leo;nac, leo;shd* and *leo;sbr* respectively at stage J+. (A), (A'), (G), (G'), (H) and (L) from *Fadeev et al., 2015*, (B–D''), (I–K) from *Irion et al., 2014* and are all licensed under CC-BY 4.0 (http://creativecommons.org/licenses/by/4.0).

In *Figure 15E*, we provide a table of results for all combinations of hypotheses 1–3. We comment that hypotheses 1–3 cannot be all-encompassing, as so far, none of our hypotheses effect melanocyte-iridophore interactions and thus cannot generate the phenotype *leo;pfe*. Meanwhile, we focus on being able to generate *leo;nac* and *leo;shd*.

In *Figure 15E*, we demonstrate, that none of the hypotheses alone can generate the *leo* pattern, nor can they generate more than one of the double mutant types. When hypotheses 1 and 2 are combined, striping is disrupted, there are more breaks than in WT and interstripes are wider than WT. When 1 and 3 are combined, stripes are the same as in WT, presumably due to compensation of other cell-cell interactions, however, the simulated *shd* pattern has less aggregation of melanocytes than typically observed in *shd*. When hypotheses 2 and 3 are combined we observe the expansion of the initial interstripe to the very edges of the domain dorso-laterally, where there are some melanocytes, unlike *leo*. However, in this case, our simulations of *leo;nac* and *leo;shd* match the real phenotype. We demonstrate that when all hypotheses are implemented we generate a unstable pattern of spots which, eventually disappear over time, leaving a domain consisting of dense S-iridophores and xanthophores. Also, we can replicate *leo;nac* and *leo;shd*.

Finally, we attempt to elucidate the melanocyte-iridophore cell-cell interactions that are affected in the *leo* mutant by considering the phenotype of adult *leo;pfe* (*Figure 15C'–C''*). The *leo;pfe* mutant displays a random distribution of melanocytes below a field of S-iridophores, suggesting that *leo* affects directed differentiation of melanocytes by dense S-iridophores. This leads us to two extra hypotheses for the effects of the *leo* gene:

- Hypothesis 4: Melanocytes lose death signals from local dense S-iridophores and, as a result, can differentiate in dense S-iridophore zones.
- Hypothesis 5: Melanocytes lose directed signalling from S-iridophores and hence, in the absence of xanthophores differentiate randomly.

In *Figure 15F*, we provide a table of representative simulated patterns for all combinations of hypotheses 1–3 with hypotheses 4 and 5. Any of the combinations considered can generate the *leo;shd* and *leo;nac* mutant phenotypes. Hypotheses 1–4 and 1–3,5 cannot generate the *leo* spots. However, if we combine all five hypotheses then we can replicate the phenotypes of all considered mutants; *leo, leo;pfe, leo;nac* and *leo;shd*. These results suggest that hypotheses 1–5 are sufficient for explaining *leo*.

## Necessity of the pre-existing hypothesis about *leo*

Next, we evaluate the necessity of hypothesis (1) Hypothesis 1 is the only assumption that comes directly from the literature (*Yamanaka and Kondo, 2014*). We show, using our model, in *Figure 15F* that there is no notable difference between the patterns generated with and without hypothesis 1 (i.e. hypotheses 1–5 versus hypotheses 2–5) other than that the spots may be slightly more irregular in when hypothesis 1 is included. This suggests that the cell–cell interaction mediating repulsion of melanocytes by xanthophores in *leo*, may not be necessary to generate the characteristic spots.

## Hypotheses 1–5 can replicate the patterns of *leo, leo;nac, leo;pfe* and *leo; shd*

*Figure 15H—K* display the flanks of *leo, leo;pfe, leo;nac* and *leo;shd* respectively. *Figure 15H'–K'* display the simulated patterns at stage J+ for in silico mutants *leo, leo;pfe, leo;nac* and *leo;shd* respectively where our assumptions for the function of *leo* are based on hypotheses 1–5 and, in the case of double mutants *leo;pfe, leo;nac* and *leo;shd*, our assumptions for *pfe, nac* and *shd* are as previously described in Section "Simulation of 'missing' cell mutants".

We observe, for our simulations of *leo* (*Figure 15H′*), that melanocyte spots are associated with loose S-iridophores in a sea of dense S-iridophores and xanthophores just as in real *leo* fish (*Figure 15H*). For our simulations of *leo;nac* (*Figure 15I′*) we observe a domain that is fully populated with xanthophores and dense S-iridophores as expected (*Figure 15I*). Our simulations of *leo; pfe* (*Figure 15J′*) is fully populated with S-iridophores and randomly distributed melanocytes as expected (*Figure 15J*). Finally, our in silico representation of *leo;shd* (*Figure 15K′*) is fully populated with xanthophores only, as seen in *Figure 15K*.

### The *leo;sbr* cross mutant phenotype is an emergent property of the model

Previously in Section "Simulation of other mutants" we considered the *sbr* mutant. The *sbr* gene encodes Tight Junction Protein 1a (Tjp-1a), which is expressed cell autonomously in dense S-iridophores and causes the shape transition from dense to loose to be delayed (*Fadeev et al., 2015*). We showed in *Figure 7K′–N′* that our model could replicate the *sbr* pattern development by altering the rate at which S-iridophores attempt to change from loose to dense to be slower than in WT.

*Figure 15G—G′* shows a typical example of the *leo;sbr* phenotype. The adult *leo;sbr* displays spots which, compared to *leo* (*Figure 15A–A′*), are more elongated. *Figure 15L* displays the flank of an adult *leo;sbr*. In *Figure 15L′*, we simulate a mutant that satisfies hypotheses 1–5 as well as our assumptions about *sbr* to generate an in silico leo;sbr. Consistent with real *leo;sbr* mutants, our simulation of *leo;sbr* has melanocyte spots associated with loose S-iridophores that are surrounded by xanthophores and dense S-iridophores. Comparison with our simulation of *leo* (*Figure 15H′*) demonstrates that our simulation of *leo;sbr* also displays more elongated spots than those of our *leo* simulations.

### Robustness of the *leo* assumptions in generating spots

As a test of robustness, we perform a rigorous robustness analysis by carrying out one hundred repeats of the mutant simulations with perturbed parameter values chosen uniformly at random from the range 0.75–1.25 of their described value as in Section "Simulation of WT pattern". Ten of these randomly sampled repeats are given in Figure *Appendix 4—figure 1* for *leo*. We observe that for all one hundred repeats that small perturbations to the rates still generate consistent spots, demonstrating the robustness of the model.

These results of this section demonstrate the remarkable ability of our model to generate the *leo* single and double mutant phenotypes under a set of specific proposed changes to the model rules; these proposals can be used to guide experimental exploration of the effects of the *leo* gene.

## Discussion

As a result primarily of beautiful experimental work, zebrafish pigment pattern formation has become the best characterised pigment patterning mechanism (*Frohnhöfer et al., 2013*; *Kelsh et al., 2009*; *Singh and Nüsslein-Volhard, 2015*; *Watanabe and Kondo, 2015*; *Patterson and Parichy, 2019*). Zebrafish pigment pattern formation requires three pigment producing cell types: melanocytes, xanthophores and S-iridophores to generate the WT stripes (*Frohnhöfer et al., 2013*). In an attempt to decipher the mechanisms underlying pattern formation, most previous mathematical models have largely focussed on xanthophores and melanocytes and neglected S-iridophores (*Nakamasu et al., 2009*; *Bullara and De Decker, 2015*; *Volkening and Sandstede, 2015*; *Painter et al., 2015*; *Bloomfield et al., 2011*). However, recent studies have shown that S-iridophores are a crucial component, with S-iridophore transitions between dense and loose driving pattern formation. First, the early provision of dense S-iridophores through the horizontal myoseptum orients stripes (*Frohnhöfer et al., 2013*; *Kelsh et al., 2009*; *Singh and Nüsslein-Volhard, 2015*; *Watanabe and Kondo, 2015*; *Patterson and Parichy, 2019*). Moreover, S-iridophores sequentially pre-determine stripes and interstripes via respective shape changes in these regions (*Frohnhöfer et al., 2013*; *Fadeev et al., 2015*; *Krauss et al., 2014*).

Here, we have taken a bottom up approach to modelling zebrafish pattern formation with the aim of testing whether the experimentally defined set of biological rules for zebrafish pigment pattern formation might be sufficient to explain both the WT and the diversity of mutant pigment patterns. We used an individual based modelling approach incorporating all five cell-types deemed important for pattern formation in zebrafish. We formalised all respective cell-cell interactions

mathematically, with interaction strengths, parametrised, where possible, by the biological literature (see *Supplementary file 5*). For the less well-studied S-iridophore transitions, we analysed key mutant phenotypes to infer biologically realistic rules for these interactions, aiming to generate assumptions that were the simplest for pattern formation changes seen, but no simpler. We proved our models ability to simulate the distinctive pattern features during developmental stages PB through J+ of each of WT and six mutant patterns that had been used to determine the biological rules. We showed that in each case, our model simulations matched qualitatively the pattern development in real fish at the various developmental stages considered. This is consistent with the proposal that our modelling assumptions were sufficient for pattern development in these cases. As a more rigorous test of the model, we then investigated its ability to successfully simulate the distinctive patterns of three further mutants with defective S-iridophore properties, including two mutants that had not been modelled mathematically before. We showed that in each case, our model also correctly replicated patterns that were qualitatively similar to the corresponding mutant fish at various developmental milestones.

We assessed multiple quantitative features of our simulations against measured data from published studies, focusing on spatial distributions of cell-types, stripe width and melanocyte numbers. We found that in each case our simulations were highly reproducible, and quantitatively matched the biological observations. We conclude that our mathematical modelling approach, built upon the biological literature, provides substantial validation of the sufficiency of that set of biological rules in explaining pattern formation in zebrafish development for WT and many other mutant fish. Furthermore our modelling provides support for the plausibility of the deduced rules for S-iridophore packing transitions during pattern development.

Finally, we demonstrated the capability of our model to give valuable insight into the patterning mechanism and to make testable predictions about the biology.

This paper represents the first demonstration, to the best of our knowledge, of a model being used explicitly to test the impact of the *sbr* mutation. The *sbr* gene encodes Tjp1a, a key tight junction protein, and is expressed at much higher levels in S-iridophores in a dense configuration than those of the loose form (*Fadeev et al., 2015*). Furthermore, double mutant and chimaeric studies show that *sbr* acts cell-autonomously within the S-iridophores to control adult pigment pattern formation (*Fadeev et al., 2015*). These authors also show that in *sbr* mutants the transition from dense to loose S-iridophores is delayed, suggesting that this transition somehow depends upon Tjp1a in dense S-iridophores (*Fadeev et al., 2015*). Here, we test the patterning impact of this interpretation, by incorporating delayed S-iridophore state transition into our model, and show that this does indeed result in pattern changes consistent with the *sbr* phenotype. This provides theoretical support for Fadeev and colleagues deductions and deepens the interest in understanding the mechanistic basis for this role for Tjp1a.

Our modelling results demonstrate the applicability of complex models to test hypotheses that are difficult to test experimentally. Previously, it has been hypothesised that S-iridophores contribute to pattern formation by orienting the stripe. In Section "An *in silico* investigation into important mechanisms for controlling pattern formation", we demonstrate that this is true. We show that simply reorientating the interstripe to a vertical position is not enough to produce vertical bars as our simulations exhibit a labyrinthine pattern instead of vertical bars. Careful observations of our simulations indicate that in addition to the initial condition, growth is important for determining the final pattern. We show that moving the initial position of the interstripe away from the centre of the flank, where growth is centred, also disrupts the patterning. We are also able to show that by enlarging the initial domain so that it is the same width but taller in height we show that we can generate more stripes and interstripes than that are usually observed at stage J+. Finally, by initialising a domain that is already populated with cells in a stripe position that we can replicate a different stripe formation mechanism seen in fish *Pomocanthus*. Whilst in zebrafish stripe formation is sequential, starting from an initial interstripe and developing bidirectionally from the middle, *Pomocanthus* develops stripes in-between other stripes. We show that if a striped pattern is fully formed when the fish is still growing (and growth is centred) that this forces new stripes to occur between the old ones, instead of at the periphery.

Our model is the first, to the best of our knowledge, to suggest that S-iridophores play a more important role in melanocyte differentiation than xanthophores. We found that by implementing a stronger signaling capacity of S-iridophores than xanthophores for long range melanocyte

differentiation then we obtained a better qualitative and quantitative match for *shd* and *pfe* mutant patterns. In particular, a stronger signalling success rate for melanocyte differentiation from S-iridophores in the long range appeared to align subsequent stripes in *pfe* and WT, resulting in consistency of pseudo-stripe and stripe width respectively. The comparatively reduced signalling success rate for melanocyte differentiation from xanthophores in the long range reduced the number of pseudo-stripes in *shd* from four, to two at stage J+ which is more consistent with real data. Real *shd* fish typically exhibit fewer stripes (approx two at stage J+) than the four of WT (*Frohnhöfer et al., 2013*; *Figure 4A–B*). We also found that this factor was important to produce melanocyte numbers that quantitatively matched data by Frohnhöfer et al when comparing between mutant and WT (*Figure 10A*) as well as a better stripe width match (*Figure 10B*). Thus, our study further reinforces findings that S-iridophores play an important role in determining stripe and interstripe width (without S-iridophores, X0 interstripe width in *shd* is increased) and not just the widely reported role of stripe alignment.

We have further built upon previous mathematical modelling work, by using our model to make predictions about the functions of the *leo* gene. The gene *leo* encodes Connexin 39.4, which is required in melanocytes and xanthophores but not S-iridophores (*Irion et al., 2014*). Connexins play an important role in cell–cell signalling and communication so it has been postulated that the *leo* gene is involved in the signalling between melanocytes and xanthophores as well as signalling cues to S-iridophores regarding shape-transitions (*Irion et al., 2014*). Previous to our investigation, there had been one study of the individual cell-cell interactions in *leo*. This study by demonstrated that unlike WT melanocytes, *leo* melanocytes are not repelled by xanthophores in the short range (*Yamanaka and Kondo, 2014*). Using our model, we first demonstrated that this cell-cell interaction alone was not enough to reproduce the *leo* mutant pattern. Then, in a systematic approach we deduced four hypotheses about cell–cell interactions that might be affected by *leo*, which, upon implementing in our model, successfully replicated the patterns observed in *leo, leo;pfe, leo;nac, leo;shd* and *leo;sbr*. This work provides testable hypotheses about the effect of *leo* which can now guide future experimental work.

In contrast to most previous mathematical studies of pattern formation, the rules we propose for zebrafish pigment patterns are complex and extensive. For example, successful S-iridophore shape transitions in our model require information from both melanocytes and xanthophores. Many other studies have condensed zebrafish pattern formation to a few simple rules that can often be described by a series of partial differential equations (*Kondo, 2017*; *Painter et al., 2015*; *Nakamasu et al., 2009*), in particular Turing reaction-diffusion-type models posit that combinations of short and long range dynamics between melanocytes and xanthophores generate stripe patterns. Indeed, a lot of the excitement around such models is the ease with which small parameter value changes can sometimes result in diverse patterns, many readily recognisable from nature (*Watanabe and Kondo, 2015*; *Metz et al., 2011*; *Maini et al., 2012*). A major difference between our model and Turing reaction-diffusion models is that small parameter changes in our model do not typically generate qualitatively different patterns, whereas Turing reaction-diffusion models can show substantial pattern changes in response to small alterations to parameters near to bifurcation points (*Watanabe and Kondo, 2015*; *Maini et al., 2012*). We suggest that the added complexity of the real system has evolved to make the patterning process robust, with partially redundant mechanisms insulating against the impact of stochastic variation during pattern formation.

Our modelling approach is analogous to that adopted in another recent study of zebrafish pattern formation, one which independently attempts to understand S-iridophore contributions to the process (*Volkening and Sandstede, 2018*). In a similar fashion, Volkening et al generated an off-lattice model incorporating S-iridophores for which the rules were based upon the experimental literature. Upon implementing these rules they attempted to test the sufficiency of the known biology in describing the patterning process. Importantly, Volkening et al's modelling approach differs from that adopted here in several ways. First, we use an on-lattice model, whilst Volkening et al use an off-lattice model. Using an on-lattice model allowed us to incorporate volume exclusion by other cells directly. Secondly, we used a continuous-time model, whereas Volkening et al update their model at simulated 24 hr intervals, with all rules implemented simultaneously. By using a continuous-time method, we are able to capture the stochasticity involved in rates of reactions and the ordering of events over time. Finally, we incorporate a hypothesis that S-iridophores contribute more strongly to promoting melanocyte differentiation than xanthophore differentiation, leading us to a better

qualitative approximation of *shd* mutants and a better quantitative estimate of melanocyte numbers in *shd*, *pfe* and *rse* than shown before.

Importantly, the model of Volkening et al. also proves highly capable in generating simulations that accurately mimic WT and various mutant pigment patterns. This observation further strengthens support for the validity of the proposed S-iridophore rules we each postulate. More broadly, we consider that our independent mathematical approaches are mutually reinforcing in reaching the conclusion that the deduced biological rules may be largely sufficient to explain pigment pattern formation in zebrafish.

Further testing of our model should focus on investigating later stages of pigment pattern development and maintenance. To date, our focus has been on the crucial dynamic development between PB and J+ stages when the pattern is evolving; we have not considered pattern maintenance between J+ and adulthood. Work by *Frohnhöfer et al., 2013* has demonstrated that in *pfe* mutants, which up to stage J+ have similar numbers of melanocytes to WTs (approximately 90% of WT numbers), this drops to approximately 50% of WT by adulthood. Whilst our model correctly predicts melanocyte numbers in *pfe* compared to WT prior to J+, preliminary simulations up to adulthood suggest this sharp decrease in melanocyte numbers shown in real fish are currently not predicted by our model. This clearly indicates that new biological mechanisms concerning maintenance, likely involving the late differentiating L-iridophores, need to be analysed and incorporated into a model that extends to these later developmental stages.

From an analytical perspective, an advantage of our on-lattice model is its amenability to the derivation of a continuum model, although we note that continuum approximations to off-lattice individual-based models can also be derived. Our model therefore opens up the opportunity for future exploratory work using a continuum model for mutants *pfe* and *nac* in order to explore whether pattern formation in these cases individually can be described as Turing patterns and to determine parameter ranges for successful pattern formation.

It will also be interesting to investigate the role of growth in pattern formation and maintenance. We observed in our simulations a lower average stripe straightness (0.92) than to that of real WT fish (0.98). We further observed in our simulations that increased height-to-length ratio is correlated with stripes becoming more tortuous (*Figure 14C*). Stripes in real fish seem, qualitatively, to not show this effect, and we suspect that our model can be further refined here. A search for mechanisms that might increase stripe straightness will be valuable here, and we note that exploration of our model should allow candidate mechanisms to be identified in silico. Further investigation of this feature in our model once extended to adult pattern maintenance may also be important. For example, casual observation suggests that fish that show a particular height to length bias do not show notably tortuous stripes. Therefore, future work will be to understand what preserves the pattern in these cases. We predict that this may be related to the position of growth. Thus, future work will be to fully evaluate the effects of growth on the final pattern formation.

Another significant aspect which deserves attention concerns dorso-ventral pattern differences. In fish this is characterised by having more pigmented cells at the dorsal region than the ventral region. This is certainly true in adult *nac* (*Figure 4C*), which has disproportionately more pigmented xanthophores in the dorsal than the ventral region). We have recently noted the subtle impact of dorso-ventral countershading on the WT zebrafish pigment pattern, including in the stripes themselves, and have identified Agouti as a key regulator of this process (*Cal et al., 2019*). Furthermore, there are clear dorso-ventral asymmetries in some of the adult mutant patterns: e.g, *nac* mutants exhibit a strong X0 interstripe and a weak ventral interstripe X1D, and completely lack dorsal interstripes. Our model will allow us to explore possible drivers of this asymmetry, which we hypothesise will include Agouti signalling and also differential domain growth.

In conclusion, our on-lattice model, implementing the current biological understanding of adult zebrafish pigment pattern formation, strongly supports the validity of these experimental interpretations, motivating the detailed investigation of their molecular bases. Our model also highlights areas where knowledge is currently incomplete and, importantly, has allowed in silico investigations to identify plausible mechanisms that require experimental testing.

# Additional information

## Funding

| Funder | Grant reference number | Author |
|---|---|---|
| Biotechnology and Biological Sciences Research Council | SWBio DTP | Jennifer P Owen |
| Biotechnology and Biological Sciences Research Council | BB/ L00769X/1 (RNK) | Robert N Kelsh |

The funders had no role in study design, data collection and interpretation, or the decision to submit the work for publication.

## Author contributions

Jennifer P Owen, Conceptualization, Data curation, Formal analysis, Investigation, Methodology; Robert N Kelsh, Resources, Supervision, Investigation, Methodology; Christian A Yates, Supervision, Investigation, Methodology, Project administration

## Author ORCIDs

Jennifer P Owen (ID) https://orcid.org/0000-0001-8440-6822
Christian A Yates (ID) https://orcid.org/0000-0003-0461-7297

## Decision letter and Author response

Decision letter https://doi.org/10.7554/eLife.52998.sa1
Author response https://doi.org/10.7554/eLife.52998.sa2

# Additional files

## Supplementary files

• Supplementary file 1. Summary of all notation used in Appendix 1.

• Supplementary file 2. Summary of all notation used in Appendix 1 regarding short and long range interactions.

• Supplementary file 3. SL measurements and HAA measurements and WT pattern description by stage are given as by *Parichy et al., 2009*. Corresponding dpf for the stages are approximated from the images of *Frohnhöfer et al., 2013*.

• Supplementary file 4. All continuous time events and their corresponding propensities for event attempts at time $t$, denoted $\alpha_i(t)$ for event $i$ in minutes.

• Supplementary file 5. Full description of all the cell–cell interaction given in main text. Melanocytes, xanthophores, loose iridophores, dense iridophores and xanthoblasts are denoted $M$, $X$, $I^l$, $I^d$, $X^b$ respectively. C1, C2 stands for cell 1, cell 2, where $C1 = X$, $X^b$ is the target cell and cell two is the signalling cell with corresponding signal range (R) that is either short (S), up to 0.04 mm, or long (L), 0.12 mm. This signal generates action (A) of cell type C1 of: movement (M), differentiation (D), proliferation (P) or survival (S) of type (T). For action (D) or (S), type (T) is denoted '+' if the resultant action is promotion of action A and '-' if the resultant action is inhibition of action (A). For action (M), type (T) is denoted '+' if the resultant action is attraction towards cell type C1 and '-' if the resultant action is repulsion away from cell type C1. *Melanocytes can also differentiate randomly, independent of any other cell type.

• Supplementary file 6. Parameters implemented in the model.

• Transparent reporting form

## Data availability

All mathematical modelling assumptions/ methods have been provided in the supplementary material. Code relating to this paper have been made available on github, at https://github.com/

JenniferOwen/Zebrafish-stripe-model (copy archived at https://github.com/elifesciences-publications/Zebrafish-stripe-model).

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

## Appendix 1

### Implementing the model

This text provides a full description of how a simulation of a WT fish is implemented. All notation used in appendix 1 is summarised in **Supplementary files 1** and **2**.

### Model overview

We model five different cell types; yellow xanthophores ($X$), unpigmented xanthoblasts ($X^b$), black melanocytes ($M$), silver dense S-iridophores ($I^d$) and blue loose ($I^l$) S-iridophores. We account for the three separate hypodermis layers upon which the cells lie as three separate lattice domains, a xanthophore layer represented by matrix $X$, a melanocyte layer represented by matrix $M$, and an S-iridophore layer represented by a matrix $I$. We denote the length and height of domain $D \in \{X, M, I\}$ at time $t$ in terms of the numbers of lattice sites as $\Pi_D^H(t)$, $\Pi_D^L(t)$ respectively. At any time, $t$, a lattice site at row $i$ and column $j$ in $M$ denoted $M(i,j,t)$ can either be occupied by $M$, i.e. $M(i,j,t) = M$ or be unoccupied, $M(i,j,t) = 0$. Similarly $I(i,j,t)$ can either be occupied by $I^l$, i.e. $I(i,j,t) = I^l$, be occupied by $I^d$, i.e. $I(i,j,t) = I^d$, or be unoccupied, $I(i,j,t) = 0$. Finally, $X(i,j,t)$ can either be occupied by $X$, i.e. $X(i,j,t) = X$, $X^b$, i.e. $X(i,j,t) = X^b$ or be unoccupied, $X(i,j,t) = 0$. This captures volume exclusion rules on each lattice layer which require that at any given time, any lattice site on the domain can be occupied by at most one cell. If at any time during the simulation an action is chosen which would break this rule, this action is aborted. To account for the different packing densities of the cell types, lattice sites on $X$ and $I$ are of size 0.02 mm $\times$ 0.02 mm, whilst lattice sites in $M$ are of size 0.04 mm $\times$ 0.04 mm (**Parichy et al., 2000a**). We denote these as $\Delta_X = \Delta_I = 0.02$ mm and $\Delta_M = 0.04$ mm. Hence the simulated height and length at any given time $t$ is given as $\Omega_H(t) = \Pi_D^L(t)\Delta_D$, $\Omega_L(t) = \Pi_D^H(t)\Delta_D$. An illustration of the three layers and their corresponding cell types are given in **Figure 2A** in the main text. The lattices correspond to matrices;

$$X = \begin{bmatrix} 0 & 0 & X^b & 0 & 0 & X & 0 & 0 & X^b & 0 & X & 0 \\ 0 & 0 & 0 & X & 0 & 0 & 0 & 0 & X & 0 & 0 & 0 \\ 0 & X & 0 & X & 0 & 0 & X^b & 0 & 0 & 0 & 0 & 0 \\ X & 0 & X & 0 & X & X & 0 & X & 0 & X & X & X \\ 0 & 0 & X^b & 0 & X^b & 0 & 0 & 0 & 0 & X & 0 & 0 \\ 0 & X & 0 & 0 & 0 & 0 & X & 0 & 0 & 0 & 0 & 0 \end{bmatrix}$$

$$I = \begin{bmatrix} 0 & 0 & 0 & 0 & 0 & 0 & I^l & 0 & 0 & 0 & 0 & 0 \\ 0 & 0 & 0 & 0 & I^l & 0 & 0 & 0 & I^l & I^l & 0 & 0 \\ I^d & I^d & I^d & I^d & I^d & I^d & I^d & I^d & I^d & I^d & I^d & I^d \\ I^d & I^d & I^d & I^d & I^d & I^d & I^d & I^d & I^d & I^d & I^d & I^d \\ 0 & 0 & I^l & 0 & I^l & I^l & I^l & 0 & 0 & 0 & 0 & 0 \\ 0 & 0 & 0 & 0 & 0 & 0 & 0 & 0 & 0 & 0 & 0 & 0 \end{bmatrix}$$

$$M = \begin{bmatrix} M & 0 & M & M & 0 & 0 \\ 0 & 0 & 0 & M & 0 & 0 \\ 0 & M & 0 & 0 & M & 0 \end{bmatrix}$$

where $\Pi_X^H = \Pi_I^H = 6$, $\Pi_X^L = \Pi_I^L = 12$, $\Pi_M^H = 3$, $\Pi_M^L = 6$.

### Relating size to stage

Zebrafish development is described in the literature with respect to the standard length (SL) of the fish (**Parichy et al., 2009**). We associate our domain at time $t$ to different zebrafish developmental stages by calculating a simulated SL ($\Omega_{SL}$) from the size of the domain at time $t$ and relating this to the development stage using **Supplementary file 3**. Since growth is linear with respect to time (in real life and modelled as so) and we initialise the domain 5.7 mm

shorter than the real width (SL), the corresponding simulated SL for any simulation at time $t$ can be calculated by:

$$\Omega_{SL}(t) = \Delta_D \times \Pi_D^L(t) + 5.7\,\text{mm}, \tag{2}$$

where $D \in \{X, M, I\}$ and $\Pi_D^H(t)$ is the number of sites in the $y$ direction of domain type $D$ at time $t$. The simulated height $\Omega_H(t) = \Delta_D \times \Pi_D^H(t)$ directly corresponds to the height at the anterior margin of the anal fin (abbreviated as HAA) of the fish at all times.

## Initial conditions (WT only)

The simulation is initialised to represent the zebrafish half way through stage PB at approximately 25dpf. At this stage the fish is approximately 1 mm in height (i.e., HAA is 1 mm) and 7.7 mm in length (i.e. SL is 7.7 mm). We model the full height and a subsection of the full length for convenience. We set $\Omega_H(0) = 1mm$ and $\Omega_L(0) = 2mm$. Therefore, $\Pi_M^L(0)$=25, $\Pi_M^H(0)$=50, $\Pi_X^H(0) = \Pi_I^H(0) = 50$ and $\Pi_X^L(0) = \Pi_I^L(0) = 100$.

The initial occupancies for a WT simulation comprise three rows of $I^d$ along the middle of the fish, that is, sites on rows 24 to 26 on $I$ are fully occupied with $I^d$ cells at time $t = 0$. This represents the S-iridophores which differentiate along the horizontal myoseptum between 21 and 25 dpf. We also place $N_M^T(0) = 50$ uniformly at random on $M$ so that the initial occupancy density of melanocytes is

$$\frac{N_M^T(0)}{\Pi_M^L(0) \times \Pi_M^H(0)} = 0.04, \tag{3}$$

where $N_C^T(t)$ for any cell type $C \in \{M, X, X^b, I^d, I^l\}$ denotes the total number of cells of that type on the domain at time $t$. These scattered melanocytes represent the initial larval pattern at low density dispersed across the domain at the beginning of pattern metamorphosis. Similarly, we occupy the domain so that the initial density of xanthoblasts is 0.4. This density is chosen from evidence that larval xanthophores, which de-differentiate around 5 dpf, proliferate and cover much of the entire domain at the start of metamorphosis (*Mahalwar et al., 2014*). We do this by placing $N_{X^b}(0) = 2000$ uniformly at random on $X$ so

$$\frac{N_{X^b}^T(0)}{\Pi_X^L(0) \times \Pi_X^H(0)} = 0.4. \tag{4}$$

This is to represent the de-differentiated larval xanthophores which initially appear around 5 dpf, subsequently lose pigment (becoming xanthoblasts) and proliferate between 5 and 25 dpf. For all other cell types i.e. $C \in \{X, I^l\}$, $N_C^T(0) = 0$.

## Model iteration

The model is updated in continuous time according to the Gillespie algorithm (*Gillespie, 1977*) (see *Figure 2B* in the main text for an illustration). In our simulation we allow two different event types: continuous and fixed. A continuous event is an event that can happen at any point throughout the simulated time. For example; melanocyte birth or death. A fixed event is an event that occurs once during the simulation, and happens upon meeting predetermined conditions. In WT fish, we define only one fixed event. We assume there is appearance of metamorphic xanthophores $X$ along the horizontal myoseptum at stage SP. This is justified by observations of *shd* in which delayed appearance of metamorphic $X$ in interstripe X0 were noted. We found this delayed appearance was important for generating pseudo-stripe patterns in *rse* and *shd* which have reduced or entirely absent S-iridophores. We model this by occupying all the sites, regardless of prior occupancy, along the middle three rows of $X$ with xanthophores ($X$) at stage SP that is sites $X(i, j, t_{SP}) = X$ for all $i \in \lceil \frac{\Pi_X^H(t_{SP})}{2} \rceil - 1 :$ $\lceil \frac{\Pi_X^H(t_{SP})}{2} \rceil + 1$ where $t_{SP}$ denotes the time that the given simulation reaches the start of stage SP.

There are 15 continuous event types. These events are summarised in **Supplementary file 4**. They comprise cell birth, death, movement and shape transitions as well as growth of the domain.

An overview of how the model is updated is given in **Figure 2B** in the main text and can be described as follows. At any given time $t$, the model is first assessed for meeting the criteria of a fixed event. If the model meets the criteria, the fixed event occurs, is subsequently marked as complete and the simulation continues. If no fixed time event is to be implemented then one of the fifteen possible continuous time events is attempted. First an exponentially distributed waiting time

$$\tau = \frac{1}{\alpha_0} \log\left(\frac{1}{u_0}\right), \tag{5}$$

is generated until the next continuous 'event' occurs where $\alpha_0 = \sum_{i=1}^{15} \alpha_i(t)$ is the sum of all of the event propensities given in **Supplementary file 4** and $u_0$ is a uniformly distributed random number in $(0,1)$ (i.e. $u_0 \sim U(0,1)$). Next, we generate $u_1 \sim U(0,1)$ to determine which event occurs at time $t + \tau$. The event $i$ that satisfies

$$\sum_{j=1}^{i-1} \frac{\alpha_j(t)}{\alpha_0} \leq u_1 < \sum_{j=1}^{i} \frac{\alpha_j(t)}{\alpha_0} \tag{6}$$

is chosen to occur and the domain is updated accordingly (provided conditions required for that event to occur are met). Time is also updated. $t = t + \tau$. We continue this process iteratively, checking for fixed events, then subsequently generating a time for the next continuous event to occur. The process repeats until we reach the end of pattern metamorphosis, marked by length condition $\Omega_{SL} = 13.5mm$. The algorithm is stochastic in the sense that, within any given simulation, there is variance with regard to the exact rate and order of event occurrence, just as in real fish.

## Modeling cell–cell interactions

Continuous events 1–15 given in **Supplementary file 4** are mediated by different cell-cell interactions. Cells interact on the fish skin at both short (neighbouring cells) and long (half a stripe width) range, possibly regulated by direct contact, dendrites, or longer extensions (filopodia or airinemes). In our model, uniform disks, with radii on the order of the distance 0.04 mm account for short-range interactions, and an annulus with an outer radius of approximately half an adult stripe width 0.24mm represent long-range dynamics.

We denote $S_D(r,k)$ and $L_D(r,k)$ where $D \in \{X, I, M\}$ is the set of site positions $(i,j)$ such that $D(i,j)$ is in the short (0.04 mm) or long (0.24 mm) distance respectively from a focal site at position $D(r,k)$. Note that $S_D(r,k)$ and $L_D(r,k)$ are both different for different domain types due to the different lattice site sizes. $S_D(r,k)$ and $L_D(r,k)$ are visualised for $D = M, X$ in **Figure 3A–H** in the main text. Formulae for these sets are given in **Supplementary file 2**.

To illustrate this see **Figure 3A–H** in the main text which compares the number of cells in the short (**Figure 3A–D**) and long (**Figure 3E–H**) range distance from a central site, $D(r,k)$, on different domain types. In this figure, $M$ are represented as black circles. $X$ are represented as yellow circles. **Figure 2A** is a visualisation of sites (marked in red) in $S_M(r,k)$, where $M(r,k)$ is the central site marked in grey. $S_M^M(r,k,t)$ are the sites marked in red in which a melanocyte resides. The number of melanocytes in the short range distance from $M(r,k)$ at time $t$ is given by $N_{M,M}^S(r,k,t) = S_M^M(r,k,t) = 2$. **Figure 2B** is a visualisation of the sites considered when calculating $N_{M,X}^S$ (formula given in **Supplementary file 2**). In this example, $N_{M,X}^S(r,k,t) = 9$. To compare the number of $M$ and $X$ in the short range distance of $M(r,k)$, we consider $wN_{M,M}^S(r,k,t) = 4 \times 2 = 8$ which corresponds to the weighted value of melanocytes in the short range distance with $N_{M,X}^S(r,k,t) = 9$ the number of xanthophores in the short range. **Figure 3C** is a visualisation of sites (marked in red) in $S_X(r,k)$, where $X(r,k)$ is the central site marked in grey. $S_X^X(r,k,t)$ are the sites marked in red within which a xanthophore resides. The number of

xanthophores in the short range distance from $X(r,k)$ at time $t$ is given by $N^S_{X,X}(r,k,t) = S^X_X(r,k,t) = 7$. **Figure 2D** is a visualisation of the sites considered when calculating $N^S_{X,M}(r,k,t)$ (formula given in **Supplementary file 2**). In this example, $N^S_{X,M}(r,k,t) = 14$. To compare the number of $M$ and $X$ in the short range distance of $X(r,k)$, at time $t$, we would compare $N^S_{X,M}(r,k,t)=14$ with $N^S_{X,X}(r,k,t) = 7$. **Figure 3E** is a visualisation of sites (marked in red) in $L_M(r,k)$, where $M(r,k)$ is the central site marked in grey. $L^M_M(r,k,t)$ are the sites marked in red in which a melanocyte resides. The number of melanocytes in the long range distance from $M(r,k)$ is given by $N^L_{M,M}(r,k,t) = L^M_M(r,k,t) = 20$. **Figure 2F** is a visualisation of the sites considered when calculating $N^L_{M,X}(r,k,t)$ (formula given in **Supplementary file 2**). In this example, $N^L_{M,X}(r,k,t) = 16$. To compare the number of $M$ and $X$ in the long range distance of $M(r,k)$, we consider $wN^L_{M,M}(r,k,t) = 4 \times 20 = 80$ which corresponds to the weighted value of melanocytes in the long range distance with $N^S_{M,X}(r,k,t) = 16$, the number of xanthophores in the long range. **Figure 2G** is a visualisation of sites (marked in red) in $L_X(r,k)$, where $X(r,k)$ is the central site marked in grey. $L^X_X(r,k)$ are the sites marked in red in which a xanthophore resides. The number of xanthophores in the long range distance from $X(r,k)$ is given by $N^L_{X,X} = L^X_X(r,k,t) = 2$. **Figure 3H** is a visualisation of the sites considered when calculating $N^L_{X,M}(r,k,t)$ (formula given in **Supplementary file 2**). In this example $N^L_{X,M}(r,k,t) = 20$. To compare the number of $M$ and $X$ in the long range distance of $X(r,k)$, we consider $N^L_{X,M}(r,k,t) = 20$ with $N^L_{X,X}(r,k,t) = 2$.

We denote $S^C_D(r,k,t) \subset S_D(r,k)$ where $C$ lies in layer $D$ to be the sites in $S_D(r,k)$ occupied by cell type $C \in \{M, X, I^d, I^l, X^b\}$ at time $t$. We denote $N^S_{C_i,C_j}(r,k,t)$, $(N^L_{C_i,C_j}(r,k,t))$ as the number of cells of type $C_j$ in the short (long) range distance 0.04 mm, (0.24 mm) from cell type $C_i$ at $D_i(r,k)$. Hence the number of cells of type $C_i$ in the short distance from another cell of the same type at $D(r,k)$ at time $t$ is given by $N^S_{C_i,C_i}(r,k,t) = |S^{C_i}_D(r,k,t)|$. The formulae for $N^S_{C_i,C_j}(r,k,t)$, $N^L_{C_i,C_j}(r,k,t)$ where $C_i$ does not equal $C_j$ are more complicated and are given in **Equations 7, 8** (and **Supplementary file 2** for reference).

$$N^S_{C_1,C_2}(r,k,t) = \begin{cases} S^{C_2}_{D_2}(r,k,t) \text{ if } D_1 = D_2 \text{ or } D_1 \text{ and } D_2 \in \{X,I\}. \\ \sum_{(i,j)\in S_{D_1}(r,k)} 1_{M(\lceil \frac{i}{2}\rceil, \lceil \frac{j}{2}\rceil)=M} \text{ if } D_1 \neq M, D_2 = M. \\ \sum_{(i,j)\in S_M(r,k)} 1_{D_2(2i,2j)=C_2} + 1_{D_2(2i-1,2j)=C_2} + 1_{D_2(2i,2j-1)=C_2} + \ldots \\ 1_{D_2(2i-1,2j-1)=C_2} \text{ if } D_1 = M, D_2 \neq M. \end{cases}$$ (7)

$$N^L_{C_1,C_2}(r,k,t) = \begin{cases} L^{C_2}_{D_2}(r,k,t) \text{ if } D_1 = D_2 \text{ or } D_1 \text{ and } D_2 \in \{X,I\}. \\ \sum_{(i,j)\in L_{D_1}(r,k)} 1_{M(\lceil \frac{i}{2}\rceil, \lceil \frac{j}{2}\rceil)=M} \text{ if } D_1 \neq M, D_2 = M. \\ \sum_{(i,j)\in L_M(r,k)} 1_{D_2(2i,2j)=C_2} + 1_{D_2(2i-1,2j)=C_2} + 1_{D_2(2i,2j-1)=C_2} + \ldots \\ 1_{D_2(2i-1,2j-1)=C_2} \text{ if } D_1 = M, D_2 \neq M. \end{cases}$$ (8)

Simply, where domain types $D$ do not have the same lattice site size $\Delta_D$, the focal site coordinates $(i,j)$ of the local neighbourhood undergo a transformation for the sites on a domain with a different size. For example, each site $X(i,j)$, corresponds to a quarter of site $M(\lceil \frac{i}{2}\rceil, \lceil \frac{j}{2}\rceil)$, this explains case two in **Equation 7**. Similarly each site $M(i,j)$ corresponds to four sites on $X$, specifically $X(2i,2j)$, $X(2i,2j-1)$, $X(2i-1,2j)$ and $X(2i-1,2j-1)$. This explains case three in **Equations 7 and 8**. Note that since $M$ is four times larger than all other cells in our simulation, we provide a weighting system when comparing $M$ with $CM$ in some cases.

## Boundary conditions

Boundary conditions are periodic along the horizontal boundaries and reflecting across the vertical boundaries. We implement periodic boundary conditions along the horizontal axis

based on the assumption that the rate at which cells leave along this axis is approximately equal to the rate at which cells enter the domain at the opposite side.

## Continuous time events

In this text, we will describe how each of the fifteen events are simulated upon being selected to occur by the Gillespie algorithm. An overview of all continuous time events and their corresponding rates is given in *Supplementary file 4*.

### Movement (continuous time events 1–5)

We implement cell movement so that cells are biased towards cell types they are attracted to and away from cell types they are repelled by. The direction of the cell's movement is determined using an on-lattice attraction-repulsion mechanism based on a model described by *Dini et al., 2018* and is detailed as follows. If a cell is chosen to move, it is able to move in one of eight different possible orientations denoted by $O \in \boldsymbol{O} = \{No, So, Ea, We, NE, SE, SW, NW\}$. (See *Figure 3(D)* in the main text for an illustration of these directions).

The directional neighbours of each cell is given in *Figure 2E,F*. *Figure 2E* demonstrates the directional neighbourhoods of a focal cell $C \in \{X, X^b, I^d, I^l\}$ located in position $D(i,j)$ (marked in grey) where $D \in \{\boldsymbol{X}, \boldsymbol{I}\}$. For the cell moving one space on $D$, the cell considers the occupancy of sites in $S_{k_{\Delta_X}}^D(i,j)$ marked in red for $D = \boldsymbol{M}$ (top) and $D \in \{\boldsymbol{X}, \boldsymbol{I}\}$ (bottom). (Q–X) *Figure 2F* demonstrates the directional neighbourhoods of a focal cell $M$ located in position $M(i,j)$ (marked in grey). For the cell $M$ moving one space on $M$, the cell considers the occupancy of sites in $S_{k_{\Delta_M}}^D(i,j)$ marked in red for $D \in \{\boldsymbol{X}, \boldsymbol{I}\}$ (top) and $D = \boldsymbol{M}$ (bottom).

We define the probability that a cell attempts to move in a given direction with orientation $O$ as $P_O$ (where $\sum_{O \in \boldsymbol{O}} P_O = 1$). We calculate $P_O$, using bias vector,

$$\underline{v} = A\underline{a} + R\underline{r}, \tag{9}$$

where $A$ ($R$) is a matrix whose entries are the number of neighbouring cells within different segments of the attraction (repulsion) range and $\underline{a}$ ($\underline{r}$) is a weight of attraction (repulsion) vector.

Matrices $A$ and $R$ are defined as follows.

$$A = \begin{bmatrix} Ea_{C,C_1}(r,k,t) & Ea_{C,C_2}(r,k,t) & Ea_{C,C_3}(r,k,t) \\ We_{C,C_1}(r,k,t) & We_{C,C_2}(r,k,t) & W_{C,C_3}(r,k,t) \\ No_{C,C_1}(r,k,t) & No_{C,C_2}(r,k,t) & No_{C,C_3}(r,k,t) \\ So_{C,C_1}(r,k,t) & So_{C,C_2}(r,k,t) & So_{C,C_3}(r,k,t) \\ NW_{C,C_1}(r,k,t) & NW_{C,C_2}(r,k,t) & NW_{C,C_3}(r,k,t) \\ NE_{C,C_1}(r,k,t) & NE_{C,C_2}(r,k,t) & NE_{C,C_3}(r,k,t) \\ SW_{C,C_1}(r,k,t) & SW_{C,C_2}(r,k,t) & NE_{C,C_3}(r,k,t) \\ SE_{C,C_1}(r,k,t) & SE_{C,C_2}(r,k,t) & SE_{C,C_3}(r,k,t) \end{bmatrix}, \tag{10}$$

$$R = \begin{bmatrix} We_{C,C_1}(r,k,t) & We_{C,C_2}(r,k,t) & We_{C,C_3}(r,k,t) \\ Ea_{C,C_1}(r,k,t) & Ea_{C,C_2}(r,k,t) & Ea_{C,C_3}(r,k,t) \\ So_{C,C_1}(r,k,t) & So_{C,C_2}(r,k,t) & So_{C,C_3}(r,k,t) \\ No_{C,C_1}(r,k,t) & No_{C,C_2}(r,k,t) & No_{C,C_3}(r,k,t) \\ NE_{C,C_1}(r,k,t) & NE_{C,C_2}(r,k,t) & NE_{C,C_3}(r,k,t) \\ NW_{C,C_1}(r,k,t) & NW_{C,C_2}(r,k,t) & NW_{C,C_3}(r,k,t) \\ SE_{C,C_1}(r,k,t) & SE_{C,C_2}(r,k,t) & SE_{C,C_3}(r,k,t) \\ SW_{C,C_1}(r,k,t) & SW_{C,C_2}(r,k,t) & NE_{C,C_3}(r,k,t) \end{bmatrix} \tag{11}$$

where for example, $O_{C,C_1}(r,k,t)$ denotes the number of cells of type $C_1 \in \{X, M, I^d, I^l, X^b\}$ within

range 0.04 mm, that are orientation $O$ from the focal cell $C$ located in position $D(r,k)$ at time $t$. This number is calculated differently depending on whether the focal site lies on lattice domain $D \in \{M, X, I\}$. To compute $O_{C,C_1}(r,k,t)$ we calculate the following. For any $O$ we define subsets $S_D^O(r,k,t)$ of $S_D(r,k,t)$ (where cell type $C$ lies on domain type $D$) for each direction $O$;

$$S_D^{No}(r,k,t) = \{(i,j) \in S_D(r,k,t) \,|\, j > k\}, \tag{12}$$

$$S_D^{So}(r,k,t) = \{(i,j) \in S_D(r,k,t) \,|\, j < k\}, \tag{13}$$

$$S_D^E(r,k,t) = \{(i,j) \in S_D(r,k,t) \,|\, (|r-i| < \Pi_D^L(t) - |r-i| \text{ and } i > r) \text{ or } (|r-i| > \Pi_D^L(t) - |r-i| \text{ and } i < r)\}, \tag{14}$$

$$S_D^W(r,k,t) = \{(i,j) \in S_D(r,k,t) \,|\, (|r-i| < \Pi_D^L(t) - |r-i| \text{ and } i < r) \text{ or } (|r-i| > \Pi_D^L(t) - |r-i| \text{ and } i > r)\}. \tag{15}$$

These are the sets of sites in the short range distance, north, south, east and west of focal site $D(r,k)$. Examples for $D = M$ and $X$ are sites marked in red on the left of **Figure 2E–F** respectively. By extension we define

$$S_D^{NE}(r,k,t) = \{(i,j) \in S_D(r,k,t) \,|\, j \geq k \text{ and } (|r-i| < \Pi_D^L(t) - |r-i| \text{ and } i \geq r) \tag{16}$$

$$\text{or} (|r-i| > \Pi_D^L(t) - |r-i| \text{ and } i \leq r)\}, \tag{17}$$

$$S_D^{NW}(r,k,t) = \{(i,j) \in S_D(r,k,t) \,|\, j \geq k \text{ and } (|r-i| < \Pi_D^L(t) - |r-i| \text{ and } i \leq r) \tag{18}$$

$$\text{or} (|r-i| > \Pi_D^L(t) - |r-i| \text{ and } i \geq r)\}, \tag{19}$$

$$S_D^{SE}(r,k,t) = \{(i,j) \in S_D(r,k,t) \,|\, j \leq k \text{ and } (|r-i| < \Pi_D^L(t) - |r-i| \text{ and } i \geq r) \tag{20}$$

$$\text{or} (|r-i| > \Pi_D^L(t) - |r-i| \text{ and } i \leq r)\}, \tag{21}$$

$$S_D^{SW}(r,k,t) = \{(i,j) \in S_D(r,k,t) \,|\, j \leq k \text{ and } (|r-i| < \Pi_D^L(t) - |r-i| \text{ and } i \leq r) \tag{22}$$

$$\text{or} (|r-i| > \Pi_D^L(t) - |r-i| \text{ and } i \geq r)\}. \tag{23}$$

These are the sets of sites in the short range distance, north-east, north-west, south-east and south-west of focal site $D(r,k)$. We denote $S_D^{C,O}(r,k,t) \subseteq S_D^O(r,k,t)$ to be all the sites in short-range distance in orientation $O \in \{No, So, Ea, We, NE, SE, SW, NW\}$ occupied by cell type $C$. Hence we define

$$S_D^{C,O}(r,k,t) = \{(i,j) \in S_D^O(r,k,t) \,|\, D(i,j) = C\}. \tag{24}$$

Therefore to compute the number of cells in each of the eight directions, we compute where $C$ lies on $D$ and $C_1$ lies on $D_1$ by;

$$O_{C,C_1}(r,k,t) = \begin{cases} |S_{D_1}^{C,O}(r,k,t)| \text{ where } D = D_1 \text{ or } D, D_1 \in \{X, I\}. \\ \displaystyle\sum_{S_{D_1}^O(r,k)} \mathbb{1}_{M(\lceil \frac{r}{2} \rceil, \lceil \frac{k}{2} \rceil) = M} \text{ where } D = M, D_1 \neq M. \\ \displaystyle\sum_{S_M^O(r,k)} \mathbb{1}_{D(2r,2k) = C} + \mathbb{1}_{D(2r-1,2k) = C} \\ + \mathbb{1}_{D(2r,2k-1) = C} + \mathbb{1}_{D(2r-1,2k-1) = C} \text{ where } D \neq M, D_1 = M. \end{cases} \tag{25}$$

Next we define the weight of the attraction-repulsion vector by:

$$\underline{a} = \begin{bmatrix} a_{CC_1} \\ a_{CC_2} \\ a_{CC_3} \end{bmatrix}, \underline{r} = \begin{bmatrix} \rho_{CC_1} \\ \rho_{CC_2} \\ \rho_{CC_3} \end{bmatrix} \tag{26}$$

where $a_{CC_1}, a_{CC_2}, a_{CC_3}$ ($\rho_{CC_1}, \rho_{CC_2}, \rho_{CC_3}$) are the weights of attraction (repulsion) of cells of type $C$ to cells of the type $C_1$, $C_2$ and $C_3$, respectively defined in **Supplementary file 5**. Vectors $\underline{a}$ and $\underline{b}$ are the vectors used to calculate $\underline{v}$ in **Equation 9**. To account for the difference in distance between adjacent neighbours and diagonal neighbours we normalise accordingly so that movement diagonally is proportionately less frequent than moving to adjacent squares.

$$\hat{P}_W = \frac{2v_1}{3\sum_{q=1}^{4} v_q}, \tag{27}$$

$$\hat{P}_E = \frac{2v_2}{3\sum_{q=1}^{4} v_q}, \tag{28}$$

$$\hat{P}_{So} = \frac{2v_3}{3\sum_{q=1}^{4} v_q}, \tag{29}$$

$$\hat{P}_{No} = \frac{2v_4}{3\sum_{q=1}^{4} v_q}, \tag{30}$$

$$\hat{P}_{NE} = \frac{v_5}{3\sum_{q=5}^{8} v_q}, \tag{31}$$

$$\hat{P}_{NW} = \frac{v_6}{3\sum_{q=5}^{8} v_q}, \tag{32}$$

$$\hat{P}_{SE} = \frac{v_7}{3\sum_{q=5}^{8} v_q}, \tag{33}$$

$$\hat{P}_{SW} = \frac{v_8}{3\sum_{q=5}^{8} v_q}. \tag{34}$$

Finally we normalise $P_O$ s.t.;

$$P_O = \frac{\hat{P}_O}{\sum_{O \in \mathbf{O}} \hat{P}_O}. \tag{35}$$

To summarise, we simulate movement as follows. Given the event is chosen such that a cell of type $C$ is allocated to move. We choose a random cell of type $C$. We call its position $D(r,k)$. Next values $P_O$ are calculated using **Equation 27–33** and **Equation 35**. A random number $u \sim U(0,1)$ is generated. The cell attempts movement to $D(r,k-1)$ (West) if $u \in [0, P_W)$, $D(r,k+1)$ (East) if $u \in [P_W, P_W + P_E)$ etc. up to $[\sum_{O \in \mathbf{O}} P_O - P_{SW}, \sum_{O \in \mathbf{O}} P_O]$. The cell movement is successful and the cell moves from site $D(r,k)$ to the nearest neighbouring site in direction $O$, provided this site is empty. Otherwise the movement is aborted. Movement bias is specified by weights of attraction and repulsion ($\underline{a}$ and $\underline{r}$). Relevant parameters for $a_{C_i C_j}$ and $\rho_{C_i C_j}$ for different $C_i$ and $C_j$ are given in **Supplementary file 5** . If no such interaction bias is specified then there is no known attraction or repulsion dynamics between those cell types in the literature and so we assume that movement is not biased by this cell type. Note that for cell $X^b$ there are no known short range interactions between $X^b$ and other cell types so we model $X^b$ movement as random, i.e, $P_O = \frac{1}{8}$, for $O \in \mathbf{O}$ at all times.

S-iridophore, xanthophore and xanthoblast proliferation (continuous time events 6–7)

Given the event is chosen such that a cell of type $C \in \{X, X^b, I^d, I^l\}$ is determined to proliferate we choose a random cell of type $C$ whose position is given by $D(r, k)$, $D \in \{\boldsymbol{X}, \boldsymbol{I}\}$. Next a random number $u \sim U(0, 1)$ is generated. This number is used to determine a neighbouring site into which a mother cell can place a daughter cell. This site is $D(r, k + 1)$ if $u \in [0, 1/4)$, $D(r + 1, k)$ if $u \in [1/4, 1/2)$, $D(r - 1, k)$ if $u \in [1/2, 3/4)$ or $D(r, k - 1)$ if $u \in [3/4, 1]$.

For cell types $C \in \{I^d, I^l, X^b\}$, a proliferation event is successful if the site chosen is empty, and a new cell of the same type is placed into the chosen site, otherwise the event is aborted. In the case $C = X$, a proliferation event is successful if the site chosen for the daughter cell is unoccupied and

$$N_{X,X}^S(r,k,t) + N_{X,I^d}^S(r,k,t) > N_{X,I^l}^S(r,k,t) + N_{X,M}^S(r,k,t). \tag{36}$$

*Equation 36* is based on the following assumptions: (1) Dense S-iridophores promote xanthoblasts differentiation into xanthophores in the short range (*Patterson and Parichy, 2013*). Dense S-iridophores express xanthogenic Colony Stimulating Factor-1 (Csf1) (*Patterson and Parichy, 2013*) which is essential for xanthophore differentiation, proliferation and survival, allowing unpigmented xanthoblasts near to the dense S-iridophores to mature into xanthophores (*Frohnhöfer et al., 2013*; *Walderich et al., 2016*); (2) melanocytes inhibit xanthophore specification in the short range (*Nakamasu et al., 2009*).

## Melanocyte differentiation (continuous time event 8)

If a melanocyte differentiation event is specified then a site $\boldsymbol{M}(r, k)$ is chosen at random. A differentiation event is successful and a new $M$ appears in this site if the site $\boldsymbol{M}(r, k)$ is empty and the following is true:

$$N_{M,X}^L(r,k,t) + N_{M,I^d}^L(r,k,t) > w\alpha N_{M,M}^L(r,k,t) + \beta \tag{37}$$

$$\text{and } N_{M,X}^S(r,k,t) \leq \gamma w N_{M,M}^S(r,k,t), \tag{38}$$

$$\text{and } N_{M,I^d}^L(r,k,t) \leq \kappa, \tag{39}$$

where $\alpha = 2.5$, $\beta = \gamma = \kappa = 3$, $w = 4$. *Equation 37* is based on the findings that dense S-iridophores and xanthophores promote the differentiation of melanocytes in the long range. This conclusion is drawn from observations in ablation experiments (*Nakamasu et al., 2009*) and in the *pfe* mutant, which retains a high number of $M$(*Frohnhöfer et al., 2013*). *Equation 38* is based on observations that melanocytes and xanthophores compete in the short range (*Nakamasu et al., 2009*). *Equation 39* is based on observations that in WT fish, melanocyte centers rarely overlap with dense S-iridophores; however, melanocytes frequently settle adjacent to dense S-iridophores, suggesting short range inhibition (*Patterson and Parichy, 2013*).

We assume there is some melanocyte differentiation into empty space that is independent of cues from other cells. This is from observations of double mutant fish *nac;pfe* that do not produce S-iridophores or xanthophores but phenotypically display uniformly distributed melanocytes at adulthood (*Frohnhöfer et al., 2013*). Therefore, alternatively we also allow successful $M$ differentiation if we generate a random number $u \in U(0, 1)$ and we find that $u < 0.01$ in combination with the condition

$$N_{M,X}^S(r,k,t) + N_{M,I^d}^S(r,k,t) + N_{M,M}^S(r,k,t) = 0. \tag{40}$$

Within each attempt, criterion (*Equation 37–39*) is tried first. If this is not successful, criteria (*Equation 40*) is tested instead.

## Xanthoblast differentiation (continuous time event 9)

If a xanthoblast differentiation event is chosen then an $X^b$ is chosen at random from $X$. Suppose the chosen $X^b$ lies in site $X(r,k)$. The differentiation event is successful and a $X$ replaces the $X^b$ in this site if the site $M(\lceil \frac{r}{2} \rceil, \lceil \frac{k}{2} \rceil)$ is not occupied by a cell $M$ (as melanocytes and xanthophores are known to compete in the short range [**Nakamasu et al., 2009**]) and if the following is true.

$$N_{X,X}^S(r,k,t) + N_{X,I^d}^S(r,k,t) > N_{X,M}^S(r,k,t) + N_{X,I^l}^S(r,k,t). \tag{41}$$

**Equation 41** is based on the following assumptions; (1) Dense S-iridophores promote xanthoblasts differentiation into xanthophores in the short range (**Patterson and Parichy, 2013**). Dense S-iridophores express Csf1, allowing unpigmented xanthoblasts near to the dense S-iridophores to mature into xanthophores (**Frohnhöfer et al., 2013**; **Walderich et al., 2016**). (2) melanocytes inhibit xanthophore specification in the short range (**Nakamasu et al., 2009**).

Alternatively the differentiation event is successful if:

$$N_{X,X}^S(r,k,t) + N_{X,M}^S(r,k,t) = 0, \tag{42}$$

holds and a randomly generated number $u \in U(0,1)$ is such that $u < 0.01$, or **Equation(42)** holds and either

$$t > 31 \, \text{dpf and} \, \frac{N_M^T(t) + N_X^T(t) + N_{I^d}^T(t)}{\Pi_X^L(t) \times \Pi_X^H(t)} < 0.2, \tag{43}$$

$$t > 51 \, \text{dpf and} \, \frac{N_M^T(t) + N_X^T(t) + N_{I^d}^T(t)}{\Pi_X^L(t) \times \Pi_X^H(t)} < 0.4, \tag{44}$$

is true i.e, the total cell density with all cells combined is less than either 0.2 or 0.4 for different time milestones. We enforce a lower probability ($u < 0.01$) for the alternative event (**Equation 42**) since this differentiation event is not influenced by cues from other cells and thus is less likely. **Equation 43** and **Equation 44** are based on assumptions that if the domain is empty after some key time points, this promotes delayed differentiation of xanthoblasts as is observed in *nac* (**Frohnhöfer et al., 2013**). Since melanocytes compete in the short range with xanthophores (**Nakamasu et al., 2009**) we only allow this to occur when constraints described by **Equation(42)** are simultaneously held, for consistency.

## Iridophore transitions (continuous time event 10–11)

If an $I^d$ to $I^l$ shape transition is chosen, then an $I^d$ is chosen uniformly at random from domain $I$ and evaluated for meeting transition criteria. The transition is successful and the chosen $I^d$ in position $I(r,k)$ is replaced with $I^l$ in this site if either both

$$N_{I^d,X}^L(r,k,t) > L_x \text{ and } N_{I^d,X}^S(r,k,t) < S_x, \tag{45}$$

or alternatively

$$N_{M,M}^S(r,k,t) \geq S_m, \tag{46}$$

where $L_x, S_x, S_m = 12, 9$, one respectively. Alternatively if an $I^l$ to $I^d$ shape transition is chosen, then an $I^l$ is chosen at random from domain $I$ and evaluated for meeting transition criteria. This transition is successful and the chosen $I^l$ in position $I(r,k)$ is replaced with $I^d$ in this site if either

$$N_{I^l,X}^L(r,k,t) < L_{x2} \text{ and } N_{I^l,M}^S(r,k,t) < S_{m2}, \tag{47}$$

or alternatively

$$N^S_{I^l,X}(r,k,t) > S_{x2} \text{ and } N^S_{I^l,M}(r,k,t) < S_{m2}, \tag{48}$$

where $L_{x2} = 16$, $S_{x2} = 4$ and $S_{m2} = 1$. These conditions are based on observations of the induction set (*Frohnhöfer et al., 2013*) (for more details see Section "Modelling assumptions" in the main text). The parameters $L_x$, $S_x$, $S_m$, $L_{x2}$, $S_{m2}$, $S_{x2}$ were chosen to give straight stripes with few breaks at stage J+ in WT simulations. However, with some small variations to these parameters, the patterns generated are qualitatively similar.

## Melanocyte death (continuous time event 12)

If a melanocyte death event is chosen, then a cell of type $M$ is chosen at random from $M$. Suppose the chosen $M$ lies in site $M(r,k)$. The death event is successful and the melanocyte is removed from site $M(r,k)$ i.e. $M(r,k)$ is set to 0, if

$$w N^S_{M,M}(r,k,t) < N^S_{M,X}(r,k,t). \tag{49}$$

**Equation 49** is based on findings that xanthophores and melanocytes compete in the short range (*Nakamasu et al., 2009*). Alternatively, the melanocyte death is successful if a randomly generated number $u \in U(0,1)$, is such that $u < 0.001$ and both of the following two cases hold:

$$\sigma N^L_{M,X}(r,k,t) \leq N^L_{M,M}(r,k,t), \tag{50}$$

$$N^S_{M,I^l}(r,k,t) < \omega, \tag{51}$$

where $\sigma = \omega = 3$. **Equation 50** is based on findings that $M$ appear to inhibit the survival of $M$ in the long range and that $X$ promote the survival of $M$ in the long range (*Takahashi and Kondo, 2008*). In **Equation 51**, $I^l$ are a proxy for L-iridophores, which promote the survival of $M$ in the short range (*Frohnhöfer et al., 2013*). We found this equation was important for maintaining melanocytes in mutant *pfe* in our simulations. Moreover, the two equations **Equation 50** and **Equation 51** combined were important for maintaining melanocytes in the interstripes in *pfe* (as seen in real fish) in our simulations. We found it was important that this alternative event had a low probability of success (0.001) but not zero. In *pfe* there are no xanthophores so **Equation 50** enforces melanocyte death where the number of melanocytes in the long range was greater than zero. Therefore, when there was too high a probability of success, in *pfe* simulations the stripe, interstripe structure that is usually maintained between WT and *pfe* was broken into randomly dispersed spots of melanocytes distance 0.24 mm apart. On the other hand, when it was too low, we did not observe melanocytes in the interstripe in our *pfe* simulations, unlike in real *pfe*.

## Xanthoblast pulling event (continuous time event 13)

Suppose a 'xanthoblast pull melanocyte' event is chosen at time $t$, then we choose an $X^b$ uniformly at random from all $X^b$ occupying $X$ that meets the following criterion: suppose the chosen $X^b$ resides in site $X(r,k)$, then the corresponding site on $M$, $M(\lceil \frac{r}{2} \rceil, \lceil \frac{k}{2} \rceil)$ must be empty. We simulate this by randomly choosing $X^b$ on $X$, and checking whether $M(\lceil \frac{r}{2} \rceil, \lceil \frac{k}{2} \rceil)$ is empty. If so, we continue to the next stage. Otherwise we repeat through all $X^b$ on $X$ until we either find a suitable $X^b$. If no $X^b$ satisfy this criteria at time $t$ then we abort the cell-cell pulling event. Given we find a suitable $X^b$, the chosen $X^b$ will attempt to attach and pull a melanocyte within range (airinemes extend to a length of up to 5–6 cell diameters away or 0.1–0.12 mm [*Eom et al., 2015*]) to position $M(\lceil \frac{r}{2} \rceil, \lceil \frac{k}{2} \rceil)$. We simulate this as follows. First, we generate a random position $(i,j)$ uniformly at random from the 28 possible lattice positions euclidean distance 0.1 mm from the site $X(r,k)$ given by $P$. These sites are shown in **Figure 1A–B** and their coordinates are given by;

$$P = \{(r+5,k),(r-5,k),(r,k+5),(r,k-5) \tag{52}$$
$$(r+1,k+5),(r-1,k+5),(r+1,k-5),(r-1,k-5), \tag{53}$$
$$(r+2,k+5),(r-2,k+5),(r+2,k-5),(r-2,k-5), \tag{54}$$
$$(r-3,k-4),(r+3,k+4),(r-3,k+4),(r+3,k-4), \tag{55}$$
$$(r-4,k-3),(r+4,k+3),(r-4,k+3),(r+4,k-3), \tag{56}$$
$$(r+5,k+1),(r-5,k+1),(r+5,k-1),(r-5,k-1), \tag{57}$$
$$(r+5,k+2),(r-5,k+2),(r+5,k-2),(r-5,k-2)\}. \tag{58}$$

Next we translate position $(i,j)$ on $X$ to its corresponding position on $M$. If $M(\lceil\frac{i}{2}\rceil,\lceil\frac{j}{2}\rceil,t)=M$, then the melanocyte is pulled from its site $M(\lceil\frac{i}{2}\rceil,\lceil\frac{j}{2}\rceil)$ into $M(\lceil\frac{r}{2}\rceil,\lceil\frac{k}{2}\rceil)$ i.e. $M(\lceil\frac{i}{2}\rceil,\lceil\frac{j}{2}\rceil,t)=0$ and $M(\lceil\frac{r}{2}\rceil,\lceil\frac{k}{2}\rceil,t)=M$. The action is deemed complete. Otherwise, if $M(i_M,j_M,t)=0$, i.e, the site is empty, then the process repeats for the same $X^b$. From here, $P$ becomes $P\backslash(i,j)$ and a new $(i,j)$ pair are generated uniformly at random from $P$ until either an $M$ is found within the range specified or until $P$ is empty (i.e. after 28 tries). At this point, we will have determined that there were no melanocytes close enough to the xanthoblast for the xanthoblast to successfully pull.

## Growth (continuous time events 14–15)

Given a growth event is chosen in the horizontal (vertical) direction, from each column (row), a site location is chosen uniformly at random at which insertion of new empty site above or below (randomly chosen) will occur. See *Appendix 1—figure 1C-D* for an example of a growth event in the vertical direction.

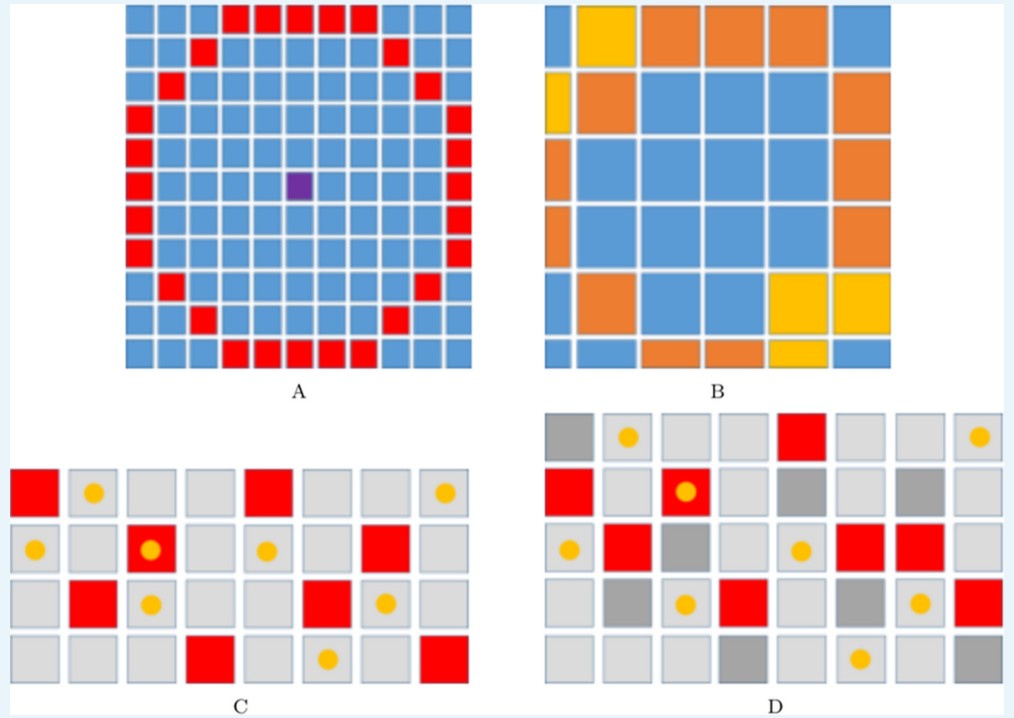

**Appendix 1—figure 1.** Implementation of xanthoblast-melanocyte communication and domain growth. (**A–B**) Range for xanthoblast cytonemes in our model. Figure:range_pull The range of xanthoblast cytonemes on $X$. The centre lattice site (marked in purple) represents site $X(r,k)$. The lattice sites labelled in red are those sites $X(i,j)$ that lie in the range of the cytonemes of an $X^b$ occupying site $X(r,k)$ (the sites given by set $P$). Figure:range_pull_mel The range of xanthoblast cytonemes on $M$. The sites indicated with in orange or yellow are those sites that

can be reached by the range of $X^b$ in $X(i,j)$. The colour of any given site $M(i,j)$ reflects the number of corresponding sites on $X(2i,2j)$, $X(2i-1,2j-1)$, $X(2i-1,2j)$, $X(2i,2j-1)$ that are within range of the cytonemes (marked in red in (**A**)) and thus the probability of the site being chosen in relation to other coloured sites. Specifically, sites labelled in orange are twice as likely to be chosen than those sites marked in yellow. (**C–D**) An example growth event in the vertical direction. Squares represent sites in a given domain. Yellow circles represent cells occupying these sites. (**C**) First a site is chosen at random from each column (marked in red). (**D**) Next a new site is inserted either above or below the first site with probability $\frac{1}{2}$. Each of the chosen sites are marked in dark grey.

To account for the different lattice sizes, each time a growth event in the horizontal (vertical) direction occurs, $\Pi_M^H(t)$ ($\Pi_M^L(t)$) increases by one and $\Pi_X^H(t)$, $\Pi_I^H(t)$ ($\Pi_X^L(t)$, $\Pi_I^L(t)$) increases by two. This means that when a growth event is chosen, the rules described above occur once in the $M$ layer, and twice consecutively in $X$ and $I$. This is to ensure that the domain size remains consistent between the three layers. That is,

$$\Pi_X^L(t) \times \Delta_X = \Pi_I^L(t) \times \Delta_I = \Pi_M^L(t) \times \Delta_M, \tag{59}$$

$$\Pi_X^H(t) \times \Delta_X = \Pi_I^H(t) \times \Delta_I = \Pi_M^H(t) \times \Delta_M, \tag{60}$$

for all time $t$.

## Appendix 2

## Mutant implementation

In this text, we describe how the WT fish simulation is altered in our model to replicate changes known to be present in a variety of mutant fish.

## Simulating the *shd* mutant

The gene *shady* (*shd*) encodes zebrafish leukocyte tyrosine kinase (Ltk) which plays a role in S-iridophore specification (*Lopes et al., 2008*). As a result, strong *shd* mutants lack S-iridophores. To simulate this defect, we remove all S-iridophores from the initial domain, i.e, we set $N_{I^l}^T(0) = N_{I^d}^T(0) = 0$. Since new S-iridophores are only generated by the proliferation of existing S-iridophores, this means that $N_{I^l}^T(t) = N_{I^d}^T(t) = 0$ for all time $t$. (Note: As a result the propensities of events 4, 5, 7, 10 and 11, that is, the movement of $I^l$, movement of $I^d$, proliferation $I^l$ and $I^d$, transition of $I^d$ to $I^l$ and transitions of $I^l$ to $I^d$ are always zero during this simulation.)

## Simulating the *pfe* mutant

Gene *pfeffer* (*pfe*) (alternatively known as *salz* (*sal*)) encodes for colony stimulating factor one receptor (csf1ra) which plays a role in xanthophore specification and migration. In strong alleles, adult fish exhibit no detectable xanthophores in the body of embryos. To simulate this defect we remove all xanthophores and xanthoblasts from the initial domain, *i.e.* we set $N_{X^b}(0) = N_X(0) = 0$. We also remove the fixed event at stage SP where newly differentiated $X$ cells appear along the horizontal myoseptum. Since new xanthophores are only generated by the proliferation of existing xanthophores, and the differentiation of xanthoblasts this means that $N_{X^b}^T(t) = N_X^T(t) = 0$ for all time $t$. (Note: As a result the propensities of events 2, 3, 6, 9, 15, that is, the movement of $X^b$, movement of $X$, proliferation $X$ and $X^b$, differentiation of $X^b$ to $X$ and $X^b$ pulling of $M$ are always zero during this simulation.)

## Simulating the *nac* mutant

The gene *nacre* (*nac*) encodes transcription factor Mitfa (*Lister et al., 1999*). *nac* mutants lack melanocytes throughout embryonic and larval development (*Lister et al., 1999*). To simulate this defect, we remove all melanocytes from the initial conditions, i.e, we set $N_M(0)$. We also set the propensity of event 8, the differentiation of new $M$, to be 0 for all time $t$. Since this is the only way new $M$ can be produced, $N_M(t) = 0$ for all time $t$. (Note: As a result the propensities of events 1 and 14, that is, the movement of $M$, and death of $M$ will also always be zero during this simulation.)

## Simulating the double mutants: nac;pfe, shd;pfe, shd;nac

To simulate the double mutants we combine the effects of two of the three aforementioned mutation types. For example, to simulate *nac;pfe* we combine the effects of *nac* and *pfe* listed above.

## Simulating the *rse* mutant

*Rose* (*rse*), encodes the Endothelin receptor B1a (*Krauss et al., 2014*) and has been shown to acts cell-autonomously in S-iridophores; homozygous mutants result in a reduction of S-iridophores to approximately 20% of that seen in WT (observed in stage PB and adult fish [*Frohnhöfer et al., 2013*]). To simulate *rse* we change the initial conditions so that $N_{I^d}(0)$ is one fifth of the usual number. In our WT simulations, $N_{I^d}(0)$ is $\Pi_I^L(0) \times 3 = 300$ as $I^d$ occupy the

three central rows. In *rse* we set $N_{I^d}(0)=60$, and we place these $I^d$ uniformly at random within the space of the central three rows, since some S-iridophores still appear along the horizontal myoseptum in *rse*. We do this by generating a random number $r \in \mathbb{Z}$ that is uniformly distributed between 24 and 26 (the centre of the horizontal axis is at 25), and a random number $k_1$ that is uniformly distributed between 0 and $\Pi_I^L(0) = 100$. If $I(r_1, k_1)$ is empty we place a $I^d$ cell in this site, otherwise we generate a new position until there are 60 cells of type $I^d$ on $I$. Furthermore since new S-iridophores are only produced by proliferation of pre-existing S-iridophores, we also reduce the rate of proliferation of $I^d$ and $I^l$ to one fifth of the usual number. Therefore, we set the propensity of event 7, $\alpha_7(t) = 0.24 \frac{N_{I^d}^T(t) + N_{I^l}^T(t)}{60 \times 24}$ for all time $t$.

## Simulating the *sbr* mutant

Adult *sbr* mutants exhibit delayed S-iridophore shape transition changes of dense to loose caused by truncations in Tight Junction Protein 1a (**Fadeev et al., 2015**). To simulate *sbr*, we reduce the rate at which $I^d$ attempts transition to $I^l$ to a fortieth of its usual value. Hence the propensity of event 10 becomes $\alpha_{10}(t) = \frac{N_{I^d}(t)}{60 \times 24 \times 40}$. This reduction acts as a proxy for the delay between receiving a signal and changing shape to loose form.

## Simulating the *cho* mutant

Mutant larvae with mutation *cho* lack the horizontal myoseptum (**Svetic et al., 2007**). As a result, dense S-iridophores cannot travel through their usual pathway to generate the initial strip of dense S-iridophores at stage PB seen in WT. Instead loose S-iridophores appear later at stage PR, uniformly across the domain. To simulate the effects of *cho*, we remove the initial three rows of $I^d$ so $N_{I^d}(0) = 0$, as S-iridophores cannot appear along the horizontal myoseptum. Furthermore, we remove the fixed event of metamorphic xanthophores appearing along the horizontal myoseptum at stage SP. To simulate the appearance of loose S-iridophores at stage PR, we provide a new fixed event when the simulation reaches stage PR. At this point, we place 300 (the same number that usually initially occupies the domain in WT) loose S-iridophores in sites uniformly at random across $I$ at stage PR. We do this by generating a random number $r$ between 0 and $\Pi_I^L(t_{PR})$ and a random number $k$ between 0 and $\Pi_I^H(t_{PR})$, where $t_{PR}$ is the time when the simulation first enters stage PR. If $I(r, k)$ is empty we place a $I^l$ cell in this site, otherwise we generate a new $r$, $k$ and continue until there are 200 cells of type $I^l$ on $I$.

## Simulating the *seurat* mutant

Homozygous *seurat* mutants develop fewer adult melanocytes, thus forming irregular spots rather than stripes. This phenotype arises from lesions in the gene encoding Immunoglobulin superfamily member 11 (Igsf11) (**Eom et al., 2012**) which encodes a cell surface receptor containing two immunoglobulin-like domains which is expressed autonomously by the melanocyte lineage. Igsfl1 promotes the migration and survival of these cells during adult stripe development as well as mediating adhesive interactions in vitro.

To model *seurat* we reduced the rate at which melanocytes could differentiate to a twentieth of the usual rate. Hence the propensity of event eight becomes $\alpha_8(t) = 0.05 \times \frac{N_{I^d}(t)}{2 \times 60 \times 24}$. This was to reflect the inhibition of migration of melanoblasts across the domain and increased the rate of attempted melanocyte death to one hundred times per day (usually once per day). Hence the propensity of event 12 becomes $\alpha_{12}(t) = \times \frac{N_M(t)}{100 \times 60 \times 24}$. No other interactions were altered.

## Simulating the *leo* mutant

The gene *leo* encodes Connexin39.4 (Cx39.4). The *leo* mutant displays a spotted pattern across the flank of the fish. In Section "An *in silico* investigation into the function of the *leo* gene" of the main text we describe how we derive the following hypotheses about the impacts of a mutation in the *leo* gene;

- Hypothesis 1: Melanocytes are not repelled by xanthophores
- Hypothesis 2: Xanthophores do not promote the survival of melanocytes in the long range.
- Hypothesis 3: Xanthophores do not promote the change of S-iridophores from dense to loose in the long range.
- Hypothesis 4: Melanocytes lose death signals from local dense S-iridophores and as a result, can differentiate in dense S-iridophore zones.
- Hypothesis 5: Melanocytes lose directed signalling from S-iridophores and hence in the absense of xanthophores differentiate randomly.

To model hypothesis 1, we change the parameter $r_{mx}$, the parameter governing repulsion of melanocytes from xanthophores to 0. To incorporate hypothesis two we remove the criteria for successful melanocyte death given in **Equation 50** in Appendix 1. To model hypothesis 3, we reduce the rate of successful signalling of xanthophores to iridophores to change to loose form. To do this, if a S-iridophore transition is chosen to occur then a number distributed uniformly at random is generated. If this number is less than 0.5, then normal transition signalling occurs (as described in Appendix 1). If the number is greater than 0.5 then the xanthophores send a signal for S-iridophores to transition to dense, that is, the number $N_I, X^S$ is changed to five and the number $N_I, X^L$ is changed to 2. To model hypotheses 4 and 5, we change the melanocyte differentiation success as follows. A melanocyte successfully differentiates into a position on the lattice, if there are no xanthophores on the domain, or if there are xanthophores on the domain and there are three times as many xanthophores in the long range than the number of melanocytes in the long-range.

## Appendix 3

## Quantitative measures

In this text, we describe in more detail how we take quantitative measurements of our simulations.

## Tortuosity

To determine the tortuosity of the X0 stripe, for a specific time point in our simulations the following steps are taken.

- An occupancy matrix $\hat{X}$ of zeros and ones is generated such that a matrix entry $\hat{X}(i,j) = 1$ if $X(i,j) = X$ and 0 otherwise. Sites $(i,j)$ such that $\hat{X}(i,j) = 1$ are yellow and are white otherwise.
- Representative matrix $\hat{X}$ is 'cleaned' using matlab functions bwmorph($\hat{X}$,'clean') and bwmorph($\hat{X}$,'bridge') consecutively. 'Clean' removes any anomalous xanthophores that are not connected (adjacent to) other xanthophores. 'Bridge' adds extra xanthophores where there are holes in the population pattern.
- An algorithm is applied to $\hat{X}$ to create the outline of the X0 stripe. The algorithm to create the top line ($l_t$) is given below in Algorithm 1. A similar algorithm is used to generate the bottom line ($l_b$).
- A line $L$ that represents the middle of the stripe is given by $l_m(i) = \frac{l_t(i) + l_b(i)}{2}$ for $i = 1, ...\Pi_X^L$.
- We smooth $L$ for analysis by applying matlab function smooth. 'Smooth' smooths the data in the column vector $y$ using a moving average filter.
- Finally, we calculate the tortuosity of the line by computing the total length of $L$ divided by the distance (algorithm given below in Algorithm 2).

**Algorithm 1. Algorithm to generate a representative line for the top of stripe X0 ($l_t$)**

$j = \left\lceil \frac{\Pi_X^H}{2} \right\rceil$ ▷*Initialise X0 interstripe search in the center.*
**for** $i = 1 : \Pi_X^L$
 Strike $= 0$ ▷*Initialise the number of consecutive empty sites in a column to be zero.*
 **if** $\hat{X}(i,j) \neq 0$ or ($\hat{X}(i,j) = 0$ and $\hat{X}(i,j+1) \neq 0$) **then**
 **while** $strike < 2$ and $j < \Pi_X^H$ **do** ▷*If we are near to other X, keep moving up to determine the upper bound of the interstripe.*
 $j = j + 1$
 **if** $\hat{X}(i,j) = 0$ **then**
 $strike = strike + 1$ ▷*Increment the number of consecutive empty sites in a column by one.*
 **else**
 $strike = 0$
 **end if**
 **end while**
 $j = j - 2$ *Remove the two consecutively empty sites from the total count.*
 **else**
 **while** $\hat{X}(i,j) = 0$ and $j > \Pi_X^H$ **do** ▷*If we have overshot the interstripe keep moving downwards until we reach the top of the interstripe.*
 $j = j - 1$
 **end while**
 **end if**
 $l_t(i) = j$
**end**

**Algorithm 2. Algorithm to calculate the tortuosity (*tort*) of line $L$ generated from a simulated X0**

$total = 0$
**for** $i = 2 : \Pi_X^L$
 $total = total + \sqrt{(l_m(i) - l_m(i-1))^2 + 1}$
**end** ▷*Compute the length of L using the euclidean distance between consecutive points.*
$tort = \frac{total}{\Pi_X^L}$

To measure the tortuosity of the stripes of real fish, a photograph was taken of the fish at stage J+. Next, matlab function 'getpts' was used to mark the outline of the X0 stripe by a set of points $[\underline{x}, \underline{y}]$ where vector $\underline{x}$ is length $k$. The tortuosity of this line was then computed using Algorithm 3.

---

**Algorithm 3. Algorithm to calculate the tortuosity (*tort*) of line $L$ generated from a real fish**

$total = 0$
**for** $i\ =\ 2\ :\ k$
$\qquad total = total + \sqrt{(x(i) - x(i-1))^2 + (y(i) - y(i-1))^2}$
**end**                                                                    ▷*Compute the length of L using the euclidean*
$tort = \dfrac{total}{\sqrt{(x(1) - x(k))^2 + (y(i) - y(k))^2}}$              *distance between consecutive points.*

---

## X0 interstripe width

To determine the width of the X0 interstripe, used in Section "Quantitative analysis of simulations" of the main text, first, occupancy matrices of $\hat{X}$ and $\hat{I}$ of zeros and ones is generated such that a matrix entry $\hat{X}(i,j) = 1$ if $X(i,j) = X$ and 0 otherwise. Similarly $\hat{I}(i,j) = 1$ if $I(i,j) = I^d$ and 0 otherwise. We then generate $l^t$ and $l^b$ using either $\hat{X}$ or $\hat{I}$ as described in Section "Necessity of S-iridophore assumptions". From these values, we generate $IS$ (X0 interstripe width) by computing;

$$IS = \sum_{i=1}^{\Pi_X^L} \frac{l_t(i) - l_b(i)}{\Pi_X^L}. \tag{61}$$

Note that the X0 interstripe width is computed using whichever is more appropriate of $X$ on $X$ and $I^d$ on $I$ given the mutation. For example, *pfe* does not have xanthophores, so we would use the distribution of $I^d$ on $I$ to determine the width of X0 in this case.

## Adapting the pair correlation function (PCF)

PCFs characterise spatial patterns by calculating a numerical value for the deviation from the situation in which the same number of agents are distributed uniformly at random. In this paper, we use the square uniform PCF (*Gavagnin et al., 2018*) developed for on-lattice systems of agents where distance is measured using the uniform norm. This PCF was originally developed for determining pair correlation between single agents types on a lattice, however, in a technique similar to *Dini et al., 2018* we adapt it here so that it can also be used for identifying correlation between two different types of agents (cell types). First we define the PCF. For each distance $m$ the PCF at distance $m$ is given by:

$$\mathrm{PCF}(m) = \frac{c^{C_1, C_2}(m)}{\mathbb{E}[\hat{c}^{C_1 C_2}(m)]} \tag{62}$$

where $c^{C_1, C_2}(m)$ is defined as the number of cells of type $C_1$ we would expect to find at distance $m$ from cells of type $C_2$ using the uniform metric under zero flux boundary conditions. $\mathbb{E}[\hat{c}^{C_1, C_2}(m)]$ if the cells were positioned uniformly at random on the domain. This can be calculated by counting the number of agents of this distance manually from the lattice. To compare cell type $M$ with cells that lie on domains other than $M$ we must transform $M$ into a matrix $\hat{M}$ of size $\Pi_X^H \times \Pi_X^L$ where $\hat{M}(r,k) = M$ if $M(\lceil \frac{r}{2} \rceil, \lceil \frac{k}{2} \rceil) = M$ and 0 otherwise. Hence the set of site positions distance $m$ from each other on any domain $D \in \{X, \hat{M}, I\}$ under zero flux boundary conditions can be given by

$$S_m = \{(r,k), (i,j) \in (\Pi_X^L, \Pi_X^H), (\Pi_X^L, \Pi_X^H) \mid \max\{|r - i|, |k - j|\} = m\}. \tag{63}$$

Therefore

$$c^{C_1,C_2}(m) = \begin{cases} \sum_{(i,j),(r,k)\in S_m} 1_{D_1(r,k)=D_2(i,j)=C}, \text{where} C_1 = C_2 = C, \\ \sum_{(i,j),(r,k)\in S_m} 1_{D_1(r,k)=C_1 \text{and} D_2(i,j)=C_2} + 1_{D_2(r,k)=C_2 \text{and} D_1(i,j)=C_1} \text{otherwise}, \end{cases} \tag{64}$$

where $C_1$ lies on $D_1$, $C_2$ lies on $D_2$ and $D_1, D_2 \in \{X, \hat{M}, I\}$. Note that $|S_m|$ is the number of site pairs distance $m$ from one another. $|S_m|$ is computed in *Gavagnin et al., 2018* and is given by

$$|S_m| = 4m\Pi_X^H\Pi_X^L - 3(\Pi_X^H + \Pi_X^L)m^2 + 2m^3. \tag{65}$$

To determine $\mathbb{E}[\hat{c}^{C_1,C_2}(m)]$ there are three cases.

Case 1: $C_1 = C_2 = C \in \{M, X, I^d, I^l, X^b\}$ lies on domain $D \in \{M, X, I\}$ (with volume exclusion on $D$). For example spatial correlation of $M$ with respect to other $M$. This is the case discussed in *Gavagnin et al., 2018*. In this case

$$\mathbb{E}[\hat{c}^{C,C}(m)] = \mathbb{P}(\text{Two agents of type } C \text{ are chosen from domain } D)|S_m|. \tag{66}$$

$$\mathbb{P}(\text{Two cells of type } C \text{ are chosen from domain } D) = \frac{N_C^T(N_C^T - 1)}{(\Pi_X^H\Pi_X^L)(\Pi_X^H\Pi_X^L - 1)}. \tag{67}$$

This is because there are $N_C^T$ ways to choose a cell of type $C$ and $N_C^T - 1$ ways to choose a cell of type $C$ given one has been chosen already. There are $\Pi_X^H\Pi_X^L$ possible positions for the first cell and then $\Pi_X^H\Pi_X^L - 1$ possible positions for the second cell due to volume exclusion on domain $D$.

Case 2: $C_1 \neq C_2$, where $C_1, C_2 \in \{X, I^d, I^l, X^b\}$ both lie on domain $D \in \{X, I\}$ (with volume exclusion on $D$). For example spatial correlation of $X$ with $X^b$. In this case

$$\mathbb{E}[\hat{c}_d^{C_1,C_2}(m)] = \mathbb{P}(\text{One cell of type } C_1 \text{ and one cell of type } C_2 \text{ are chosen from domain } D)2|S_m|, \tag{68}$$

Notice, $|S_m|$ is multiplied by two here because for each pair $(i,j), (r,k) \in S_m$ there are two positions $C_1$ and $C_2$ can take and be distance $m$ away from each other. Specifically these are the cases $D_1(i,j) = C_1$, $D_2(r,k) = C_2$ and $D_2(i,j) = C_2$, $D_1(r,k) = C_1$. We can compute

$$\mathbb{P}(\text{One cell of type } C_1 \text{ and one cell of type } C_2 \text{are chosen on D}) = \frac{N_{C_1}^T N_{C_2}^T}{(\Pi_X^H\Pi_X^L)(\Pi_X^H\Pi_X^L - 1)}. \tag{69}$$

This is because there are $N_{C_1}^T$ ways to choose a cell of type $C_1$ and $N_{C_2}^T$ ways to choose a cell of type $C_2$. There are $\Pi_X^H\Pi_X^L$ possible positions for the first cell and then $\Pi_X^H\Pi_X^L - 1$ possible positions for the second cell due to volume exclusion on domain $D$.

Case 3: $C_1, C_2$ where $C_1, C_2 \in \{M, X, I^d, I^l, X^b\}$ lie on domains $D_1, D_2 \in \{\hat{M}, X, I\}$ where $D_1 \neq D_2$, for example the spatial correlation of $X$ and $I^d$. In this case

$$\mathbb{E}[\hat{c}^{C_1,C_2}(m)] = \\ \mathbb{P}(\text{One cell of type } C_1 \text{ is chosen on } D_1 \text{ and one cell of type } \\ C_2 \text{ is chosen on } D_2)2|S_m|, \tag{70}$$

where

$$\mathbb{P}(\text{One cell of type } C_1 \text{ is chosen on } D_1 \text{ and one cell of type } C_2 \text{ is chosen on } D_2) = \frac{N_{C_1}^T N_{C_2}^T}{(\Pi_X^H\Pi_X^L)^2}. \tag{71}$$

This is because there are $N_{C_1}^T$ ways to choose a cell of type $C_1$ and $N_{C_2}^T$ ways to choose a cell of type $C_2$. There are $\Pi_X^H\Pi_X^L$ possible positions for the first cell of type $C_1$ and $\Pi_X^H\Pi_X^L$ possible positions for the second cell of type $C_2$ since they lie on different domains (no volume exclusion). A summary of all these cases can be given as follows;

- $C_1 = C_2 = C$ lies on domain $D$ (with volume exclusion on $D$). For example, spatial correlation of $M$ with respect to other $M$.

$$PCF(m) = \frac{c^{C,C}(m)(\Pi_X^H \Pi_X^L)(\Pi_X^H \Pi_X^L - 1)}{(4m\Pi_X^H \Pi_X^L - 3(\Pi_X^H + \Pi_X^L)m^2 + 2m^3)(N_C^T(N_C^T - 1))} \tag{72}$$

- $C_1 \neq C_2$, $C_1, C_2$ lie on domain $D$ (with volume exclusion on $D$). For example spatial correlation of $X$ with respect to $X^b$.

$$PCF(m) = \frac{c^{C_1,C_2}(m)(\Pi_X^H \Pi_X^L)(\Pi_X^H \Pi_X^L - 1)}{2(4m\Pi_X^H \Pi_X^L - 3(\Pi_X^H + \Pi_X^L)m^2 + 2m^3)(N_{C_1}^T(N_{C_2}^T))} \tag{73}$$

- $C_1 \neq C_2$, $C_1, C_2$ lie on domains $D_1, D_2$ where $D_1 \neq D_2$ for example spatial correlation of $X$ with respect to $M$.

$$PCF(m) = \frac{c^{C_1,C_2}(m)(\Pi_X^H \Pi_X^L)^2}{2(4m\Pi_X^H \Pi_X^L - 3(\Pi_X^H + \Pi_X^L)m^2 + 2m^3)(N_{C_1}^T(N_{C_2}^T))} \tag{74}$$

## Appendix 4

## Testing robustness

Due to the abundance of parameters and cell–cell interactions necessary to capture what is known biologically about zebrafish pigment pattern formation, it is not feasible to perform an exhaustive parameter sweep to demonstrate the robustness of the model. Instead, as a test of robustness, we perform a rigorous robustness analysis by carrying out one hundred repeats of all WT and mutant simulations with perturbed parameter values chosen uniformly at random from the range 0.75–1.25 of their described value. The value of each parameter is sampled uniformly from this region, independently for each parameter and each repeat. Ten of these randomly sampled repeats for *shd*, *nac*, *pfe* and *leo* are given in **Appendix 4—figure 1**. We observe, for all one hundred repeats, that small perturbations to the rates still generate consistent patterning, demonstrating the robustness of the model.

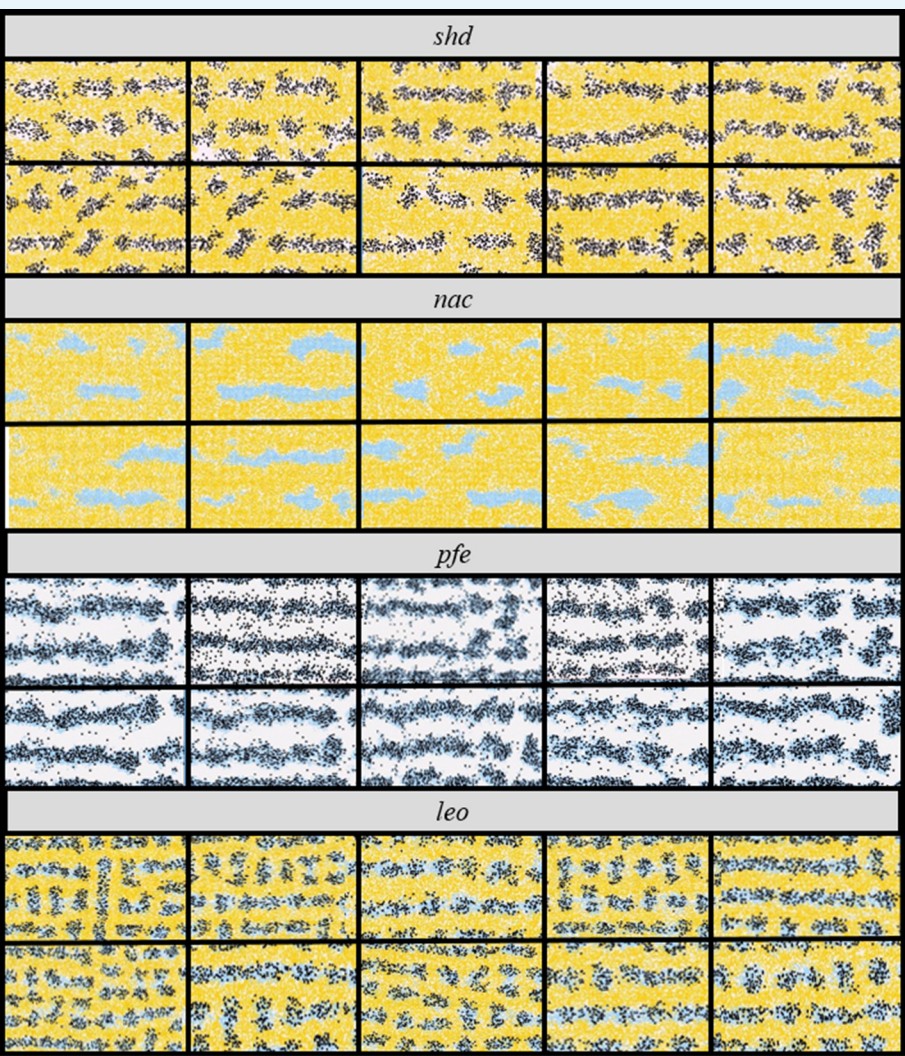

**Appendix 4—figure 1.** Example *shd*, *nac*, *pfe* and *leo* simulations at stage J+ when the parameters governing the rate of proliferation, movement, differentiation and death are perturbed. Each rectangle is an example simulation at stage J+ for which each rate parameter is perturbed to 1+$x$ times its normal value. The value $x$ is chosen uniformly at random from the interval $[-0.25, 0.25]$.

## Appendix 5

### Predicting pattern formation for an *seurat/sbr* cross

In this section, we give an example of how the model can be manipulated to investigate aspects of pattern development and to predict the outcome of mutant pigment patterns.

To the best of our knowledge, the adult pattern of crossed *seurat* and *sbr* mutants have not have been published previously. By changing the parameters so that we match both *sbr* and *seurat* we can predict pattern formation for a double mutant *seurat/sbr*. As previously stated (in more detail) in Appendix 2, the mutation in *seurat* affects melanocyte differentiation (*Eom et al., 2012*) and the mutation in *sbr* affects the dense-to-loose S-iridophore transition (*Fadeev et al., 2015*). To simulate the cross, we simply incorporate the effects of both mutations simultaneously. The results at stage J+ is given in *Appendix 5—figure 1*. Our model predicts that when both of these mutations occur, dense S-iridophores and associated xanthophores would cover most of the flank of the fish. A few melanocytes associated with loose S-iridophores survive at the very dorsal and ventral regions of the fish.

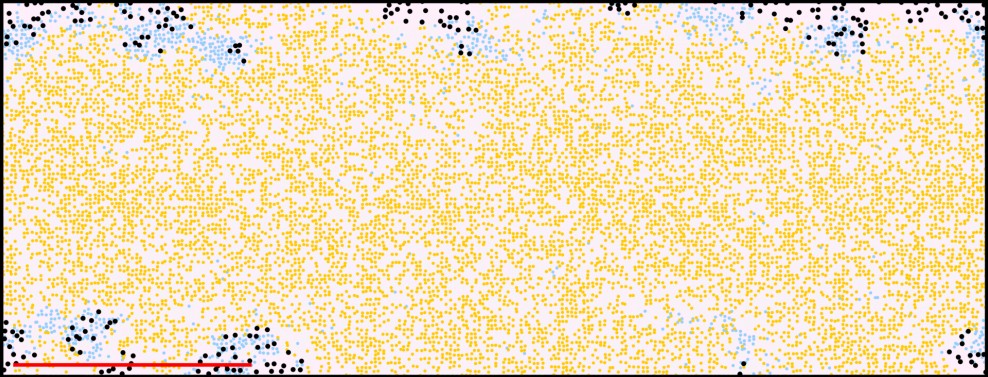

**Appendix 5—figure 1.** Prediction for the pattern of a *seurat/sbr* cross at stage J+.

