## [Decision Letter]

**Acceptance summary:**

This paper constructs and analyses a detailed computational model of zebrafish pigment stripe formation. It highlights the role of iridophores in organizing the melanophores and xanthophores, a facet that has largely been unaccounted for in computational modeling of zebrafish stripe patterns. Of particular interest is the analysis of the *schachbrett* mutant, which points to the role of tight junctions in the transition of S-iridophore packing.

**Decision letter after peer review:**

Thank you for submitting your article "A quantitative model for zebrafish pattern formation" for consideration by *eLife*. Your article has been reviewed by three peer reviewers, and the evaluation has been overseen by a Reviewing Editor and Aleksandra Walczak as the Senior Editor. The following individuals involved in review of your submission have agreed to reveal their identity: Sreelaja Nair (Reviewer #1); Raj K Ladher (Reviewer #3).

The reviewers have discussed the reviews with one another and the Reviewing Editor has drafted this decision to help you prepare a revised submission.

The manuscript describes a computational model that attempts to capture the complexities of xanthophore and iridophore interactions to explain zebrafish pigment stripe pattern formation. While the reviewers appreciate that the model is sufficient to reproduce known stripe patterns, it has not been demonstrated that the interactions included are necessary, and all reviewers were of the opinion that the model has not been used to make sufficiently interesting predictions. In order for a revision to be publishable, it must distinguish the work sufficiently from the work of Volkening and Sandstede by including several more predictions (see points 1,2,3 of reviewer 1, and point 3 of reviewer 3). In addition, as pointed out by reviewer 2, it is essential that the existing and added predictions be shown to be robust to changing parameter values within biologically reasonable ranges – tuning parameters one by one, keeping all others fixed, only between low and high values, is not a sufficient exploration of the parameter space to prove such robustness. A more comprehensive exploration of parameter space is also essential to satisfy reviewer 2 that the interactions included in the model are indeed necessary as claimed in the manuscript, and that despite the very large number of parameters the behaviour of the model is constrained enough to make robust predictions. The manuscript should also tone down claims of the novelty of the methods and the superiority of this approach compared to reaction-diffusion approaches – in particular, as reviewer 2 points out, stochasticity is easily included in reaction-diffusion approaches so that cannot count as a reason to favour a lattice-based model.

*Reviewer #1:*

In the manuscript entitled "A quantitative modeling approach to zebrafish pigment pattern formation", Owen et al. describe in silico modeling strategies to explain zebrafish pigment stripe formation. The authors use on-lattice modeling, in which five pigment cell types that are necessary to form the adult zebrafish pigment stripe pattern are treated as fixed to a mutually exclusive space within the domain in which the pattern will emerge. Onto this basic framework, the authors model known behaviors of the pigment cell types and intercellular interactions to simulate emergence of stripe pattern. Their modeling approach incorporates the role of iridophores in organizing the melanophores and xanthophores, a facet that has largely been unaccounted for in computational modeling of zebrafish pigment stripe pattern formation. The authors validate their model by recapitulating the wild type stripe pattern and test the model by simulating stripe pattern formation in the absence of one or more of the cell types/interactions. These additional simulations very nicely recreate stripe patterns observed in known zebrafish mutants. Overall the study is well carried out and the authors comprehensively validate their modeling strategy, making a strong case for on-lattice agent based modeling rather than a Reaction Diffusion modeling for pigment stripe pattern in zebrafish.

It is not surprising that the model recapitulates known pigment patterns, since the model was built based on biological evidences. On the basis of this the authors state that the current experimental evidences together with their model’s assumptions of S-iridophore behavior is sufficient to explain pigment pattern formation. For a set of biological interactions to be termed as sufficient for formation of a pattern, the model could perhaps be challenged by stating predictions that are currently not easy to test experimentally.

1) For example: As the authors point out, zebrafish do not incorporate additional pigment stripes as they grow. The biological basis for this is not known. In their wild type simulation starting at stage PB, how would a pattern evolve if initially the melanophores were in 3 or 5 stripes instead of the normal 4 stripes?

2) At the start of the simulations at stage PB, the dense iridophores appear along the horizontal myoseptum. Is this spatial location of iridophores relevant to formation of the pattern? Would the pattern that emerges be different if the original metamorphic pattern of dense iridophores at the horizontal myoseptum was displaced dorsally or ventrally?

3) The authors very nicely factor in the layered arrangement of the different cell types along the z-axis, to mirror the biological scenario. It is experimentally tough to switch the order of the layers, but should be possible to do in the simulations? This would be interesting because developmentally the three pigment cell types take distinct dorsal or ventral migratory routes along the horizontal myoseptum to reach their eventual z-layer. The route of migration has been thought to be critical in determining the fate of the pigment cell type (or of the stem cells that resides in the adult) and in establishing the stripe pattern. Could this assumption be partially tested to determine the relevance of the spatial order of the layers in eventual stripe formation?

Overall, I enjoyed the manuscript and the insights mathematical modeling can provide to understand complex phenomena such as pattern formation. My comments stem from limitations that mutants etc pose for understanding what information is sufficient to generate a pattern. The model is based on a finite set of proven interactions, several untested permutations and combinations of these and additional novel interactions are always possible. A major advantage of mathematical modeling is in the predictive zone, some of which experimentalists may find hard/impossible to venture into. The current study falls short of taking that leap, which would have been an interesting and informative exercise.

*Reviewer #2:*

This manuscript reports the study of pigment patterns in zebrafish embryos claiming a novel bottom-up stochastic model. My understanding is that this is an extensively employed standard approach using the chemical master equation for interacting (and diffusing) species to simulate emergent stochastic patterns. The authors incorporate many detailed and complex interactions between five different cell types and claim that all the interactions are essential to explain the observed patterns. My feeling is that such a study with a large number of control parameters is overkill and does not enhance our understanding of stripe patterns. As such, I do not recommend publishing this study in *eLife*.

1) In this manuscript, the authors present a quantitative model for simulating the stripe patterns seen in wt and mutant embryos of zebrafish. They develop a stochastic description based on the purported interactions between xanthophores, xanthoblasts, melanocytes and (two kinds of) iridophores while also incorporating motility that is biased by short- and long-ranged interactions between the cell types. These are augmented by cell division, death, differentiation and tissue growth. The resulting detailed numerical simulations are shown to qualitatively reproduce pigment patterns in wt and mutant conditions. Further, the authors quantitatively compare cell density, straightness of stripes and interstripe widths with experimental results. Several other results, such as the pair-correlation-function of cell densities, provide further justifications to the model developed.

2) The main point of this study seems to be the development of a bottom-up stochastic model incorporating as many complex and varied interactions between the cells to simulate pigment patterns. The authors contrast their approach with reaction-diffusion modelling approaches (such as Turing patterns) by saying that such approaches lack the stochasticity observed inherently in pigment patterns.

I would disagree with this viewpoint. The point of reaction-diffusion approaches is minimalistic coarse-grained descriptions of the complex effective interactions between cells (or any other constituents of the patterns) that can capture the essence of the emergent patterns. It is easy to incorporate stochasticity even in approaches that use partial-differential equations.

A complementary approach is to simulate the underlying master equation that governs the chemical interactions, supplemented by a compartment-based approach to diffusion. There are several such studies done in the past using the standard Gillespie algorithm for simulating the chemical master equation. Such studies incorporate the essential chemistry (e.g., activator-inhibitor dynamics) and coupled with diffusion can lead to emergent stripe patterns with inherent stochasticity built-in.

3) The authors, instead, develop a detailed lattice-based model with sufficiently complex interactions between 5 species of cells along with short-ranged local interactions and long-ranged non-local interactions. These interactions bias the hopping rates of the cells along the lattice. The authors speculate on the possible physical origins of these interactions. However, it is not clear if there aren't other possible interactions that could also lead to the same emergent dynamics. And hence the questions arises as to what are the essential features required at this level of modeling and what details are incidental. It should be noted that even though the authors claim that their approach is bottom-up, they are not modelling interactions between molecules. Rather, the interactions incorporated in their modeling are also effective cell-level interactions. In essence, this study has considered a master-equation like approach to simulate zebrafish stripe patterns.

4) The authors claim that all the interactions included in their model are essential for stripe patterns. This is demonstrated by turning off each interaction in-turn and showing that some feature of the pattern is lost. This is an 'all or none' switch. For the set of other parameters chosen, this might indeed lead to the conclusion that all interactions are absolutely needed. One could vary the interaction strengths in a continuous manner and then it is not clear if all the said interactions are absolutely needed for all parameter values.

5) Looking at the appendix and the supplementary material, I feel that the model is sufficiently complicated, and has so many turning parameters, that any range of behaviors is possible. It is not unreasonable to see that the large-scale emergent dynamics of such a complex model is essentially that of Turing systems which could have been simulated with much less complexity.

For the above reasons, I cannot recommend publishing in *eLife*.

*Reviewer #3:*

Owen et al. describe the formulation of a model to understand the formation of stripes in zebrafish. They revisit the Turing model that previous studies have described and refine the interaction parameters that were simplified in those models. The model presented recapitulates many of the patterns found in WT and in mutants, and models the formation of these stripes.

The study does capture the complexities of xanthophore and iridophore interactions well. Furthermore, the ability to place weights on the strength of interactions also gives the model flexibility.

The paper is well-written. However, I would like to make some suggestions, listed below.

1) The assumptions and the way the model works could be made more explicit. For this I would suggest that the authors consider incorporating Figure 1, 2 and 3 from Appendix 1 into the main manuscript.

2) I would suggest that the authors emphasise the differences of their model from the Volkening paper published in Aug 2018. I do take issue with the authors characterisation of this paper as very recent, and would suggest that they make greater reference to it in the Introduction.

3) One very interesting piece of data from this study, and the one that does differentiates this study from that of Volkening and Sandstede, is the analysis of the *schachbrett* mutant. This mutant points to the role of tight junctions in the transition of S-iridophore packing. It would be worth extending this analysis to the compunctiond *sbr*/*leo* and *sbr*/luc mutants described in Fadeev et al. Additionally, there is a suggestion of an interaction between *sbr* and *seurat* – I wonder if the model were able to predict what a double *sbr/seurat* would look like?

[Editors' note: further revisions were suggested prior to acceptance, as described below.]

Thank you for re-submitting your article "A quantitative model for zebrafish pattern formation" for consideration by *eLife*. Your article has been re-reviewed by three peer reviewers, and the evaluation has been overseen by a Reviewing Editor and Aleksandra Walczak as the Senior Editor. The following individuals involved in review of your submission have agreed to reveal their identity: Sreelaja Nair (Reviewer #1); Raj K Ladher (Reviewer #3).

The reviewers have discussed the reviews with one another and the Reviewing Editor has drafted this decision to help you prepare a revised submission.

The manuscript has been improved but there are some remaining issues that need to be addressed before acceptance, as outlined below:

1) In the revised manuscript, you write:

"The most commonly used mathematical paradigm for stripe formation takes the form of a Turing reaction diffusion model. In these representations, melanocytes and xanthophores diffuse and interact via a few long and short range 'reactions'. This class of model typically rely on a small number of parameters which, upon being altered, can generate a diverse range of patterns. Simplified models such as these have the benefit that they are often analytically tractable, allowing a deep understanding of the model. However, their main limitation is that, due to the simplicity of the approach, there is often no consistent way to link the parameters with measurable data, making it difficult to relate the model results back to the biology."

This gives the impression that (i) Turing pattern models are always simplified, and analytically tractable, and (ii) Turing models face a difficulty in relating their parameters to measurable data. Both are untenable statements and should be removed – there are several examples of both highly nonlinear Turing-type or reaction-diffusion models for pattern formation that are neither simple nor analytically tractable, and in several cases now people have shown how to relate their parameters to measurable data.

2) The authors write:

"Turing reaction-diffusion type models posit that combinations of short and long range dynamics between melanocytes and xanthophores generate stripe patterns. Indeed, much of the excitement around such models is the ease with which small parameter value changes result in diverse patterns, many readily recognisable from nature. […] A major difference between our model and Turing reaction diffusion models is that small parameter changes in our model do not typically generate qualitatively different patterns, whereas Turing reaction diffusion models can show substantial pattern changes in response to small changes (Budi, Patterson and Parichy, 2011; Yamaguchi, Yoshimoto and Kondo, 2007)."

This gives the impression that large changes in behaviour due to small changes in parameters is a generic feature of Turing models, as opposed to the model in this manuscript. This is not true, and the statements implying this should be removed – it is quite possible for Turing and reaction-diffusion models to produce small changes upon small changes in parameters, and equally it is possible for stochastic lattice models to exhibit large changes in behaviour upon small changes in parameters – it depends on the interactions and nonlinearities included in the model.

3) Finally:

"From an analytical perspective, a significant advantage of our on-lattice model in contrast to off-lattice models is their amenability to the derivation of a continuum model. Our model therefore opens up the opportunity for future exploratory work using a continuum model for mutants *pfe* and *nac* in order to explore whether pattern formation in these cases individually can be described as Turing patterns and to determine parameter ranges for successful pattern formation."

Here you argue, and we agree, that the model in the manuscript is in fact simply a discretized version of a reaction-diffusion model (albeit a quite complex one with many dynamical variables and many interactions). Thus, there is no great difference in approach between the discretized stochastic model you analyze and reaction-diffusion models, so please remove all statements that imply a large difference in your approach vs. reaction-diffusion or Turing type models. Further, a continuum limit can easily be constructed for off-lattice models as well, so this is not an advantage of your approach over that of Volkening et al. Please remove this statement, and present the opportunity for building continuum models as one that applies to both your model as well Volkening et al.'s model.

---

## [Author Response]

The manuscript describes a computational model that attempts to capture the complexities of xanthophore and iridophore interactions to explain zebrafish pigment stripe pattern formation. While the reviewers appreciate that the model is sufficient to reproduce known stripe patterns, it has not been demonstrated that the interactions included are necessary, and all reviewers were of the opinion that the model has not been used to make sufficiently interesting predictions. In order for a revision to be publishable, it must distinguish the work sufficiently from the work of Volkening and Sandstede by including several more predictions (see points 1,2,3 of reviewer 1, and point 3 of reviewer 3).

We have extended our work substantially to include a number of new predictions as suggested by the reviewers, including:

1) Predicting the potential effects of changing the initial conditions for stripe formation

a) by changing the initial iridophore horizontal striping to be vertical (see subsection “Initial S-iridophore interstripe orientation alone does not determine the orientation of stripes and interstripes and Figure 13A);

b) by displacing the initial horizontal interstripe vertically, so that it appears lower down the initial domain (see subsection “The position of the initial S-iridophore interstripe is important for successful pattern formation” and Figure 13B);

c) by changing the initial domain size while maintaining the usual growth rates (see subsection “Initial domain size contributes to the number of stripes and interstripes” and Figure 13C);

d) by populating the initial domain with pre-formed stripes and observing stripe insertion (see subsection “Initial domain size contributes to the number of stripes and interstripes” and Figure 13D).

2) Formulating and then testing hypotheses about the functions of the *leo* gene by replicating the range of mutant and cross mutant patterns (see subsection “An in silico investigation into the function of the *leo* gene” and Figure 14)

3) Predicting the patterning of *seurat* mutants and double mutants based on the known biology. These extensions are included in the subsection “Model predictions” of the main text along with our original predictions and (for *seurat* mutants) in the Discussion and Appendix 5.

In addition, as pointed out by reviewer 2, it is essential that the existing and added predictions be shown to be robust to changing parameter values within biologically reasonable ranges – tuning parameters one by one, keeping all others fixed, only between low and high values, is not a sufficient exploration of the parameter space to prove such robustness.

We performed a rigorous robustness analysis by carrying out one hundred repeats for simulations of each of the WT and four of the main mutant types – *nac*, *pfe*, *shd* and *leo –* with perturbed parameter values chosen uniformly at random from the range between 0.75-1.25 of their described value. The process is described in detail in subsection “Necessity of S-iridophore assumptions”. Twenty (randomly selected) of the repeats for WT are displayed in the new Figure 9 of the manuscript and ten (randomly selected) of the repeats for each of *nac*, *pfe*, *shd* and *leo* are given in Appendix 4—figure 1. For all one hundred repeats these significant perturbations to the rates still generate consistent patterning, in both WT and all mutant cases, demonstrating the robustness of the model.

A more comprehensive exploration of parameter space is also essential to satisfy reviewer 2 that the interactions included in the model are indeed necessary as claimed in the manuscript, and that despite the very large number of parameters the behaviour of the model is constrained enough to make robust predictions.

Most of the features we have included in the model have been demonstrated through biological experimentation and therefore are incorporated for completeness. A small number of the interactions included in the model are predictions focussed on the role of iridophores. We tested the necessity of these predictions by removing the corresponding model feature and demonstrating that the reduced model fails to replicate either the WT and/or mutant pattern predictions (Figure 11 of the revised manuscript). We further tested our predictions by applying our model to other mutants such as *choker* and *sbr* in order to demonstrate that these mutants are also successfully replicated by the model without the need for further assumptions (Figure 7O-Q’ of the revised manuscript). We have also carried out a more comprehensive test of the biologically suggested features demonstrating that the removal of each model feature independently leads to incorrect pattern formation in the model (Figure 12). This acts as a clear demonstration of the necessity of all the interactions included in the model.

The manuscript should also tone down claims of the novelty of the methods and the superiority of this approach compared to reaction-diffusion approaches – in particular, as reviewer 2 points out, stochasticity is easily included in reaction-diffusion approaches so that cannot count as a reason to favour a lattice-based model.

We have removed the suggestion that Turing models cannot incorporate stochasticity and toned-down claims of the superiority of the approaches in comparison to reaction-diffusion methods in the Introduction. Instead we have focussed on the important differences between the Turing reaction-diffusion modelling approach and our more biologically interpretable modelling approach.

Reviewer #1:[…] It is not surprising that the model recapitulates known pigment patterns, since the model was built based on biological evidences. On the basis of this the authors state that the current experimental evidences together with their model`s assumptions of S-iridophore behavior is sufficient to explain pigment pattern formation. For a set of biological interactions to be termed as sufficient for formation of a pattern, the model could perhaps be challenged by stating predictions that are currently not easy to test experimentally.1) For example: As the authors point out, zebrafish do not incorporate additional pigment stripes as they grow. The biological basis for this is not known. In their wild type simulation starting at stage PB, how would a pattern evolve if initially the melanophores were in 3 or 5 stripes instead of the normal 4 stripes?

It appears that the reviewer may have slightly misinterpreted our initial conditions. Initially, within our simulations, the melanophores are not in a striped arrangement, instead they appear dynamically across the domain, in a randomly dispersed pattern (Figure 4A’), as reported in real zebrafish (see Figure 4A). Whilst in real WT zebrafish shown in Figure 4A there are some melanocytes in a stripe-orientation that can be observed under the skin, these are much deeper below the skin layer and do not contribute to pattern development. We commented on this in the caption of Figure 4: “White arrows indicate the embryonic pattern of melanocytes in four stripes that are deeper than the skin level and are consequently *not* included in the model.”

Furthermore, we note that zebrafish do incorporate additional pigment stripes as they grow. However, it is important to note that these new stripes are added at the leading edges of a spreading pattern rather than between pre-formed stripes as has been documented, for instance, in the angelfish genus *Pomocanthus*.

Nevertheless, in order to investigate the potential for stripe insertion in zebrafish, we simulated a domain which is initially striped (two stripes and one interstripe) at stage PB. The resulting patterns demonstrate, as the reviewer suggested, that in this case new interstripes appear in-between already developed stripes, consistent with the *Pomocanthus* study we now reference. We have illustrated our results in Figure 13D and provided an associated commentary in the subsection “Stripe insertion can occur on an initially striped domain”.

2) At the start of the simulations at stage PB, the dense iridophores appear along the horizontal myoseptum. Is this spatial location of iridophores relevant to formation of the pattern? Would the pattern that emerges be different if the original metamorphic pattern of dense iridophores at the horizontal myoseptum was displaced dorsally or ventrally?

We have addressed the reviewer’s interesting suggestion by exploring what happens if, hypothetically, the initial iridophore stripe was in a different place. Given how we envisage the role of that initial iridophore stripe in the organisation of the melanophores, we predicted that, if we were to move the position of the initial interstripe, but maintain its horizontal orientation, that the stripes would still form sequentially around this initial interstripe, but in a pattern displaced along the dorso-ventral axis. We tested this hypothesis in the model and confirmed that this is exactly what happens. We have included simulations of pattern development from just such an initial condition in Figure 13B, discussing the similarities and differences to biologically plausible wild-type patterning in subsection “The position of the initial S-iridophore interstripe is important for successful pattern formation”. The stripes are more disrupted than with the wild-type initial condition. This effect is likely due to the altered impact of domain growth on the pattern: In the model, growth is centred at the middle of the domain and so when the initial stripe is not similarly centred, growth disrupts pattern formation.

Another interesting test case inspired by the reviewer’s comments, concerns the implication for pattern formation of altering the orientation of the initial iridophore stripe, so that it is oriented vertically (rather than horizontally as in wild-type fish). Based on the assumed role of the iridophore stripe in orienting the stripe pattern, we predicted that this change would result in vertical barring. In the simulation in Figure 13A and commentary in subsection “Initial S-iridophore interstripe orientation alone does not determine the orientation of stripes and interstripes”, it can be seen that, whilst initially the vertical iridophores drive a vertically oriented banding pattern, this gradually reorganises, so that at the end of the simulation, the pattern is a meshwork of vertical and horizontal striping. This result clearly supports the idea that the initial iridophore pattern is important in establishing stripe/interstripe orientation, but importantly reveals the previously unsuspected importance of domain growth in orienting striping in zebrafish.

3) The authors very nicely factor in the layered arrangement of the different cell types along the z-axis, to mirror the biological scenario. It is experimentally tough to switch the order of the layers, but should be possible to do in the simulations? This would be interesting because developmentally the three pigment cell types take distinct dorsal or ventral migratory routes along the horizontal myoseptum to reach their eventual z-layer. The route of migration has been thought to be critical in determining the fate of the pigment cell type (or of the stem cells that resides in the adult) and in establishing the stripe pattern. Could this assumption be partially tested to determine the relevance of the spatial order of the layers in eventual stripe formation?

The reviewer raises an interesting question: What determines the fate of pigment cells generated from the adult stem cells? To our knowledge there has been little study of the way in which the layered arrangements are built up over juvenile stages. For this reason, our model does not capture the migration pathway to the epidermis, but only the patterning process once the cells have arrived. Cells are modelled as interacting without explicit reference to layering. Thus, the order of the layers does not influence the output of our model. The distinct layers are incorporated in order to aptly represent the role of excluded volume between cells of the same type. That is, melanophores should occupy space available to other melanophores, but not to xanthophores for example, since they occupy a separate *z*-layer. We have added the sentence ‘We note that in our simulations, the ordering of the layers does not play any role in determining pattern formation.’ to the manuscript in the caption of Figure 2 in order to make this clear.

Reviewer #2:This manuscript reports the study of pigment patterns in zebrafish embryos claiming a novel bottom-up stochastic model. My understanding is that this is an extensively employed standard approach using the chemical master equation for interacting (and diffusing) species to simulate emergent stochastic patterns. The authors incorporate many detailed and complex interactions between five different cell types and claim that all the interactions are essential to explain the observed patterns. My feeling is that such a study with a large number of control parameters is overkill and does not enhance our understanding of stripe patterns. As such, I do not recommend publishing this study in eLife.

We feel that the reviewer has perhaps misunderstood the motivation for our paper. As they have stated there are many demonstrations in the literature, including those we cite, which show that diverse patterns resembling those in wild-type and mutant zebrafish *can be* reproduced by a reaction-diffusion master equation (RDME) approach. Our aim, rather than asking if reaction-diffusion mechanisms could generate a pattern, was to ask whether the cell-cell interactions that have already been characterised biologically in the literature are necessary and sufficient to generate the observed biological patterning. Moreover, we investigate whether the resulting model can account for the diverse array of patterns seen in a variety of mutants, if the known biology corresponding to these mutants is incorporated. Though we agree that the number of parameters in this model is large, we argue that this reflects the demonstrated complexity of the underlying biology. Moreover, the iridophore interactions that are incorporated are based on hypotheses derived from biological studies; they have *not* been chosen for modelling convenience or to replicate particular patterns. Our model then provides a tool to determine whether these hypotheses are consistent with wild-type and mutant pattern formation.

1) In this manuscript, the authors present a quantitative model for simulating the stripe patterns seen in wt and mutant embryos of zebrafish. They develop a stochastic description based on the purported interactions between xanthophores, xanthoblasts, melanocytes and (two kinds of) iridophores while also incorporating motility that is biased by short- and long-ranged interactions between the cell types. These are augmented by cell division, death, differentiation and tissue growth. The resulting detailed numerical simulations are shown to qualitatively reproduce pigment patterns in wt and mutant conditions. Further, the authors quantitatively compare cell density, straightness of stripes and interstripe widths with experimental results. Several other results, such as the pair-correlation-function of cell densities, provide further justifications to the model developed.2) The main point of this study seems to be the development of a bottom-up stochastic model incorporating as many complex and varied interactions between the cells to simulate pigment patterns. The authors contrast their approach with reaction-diffusion modelling approaches (such as Turing patterns) by saying that such approaches lack the stochasticity observed inherently in pigment patterns.I would disagree with this viewpoint. The point of reaction-diffusion approaches is minimalistic coarse-grained descriptions of the complex effective interactions between cells (or any other constituents of the patterns) that can capture the essence of the emergent patterns. It is easy to incorporate stochasticity even in approaches that use partial-differential equations.

Whilst we acknowledge that Turing patterns are an excellent tool for inspiring scientists to investigate the mechanisms behind pattern formation in a variety of biological systems, we believe they are not the most appropriate tools for understanding the detailed mechanisms underlying zebrafish pigmentation patterning at the microscale.

A lack of stochasticity in classical partial differential equation models of Turing reaction-diffusion mechanisms is just one reason why we view such models as less appropriate for this inherently stochastic pattern formation process but is of relatively minor concern in comparison to the other drawbacks we perceive these models have. That being said, we acknowledge that reaction-diffusion models can incorporate stochasticity and we have now removed this comment from the manuscript.

More troubling, in our point of view, is the lack of cellular level resolution that such continuum models provide. In an age in which computational power is not an issue and cell-level biological data is available, individual-based models are an appropriate tool for investigating biological hypotheses made at a cellular level.

Even more concerning to us is the lack of detailed insight provided by coarse-grained Turing models. It is precisely the coarse-grained ‘effective’ descriptions of the complex interactions that the reviewer describes, which rob Turing models of zebrafish pigment pattern formation of their predictive power. It is often impossible, in such models, to link parameters specifically to the microscopic individual-level data, especially because many different types of reaction kinetics are capable of generating the same course-grained patterns. Consequently, Turing models are often unable to provide mechanistic insight into the underlying biological processes, something which our detailed, individual-based model is manifestly capable of doing.

Perhaps the starkest illustration of this limitation is that Turing models of zebrafish pigmentation patterns are capable of producing stripes with just two cell species – melanophores and xanthophores – whereas the real biology requires iridophores as well. Early two-species reaction-diffusion models of zebrafish pigmentation pattern formation gave no hint of the importance of the missing cell type or the necessity of the corresponding interactions, whereas detailed individual-based models such as ours, incorporating all the known biology, surely would have. That a model can produce a pattern which looks qualitatively similar to a biological pattern is not sufficient to suggest it can provide biological insight.

In summary, whilst we do not dispute that Turing reaction-diffusion models are useful for suggesting potential pattern formation mechanisms in a variety of biological contexts, and indeed were what first drew one of us, and many others, to become interested in understanding patterning processes many years ago, we do not believe them the most appropriate tool to test the complex individual-cell-based hypotheses that we investigate in this paper.

To address these comments, we have refined our comments about Turing mechanisms to the following:

“They [Turing mechanisms] typically rely on a small number of parameters which, upon being altered, can generate a diverse range of patterns. Simplified models such as these have the benefit that they are often analytically tractable, allowing a deep understanding of the pattern formation mechanism. However, their main limitation is that, due to the simplicity of the approach, there is often no consistent way to link the parameters with measurable data, making it difficult to relate the model results back to the biology.”

A complimentary approach is to simulate the underlying master equation that governs the chemical interactions, supplemented by a compartment-based approach to diffusion. There are several such studies done in the past using the standard Gillespie algorithm for simulating the chemical master equation. Such studies incorporate the essential chemistry (e.g., activator-inhibitor dynamics) and coupled with diffusion can lead to emergent stripe patterns with inherent stochasticity built-in.

Compartment-based reaction-diffusion simulations (often known as a reaction-diffusion master equation (RDME) formalism) are, as the reviewer points out, an alternative way to simulate reaction-diffusion processes which naturally incorporate stochasticity. Indeed, these models are capable of giving stripe formation consistent with the continuum models from which they are derived, but not, crucially, consistent with the biology underlying zebrafish pigmentation pattern formation for the following reasons:

1) In order to produce striped patterns, these RDME models must have multiple cells per compartment (as opposed to the single cell per compartment suggested by the volume-excluding property exhibited by the real cells). Such models typically also rely on reaction kinetics which bear little or no correspondence to the underlying biological interactions between different types of cells.

2) Once antiphase patterns of peaks and troughs have been produced by such models, an appropriately tuned threshold must be chosen in order to decide which regions constitute “stripe” and which “interstripe”. Consequently, the width of the stripes that result from such models is not an inherent property of the model, but an arbitrary choice on behalf of the modeller. The arbitrary threshold beyond which one region is considered stripe and another interstripe also belies the fact that these models result in non-trivial densities of xanthophores in the stripes and non-trivial densities of melanophores in the interstripes. This fact often goes unmentioned in such models but represents a significant departure from the real biology of the system.

Our approach shares some similarity to the RDME approach but incorporates more biological realism. Indeed, we use the Gillespie algorithm to simulate the stochastic events in our individual-based model, but rather than incorporating “the essential chemistry (e.g., activator-inhibitor dynamics)” which corresponds to a phenomenological Turing reaction-diffusion model that the reviewer suggests (NB ‘essential chemistry’ here is a reference to a required feature for the model-type, rather than an ‘essential known feature of the biological mechanism’), our detailed individual-based model begins with the biology. Consequently, we do not incur the limitations associated with Turing models that we have highlighted above.

3) The authors, instead, develop a detailed lattice-based model with sufficiently complex interactions between 5 species of cells along with short-ranged local interactions and long-ranged non-local interactions. These interactions bias the hopping rates of the cells along the lattice. The authors speculate on the possible physical origins of these interactions. However, it is not clear if there aren't other possible interactions that could also lead to the same emergent dynamics. And hence the questions arises as to what are the essential features required at this level of modeling and what details are incidental. It should be noted that even though the authors claim that their approach is bottom-up, they are not modelling interactions between molecules. Rather, the interactions incorporated in their modeling are also effective cell-level interactions. In essence, this study has considered a master-equation like approach to simulate zebrafish stripe patterns.

A bottom-up modelling approach does not require that modelling begins at the level of molecules (or, to go into even more detail, atoms), rather it should be at the lowest level at which pertinent details are available. Given the current state of biological knowledge, the level of cell-cell interactions is the lowest level at which we have sufficient information to build a model capable of recapitulating zebrafish pigmentation patterns. As a result, our model is capable of investigating biological hypotheses about interactions at this level, which higher-level, top-down models, focussed solely on pattern replication irrespective of the details of the underlying biology, would not be. Our model incorporates what is known of the biology and consequently provides a tool to investigate hypotheses about that which is unknown.

For the reasons we highlighted in the response to the reviewer’s previous point, our study is distinct from an RDME approach and avoids the biological unrealism associated with it. We consider fine-grained, biologically informed cell-level interactions as opposed to the coarse-grained interactions employed by the RDME approach.

Whilst it is possible that other cell-cell interaction dynamics might lead to similar emergent dynamics, the cell-cell interactions we have chosen are either biologically evidenced or (in the case of missing iridophore dynamics) previously hypothesised. The origins of the known cell-cell interactions are given in text [1-4]. Including alternative interactions with no basis in biological observation seems illogical.

4) The authors claim that all the interactions included in their model are essential for stripe patterns. This is demonstrated by turning off each interaction in-turn and showing that some feature of the pattern is lost. This is an 'all or none' switch. For the set of other parameters chosen, this might indeed lead to the conclusion that all interactions are absolutely needed. One could vary the interaction strengths in a continuous manner and then it is not clear if all the said interactions are absolutely needed for all parameter values.

An exhaustive continuous parameter sweep over all model parameters is not feasible. Such a parameter sweep would also be inappropriate since the majority of parameters are determined biologically. However, investigating the broad range of patterns that the model can produce is certainly a question of biological interest which we intend to follow up in a subsequent project.

5) Looking at the appendix and the supplementary material, I feel that the model is sufficiently complicated, and has so many turning parameters, that any range of behaviors is possible. It is not unreasonable to see that the large-scale emergent dynamics of such a complex model is essentially that of Turing systems which could have been simulated with much less complexity.

As we have previously stressed, the majority of the model’s parameters are biologically determined leaving the range of behaviours relatively restricted, although still capable of replicating the relevant biological patterns.

Without doubt, the basic striped patterns of wild-type zebrafish could be (and indeed has been) simulated with (simplified two-variable) Turing models. However, the point of our study is not simply to replicate patterns, but to provide mechanistic insight into the biology by which they are generated – something that Turing models are poorly equipped to do.

In order to demonstrate the robustness of our model to changes in the parameters we carried out a rigorous robustness analysis in which we varied the parameter values independently of each other. For 100 repeat simulations each parameter value was assigned at random from the range between 0.75-1.25 of their described value, effectively sampling the high-dimensional parameter space randomly. The process is described in detail in subsection “Necessity of S-iridophore assumptions” and twenty (randomly selected from the total of 100) repeats are displayed in Figure 9. It is clear to see, by looking at the figure, that the wild-type striped pattern is not significantly perturbed by any of the random parameters choice, demonstrating the robustness of the model to parameter changes. Note that we then repeated this robustness analysis for each of four of the key mutants, with similar confirmation of the robustness of the model’s outputs to reasonable parameter value changes. Twenty (randomly selected) of the repeats for WT are displayed in the new Figure 9 of the manuscript and ten (randomly selected) of the repeats for each of *nac*, *pfe*, *shd* and *leo* are given in Appendix 4—figure 1.

Reviewer #3:[…] The paper is well-written. However, I would like to make some suggestions, listed below.1) The assumptions and the way the model works could be made more explicit. For this I would suggest that the authors consider incorporating Figure 1, 2 and 3 from Appendix 1 into the main manuscript.

As suggested, we have added Appendix 1—figures 1, 2 and 3 to the main manuscript, reconstituting them as Figure 2 and Figure 3 to ease understanding. We thank the reviewer for this suggestion which has, we think, strengthened the manuscript by making it easier to understand.

2) I would suggest that the authors emphasise the differences of their model from the Volkening paper published in Aug 2018. I do take issue with the authors characterisation of this paper as very recent, and would suggest that the make greater reference to it in the Introduction.

We have changed ‘very recent’ to simply ‘recent’ where this appears. In the conclusion we had provided a paragraph addressing the similarities and differences between our work and Volkening’s, which we retain. To address the reviewers comment we have also added a sentence about the work of Volkening in the Introduction:

“These findings have paved the way for more detailed modelling, such as that of Volkening et al., 2018, who demonstrated (using an off-lattice individual-based model) the need for understanding S-iridophore behaviour when representing all three cell-types.”

3) One very interesting piece of data from this study, and the one that does differentiates this study from that of Volkening and Sandstede, is the analysis of the schachbrett mutant. This mutant points to the role of tight junctions in the transition of S-iridophore packing. It would be worth extending this analysis to the compunctiond sbr/leo and sbr/luc mutants described in Fadeev et al.

Upon the reviewer’s suggestion we incorporated the known features of *leo*, *leo;sbr*, *leo;pfe*, *leo;nac* and *leo;shd* mutants into our model. We found that, upon the incorporation of suitable assumptions we could, in fact, generate all the known patterns of these single and double mutants. We have stated the assumptions that we required in order to generate this range of distinctive patterns as a series of testable hypotheses about the function of the *leo* gene and have demonstrated the patterns that result when different combinations of these hypotheses are not satisfied in the model. We have included this now in the manuscript (see subsection “The position of the initial S-iridophore interstripe is important for successful pattern formation” and Figure 15). Additionally, we have tested the robustness of this pattern formation in Appendix 4—figure 1.

Additionally, there is a suggestion of an interaction between sbr and seurat – I wonder if the model were able to predict what a double sbr/seurat would look like?

The mutation in *seurat* affects melanocyte differentiation [3] and the mutation in *sbr* effects iridophore transition from dense to loose [5]. By changing the parameters to match this known biology we can effectively replicate the patterns associated with both *sbr* and *seurat* independently. Consequently, we can model and hence predict pattern formation for a double *seurat/sbr* mutant by changing the parameters so that the effects of both mutations occur simultaneously. We have added an example of a typical simulation result for a *seurat/sbr* at stage J+ in Appendix 5—figure 1. Our model predicts, when both of these mutations occur, that by stage J+, dense iridophores and xanthophores would cover most of the flank of the fish. A few melanophores associated with loose iridophores survive at the very dorsal and ventral regions of the fish. We have included a discussion of this double mutant along with the figure in in Appendix 5.

References:

1] Patterson, L. B., and Parichy, D. M. (2013). Interactions with Iridophores and the Tissue Environment Required for Patterning Melanophores and Xanthophores during Zebrafish Adult Pigment Stripe Formation. PLoS Genet., 9(5), 1-14.

2] Takahashi, G., and Kondo, S. (2008). Melanophores in the stripes of adult zebrafish do not have the nature to gather, but disperse when they have the space to move. Pigment Cell Melanoma Res., 21(6), 677–686.

3] Eom, D. S., Inoue, S., Patterson, L. B., Gordon, T. N., Slingwine, R., Kondo, S., Watanabe, M., and Parichy, D. M. (2012). Melanophore Migration and Survival during Zebrafish Adult Pigment Stripe Development Require the Immunoglobulin Superfamily Adhesion Molecule Igsf11. PLoS Genetics, 8(8), 1-16.

4] Singh, A. P., Schach, U., and Nüsslein-Volhard, C. (2014). Proliferation, dispersal and patterned aggregation of iridophores in the skin prefigure striped colouration of zebrafish. Nat. Cell Biol., 16(6), 607–614.

5] Fadeev, A., Krauss, J., Frohnhöfer, H. G., Irion, U., and Nüsslein-Volhard, C. (2015). Tightjunction protein 1a regulates pigment cell organisation during zebrafish colour patterning. *eLife*, 2015(4), 1–25.

[Editors' note: further revisions were suggested prior to acceptance, as described below.]

The manuscript has been improved but there are some remaining issues that need to be addressed before acceptance, as outlined below:1) In the revised manuscript, you write:"The most commonly used mathematical paradigm for stripe formation takes the form of a Turing reaction diffusion model. In these representations, melanocytes and xanthophores diffuse and interact via a few long and short range 'reactions'. This class of model typically rely on a small number of parameters which, upon being altered, can generate a diverse range of patterns. Simplified models such as these have the benefit that they are often analytically tractable, allowing a deep understanding of the model. However, their main limitation is that, due to the simplicity of the approach, there is often no consistent way to link the parameters with measurable data, making it difficult to relate the model results back to the biology."This gives the impression that (i) Turing pattern models are always simplified, and analytically tractable, and (ii) Turing models face a difficulty in relating their parameters to measurable data. Both are untenable statements and should be removed – there are several examples of both highly nonlinear Turing-type or reaction-diffusion models for pattern formation that are neither simple nor analytically tractable, and in several cases now people have shown how to relate their parameters to measurable data.

We appreciate the reviewer’s considered points and consequently have rectified the potential ambiguity of our statements to reflect the reviewer’s concerns. We have changed the text as follows:

“The most commonly used mathematical paradigm for stripe formation takes the form of a Turing reaction diffusion model. […] However, a potential limitation is that parameters do not always have a clear biological interpretation which, can sometimes make it difficult to link parameters to measurable data.”

2) The authors write:"Turing reaction-diffusion type models posit that combinations of short and long range dynamics between melanocytes and xanthophores generate stripe patterns. Indeed, much of the excitement around such models is the ease with which small parameter value changes result in diverse patterns, many readily recognisable from nature. […] A major difference between our model and Turing reaction diffusion models is that small parameter changes in our model do not typically generate qualitatively different patterns, whereas Turing reaction diffusion models can show substantial pattern changes in response to small changes (Budi, Patterson and Parichy, 2011; Yamaguchi, Yoshimoto and Kondo, 2007)."This gives the impression that large changes in behaviour due to small changes in parameters is a generic feature of Turing models, as opposed to the model in this manuscript. This is not true, and the statements implying this should be removed – it is quite possible for Turing and reaction-diffusion models to produce small changes upon small changes in parameters, and equally it is possible for stochastic lattice models to exhibit large changes in behaviour upon small changes in parameters – it depends on the interactions and nonlinearities included in the model.

Whilst in some cases small parameter value changes specifically near bifurcation points can result in diverse patterns for Turing patterns (see Lee et al., 2012; Metz et al., 2011; Watanabe and Kondo, 2015)), we acknowledge that it is not a generic feature of all Turing models. We appreciate that these sentences gave a misleading impression and for this reason we have added ‘can sometimes’ to the first paragraph as follows:

"Turing reaction-diffusion-type models posit that combinations of short and long range dynamics between melanocytes and xanthophores generate stripe patterns.” Indeed, a lot of the excitement around such models is the ease with which small parameter value changes can sometimes result in diverse patterns, many readily recognisable from nature.

Furthermore we have modified the following sentence to be more specific:

“A major difference between our model and Turing reaction-diffusion models is that small parameter changes in our model do not typically generate qualitatively different patterns, whereas Turing reaction-diffusion models can show substantial pattern changes in response to small alterations to parameters near to bifurcation points (Budi, Patterson and Parichy, 2011; Yamaguchi, Yoshimoto and Kondo, 2007)."

3) Finally:"From an analytical perspective, a significant advantage of our on-lattice model in contrast to off-lattice models is their amenability to the derivation of a continuum model. Our model therefore opens up the opportunity for future exploratory work using a continuum model for mutants pfe and nac in order to explore whether pattern formation in these cases individually can be described as Turing patterns and todetermine parameter ranges for successful pattern formation."Here you argue, and we agree, that the model in the manuscript is in fact simply a discretized version of a reaction-diffusion model (albeit a quite complex one with many dynamical variables and many interactions). Thus, there is no great difference in approach between the discretized stochastic model you analyze and reaction-diffusion models, so please remove all statements that imply a large difference in your approach vs. reaction-diffusion or Turing type models.

We agree with the reviewer that we can likely derive a continuum reaction-diffusion model from our on-lattice individual model, but we note that this will not necessarily be a *Turing* reaction-diffusion model. It is for this reason that we express interest in deriving the continuum model, i.e. to assess this. Furthermore, being able to derive a continuum model from an individual-based model does not mean that the individual-based model is simply a discretisation of the resulting continuum model. Undoubtedly, in complex individual-based models which employ non-linear interactions, approximations including (but not limited to) moment closure assumptions will have to be made in order to reach the continuum limit, meaning that the continuum model will exhibit different behaviour to the individual-based model in certain parameter regimes. The advantage of deriving a continuum reaction-diffusion model from an individual-based model, rather than the other way around, is that we know exactly what assumptions have been made in passing from the individual-based model to the continuum model.

Further, a continuum limit can easily be constructed for off-lattice models as well, so this is not an advantage of your approach over that of Volkening et al. Please remove this statement, and present the opportunity for building continuum models as one that applies to both your model as well Volkening et al.'s model.

We acknowledge and address this concern as follows:

"From an analytical perspective, an advantage of our on-lattice model is its amenability to the derivation of a continuum model, although we note that continuum approximations to off-lattice individual-based models can also be derived. Our model therefore opens up the opportunity for future exploratory work using a continuum model for mutants *pfe* and *nac* in order to explore whether pattern formation in these cases individually can be described as Turing patterns and to determine parameter ranges for successful pattern formation."

That it is also possible to derive continuum limits from off-lattice models does not diminish the utility of the approach for our on-lattice models. That said, we have moved this statement, so that it fits within our summary of future work instead of our comparison with Volkening et al. as we believe its previous positioning gave a misleading impression of the purpose of the statement. Having checked through the manuscript carefully, we do not see anywhere else where we claim that our model has this advantage over the Volkening model.